# WIDESEARCH: BENCHMARKING AGENTIC BROAD INFO-SEEKING

**Ryan Z.Y. Wong**[*], **Jiawei Wang**[*], **Junjie Zhao, Li Chen, Yan Gao, Long Zhang, Xuan Zhou, Zuo Wang, Kai Xiang, Ge Zhang, Wenhao Huang, Yang Wang**[†], **Ke Wang**[†]
Bytedance Seed
https://widesearch-seed.github.io

## ABSTRACT

From professional research to everyday planning, many tasks are bottlenecked by wide-scale information seeking, which is more repetitive than cognitively complex. With the rapid development of Large Language Models (LLMs), automated search agents powered by LLMs offer a promising solution to liberate humans from this tedious work. However, the capability of these agents to perform such "wide-context" collection reliably and completely remains largely unevaluated due to a lack of suitable benchmarks. To bridge this gap, we introduce WideSearch, a new benchmark engineered to evaluate agent reliability on these large-scale collection tasks. The benchmark features 200 manually curated questions (100 in English, 100 in Chinese) from over 15 diverse domains, grounded in real user queries. Each task requires agents to collect large-scale atomic information, which could be verified one by one objectively, and arrange it into a well-organized output. A rigorous five-stage quality control pipeline ensures the difficulty, completeness, and verifiability of the dataset. We benchmark over 10 state-of-the-art agentic search systems, including single-agent, multi-agent frameworks, and end-to-end commercial systems. Most systems achieve overall success rates near 0%, with the best performer reaching just 7%. However, given sufficient time, cross-validation by multiple human testers can achieve a near 100% success rate. These results demonstrate that present search agents have critical deficiencies in large-scale information seeking, underscoring urgent areas for future research and development in agentic search.

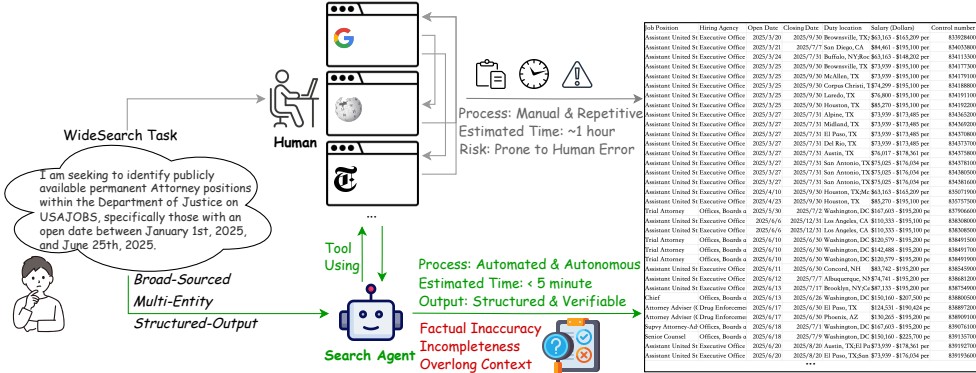

Figure 1: A conceptual comparison of manual and agent-based approaches for WideSearch tasks. It demonstrates the labor-intensive nature of the manual approach and the failure modes of agent approach, which calls for a systematic evaluation to quantify agent performance and reliability.

[*]Co-first authors. [†]Co-corresponding authors. Correspondence: ryan.wong.066@gmail.com.

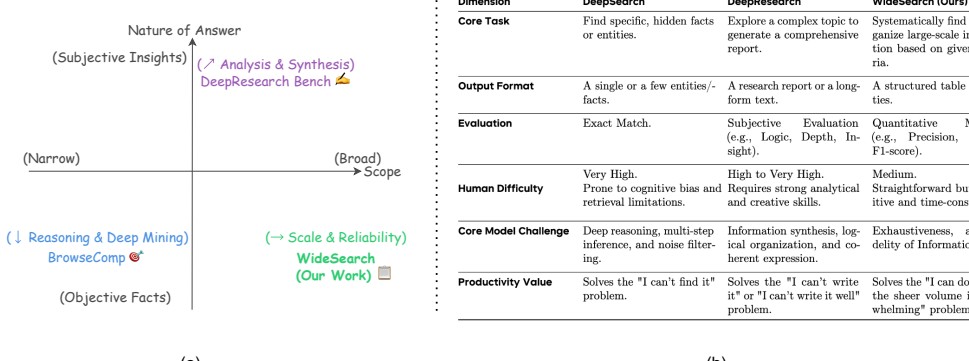

Figure 2: An overview and detailed comparison of DeepSearch, DeepResearch, and our WideSearch. (a) illustrates the high-level relationships and operational domains of the three paradigms. (b) provides a detailed breakdown, contrasting them across key dimensions including core tasks, evaluation methods, and primary value propositions.

# 1 INTRODUCTION

With the advent of advanced agentic frameworks such as OpenAI DeepResearch (OpenAI, 2024) and Manus (Manus, 2025), the development of agent systems based on Large Language Models (LLMs) is entering its second half, where the focus is rapidly shifting from demonstrating novel capabilities to achieving practical, real-world reliability. This transition is driven by a fundamental recognition of the inherent limitations in standalone models: their finite parameters make it impossible to store all knowledge, the prohibitive cost of retraining makes them lag behind real-time information, and they naturally struggle with long-tail or specialized facts. Consequently, in this evolving domain, the ability to effectively utilize search tools has become paramount. The most critical question in this race is no longer just what an agent can do, but how we can measure and improve its ability to leverage search in authentic user scenarios to deliver tangible value and drive meaningful product iteration.

Our in-depth analysis of real-world user queries reveals a significant gap: a common and critical class of information-seeking tasks is not adequately evaluated by existing agent benchmarks. We term this category WideSearch, which involves tasks that require an agent to thoroughly and accurately acquire all large-scale atomic information meeting a series of criteria, and then arrange it in a well-organized output. For example, a financial analyst may need to find all companies in a sector that meet specific revenue and growth criteria, or a job seeker may need to find all job vacancies that match their criteria for role, location, and experience level. For humans, executing such tasks is excruciatingly tedious; as depicted in Figure 1, the transition from this laborious manual process to an automated agent workflow promises immense efficiency gains, but also introduces new failure modes that demand systematic evaluation. Consequently, WideSearch carves out a distinct problem space. As illustrated in Figure 2, it diverges from DeepSearch, which targets the "I can't find it" problem of locating specific, hard-to-find facts, and DeepResearch, which addresses the "I can't write it well" problem of synthesizing complex narratives. Instead, WideSearch tackles tasks whose primary challenge is not cognitive difficulty but operational scale and fidelity—the "I could do it, but the sheer volume is overwhelming" problem—a domain largely overlooked by current benchmarks.

To systematically evaluate this paradigm, we introduce WideSearch, the first benchmark specifically designed for this purpose, supported by a sophisticated multi-stage data collection and verification framework, as well as a hybrid automated evaluation system that ensures objectivity. Benchmarking more than 10 state-of-the-art agent systems reveals a stark reality: current systems are profoundly challenged by the demands of comprehensiveness and fidelity at scale. The overall success rate is exceptionally low, with even the top-performing multi-agent framework achieving a mere 5.1%, and individual humans also struggling at 20%. Our key insight, derived from a test-time scaling analysis, is that this failure does not stem from an inability to find individual facts—item-level F1 scores can approach 80% with sufficient retries. Rather, the bottleneck is that we must ensure the absolute

completeness and accuracy of each atomic unit of information within a large-scale search. Any single data omission or error, or the integration of extra data into the final result, results in total failure of the task execution. Our detailed error analysis traces this failure to fundamental deficiencies in advanced agentic capabilities, such as incomplete planning, a lack of reflection to iterate on failed searches, and the inability to correctly use retrieved evidence. These findings provide a clear roadmap, suggesting that future progress hinges on developing more sophisticated agent models and architectures, particularly multi-agent frameworks that enable parallel search and cross-validation, mimicking the collaborative human processes required to tackle these complex, large-scale tasks.

## 2 WideSearch Benchmark

The construction of the WideSearch benchmark is a meticulous, human-centered process designed to ensure that each task is challenging, realistic, and aligned with our goal of evaluating wide-context information gathering. The entire workflow, from question design to final inclusion, is governed by a strict set of criteria and a multi-stage validation protocol.

### 2.1 Task Definition

The fundamental task in the WideSearch benchmark challenges an LLM agent to act as a diligent information seeker. Given a complex natural language query and a predefined table schema, the agent's objective is to populate the table by systematically gathering, synthesizing, and verifying information from the live web. This emulates real-world information-seeking scenarios that require discovery and aggregation rather than simple fact retrieval. Formally, each task instance in WideSearch is defined by a tuple $(Q, S)$, where:

**A Query ($Q$):** A natural language question that implicitly specifies a set of target entities and the information required about them. For example, $Q$ could be: "I want to apply for Master's programs in civil engineering starting in 2026. Could you help me find the minimum GPA requirements for admission to Ivy League institutions in the US and Group of Eight universities in Australia?"

**A Table Schema ($S$):** A predefined structure $S = \{C_1, C_2, \ldots, C_m\}$, where each $C_j$ is a column header representing an attribute (e.g., 'Country', 'University', 'Alliance', 'Minimum GPA Requirement'). The schema defines the exact structure of the required output for objective evaluation.

The agent's goal is to interact with a web environment, primarily via search tools, to produce a final, populated table, $T_{\text{agent}}$. This objective decomposes into two primary challenges:

**Entity Set Identification:** The agent must first identify the complete and correct set of entities, $E = \{e_1, e_2, \ldots, e_n\}$, that satisfy the constraints of the query $Q$. In this example, the entities are the 8 Ivy League institutions and the 8 Group of Eight universities. This tests the agent's ability to conduct a comprehensive search across different domains (US and Australian higher education).

**Attribute Filling:** For each identified entity $e_i \in E$, the agent must find corresponding values for each attribute $\{C_1, C_2, \ldots, C_m\}$ defined in the schema $S$, sourcing the information from web pages.

The final output, $T_{\text{agent}}$, is therefore a table with $n$ rows and $m$ columns (as defined by $S$), where each cell $T_{\text{agent}}(i, j)$ contains the value of attribute $C_j$ for entity $e_i$. The quality of this output is then measured against a ground-truth table to assess its completeness and factual accuracy.

### 2.2 Task Construction Principles

The construction of tasks within the WideSearch benchmark is guided by a rigorous, principled methodology to ensure their quality, relevance, and alignment with the challenges of wide-context information seeking. Each task is manually curated by domain experts and must satisfy the following six fundamental principles:

**High Search Volume and Breadth**: Tasks are defined by their extensive information breadth, requiring the agent to collate numerous distinct data points across multiple entities. This inherent breadth necessitates a high volume of search interactions and a prolonged, multi-step procedural trajectory, distinguishing them from tasks that require only a singular, deep line of inquiry.

**Temporal and Contextual Invariance**: The ground-truth answers exhibit high stability. They are static over time and are independent of geographical, ideological, or socio-cultural contexts, thereby guaranteeing the benchmark's long-term validity and global applicability.

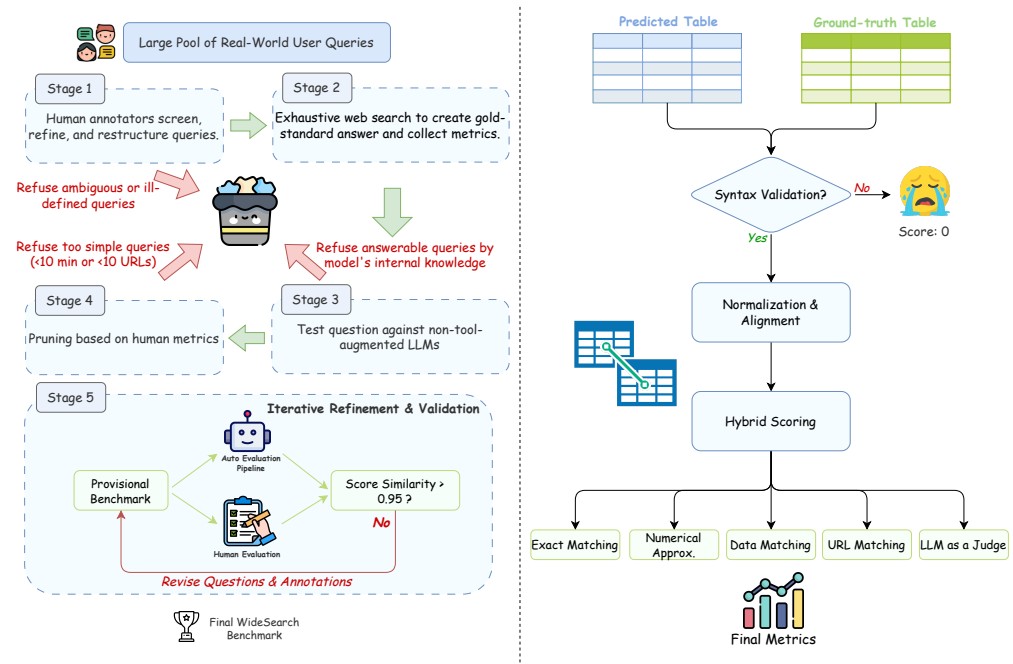

Figure 3: An overview of our integrated data pipeline, detailing the five-stage data curation and validation pipeline (left), and the automated evaluation pipeline (right).

**Objective Verifiability**: Each task is associated with a deterministically verifiable set of facts. This allows for objective, consistent, and reproducible scoring against a predefined gold standard.

**Public Accessibility**: The entire corpus of information required to formulate a complete answer is guaranteed to be publicly accessible via standard web search engines, ensuring that tasks are solvable without privileged access to information.

**Reliance on External Tools**: Tasks are explicitly designed to exceed the bounds of an LLM's parametric knowledge. Successful completion is therefore contingent upon the agent's ability to engage in active, iterative, and effective web search, rather than relying on memorized information.

**Scenario Diversity**: The benchmark encompasses a heterogeneous collection of scenarios. This cross-domain diversity ensures that we evaluate generalizable agent capabilities, such as planning and synthesis, rather than task-specific or domain-dependent knowledge.

## 2.3 DATA CURATION AND VALIDATION PROCESS

We implement a multi-stage data curation and validation pipeline (shown in Figure 3). It transforms real-world information needs into standardized, high-quality tasks. A task is only accepted into the final benchmark after successfully passing all stages and the final iterative validation loop.

**Sourcing and Refinement of Candidate Questions**: We start with a large pool of real user queries from diverse domains. Human annotators screen these raw queries, refining promising candidates into clear, unambiguous questions that align with our design principles.

**Gold Standard Annotation and Metric Collection**: For each question, an annotator conducts an exhaustive web search to create a gold-standard answer. During this process, they record key metrics: time to completion, number of search queries, and the count of unique web pages consulted.

**Parametric Knowledge Filtering**: To ensure tasks necessitate tool use, we test each question against powerful LLMs. If any model can answer correctly from its internal knowledge, the question is discarded, guaranteeing our benchmark evaluaes genuine search and synthesis capabilities.

**Difficulty-Based Pruning**: To control the quality of the dataset, we enforce a minimum complexity threshold using the human-collected metrics. Tasks that an annotator completes in less than 10 minutes or by consulting fewer than 10 unique web pages are discarded.

**Iterative Refinement and Validation**: Tasks that pass the initial filters undergo a final validation loop to align automated scoring with human judgment. We generate a response for each task using a commercial agentic system and have it rated by both our evaluation system (details in Section 2.4) and human experts. If the scores show a discrepancy (i.e., similarity is below a 95% threshold), the task is revised. This cycle continues until our automated metrics reliably predict human assessment.

This five-stage pipeline ensures every WideSearch task is complex, verifiable, grounded in real-world needs, and resistant to memorization. Most importantly, it calibrates our automated evaluation to be predictive of human judgment, creating a robust and reliable testbed for advanced search agents. An **illustrative example** of a final task is provided in Appendix C. Moreover, the **benchmark composition and statistics** are shown in Appendix D.

## 2.4 EVALUATION FRAMEWORK AND METRICS

To facilitate an accurate, scalable, and nuanced assessment of agent performance, we develop a comprehensive evaluation framework centered around an automated scoring pipeline. This framework is designed to handle the structured nature of our ground-truth data—which is stored in tables—and to address the inherent complexities of natural language responses.

### 2.4.1 AUTOMATED EVALUATION PIPELINE

To ensure robust and scalable evaluation, we designed an automated pipeline that formalizes scoring as table alignment and cell-wise verification. As depicted in Figure 3, our hybrid approach combines deterministic rule-based checks with semantic judgments from an LLM-as-a-judge. We use GPT-4.1-2025-04-14 as default judge model. The entire process is grounded in annotated data containing pre-defined primary keys for row alignment and column-specific methods for cell scoring.

**Syntax Validation and Alignment:** The pipeline begins with syntax validation. A response receives a score of zero if it is not a parsable Markdown table or if its column headers do not semantically match the ground truth in number and name; minor string differences are aligned using a mapping prompt (Appendix H). Passing tables are then normalized. To enable alignment, we use a mapping prompt to standardize entities in the primary key columns. Finally, the predicted table is joined with the ground-truth table on these keys to identify matched rows, false positives, and false negatives.

**Hybrid Item-level Scoring:** For each pair of aligned rows, we iterate through the corresponding cells. The evaluation method for each cell is dictated by its column's pre-annotated type, enabling nuanced and accurate scoring. Our framework supports a comprehensive set of categories: **(1) Exact Match**: For strings where absolute precision is paramount. **(2) Numerical Approximation**: To validate numbers while allowing for minor, acceptable floating-point or formatting variations. **(3) Date Matching**: To semantically compare dates that may appear in different but equivalent formats (e.g., "July 4th, 2024" vs. "2024-07-04"). **(4) URL Matching**: To normalize and validate the correctness of web links. **(5) LLM-as-a-judge**: Reserved for complex cases with high lexical variation (such as translated names or nuanced descriptions) that require semantic understanding for a fair assessment. The prompt of the LLM-as-a-judge is shown in Appendix H.

### 2.4.2 EVALUATION METRICS

We assess agent-generated tables using three metrics of increasing granularity. The most stringent, **Success Rate (SR)**, is a binary metric requiring the predicted table to be a perfect match to the ground-truth table in both content and structure. For a more detailed analysis, the **Row-level F1 Score** evaluates performance by treating each row as a fundamental unit, comparing the set of predicted rows ($P_{rows}$) against the ground-truth rows ($G_{rows}$) to calculate precision and recall. For the most fine-grained assessment, the **Item-level F1 Score** considers each individual cell as the basic unit, computing precision and recall based on the multiset of items in the predicted table ($P_{items}$) versus the ground-truth table ($G_{items}$).

Furthermore, to provide a more comprehensive evaluation, we perform N independent runs for each task and report performance using three aggregation strategies: **Avg@N**: This metric calculates the arithmetic mean of a given score (e.g., Success Rate, F1 Score) over N trials for a task. **Pass@N**: This metric measures the percentage of tasks that were solved successfully at least once in N trials.

It is used specifically for Success Rate. **Max@N**: For metrics like the F1 Score, this takes the single highest score achieved across the N trials for each task, and then averages these maximum values over all tasks in the dataset.

## 3 EXPERIMENTS

### 3.1 EXPERIMENTAL SETUP

To evaluate agent capabilities on our WideSearch benchmark, our experiments include three aspects: Single Agent, Multi-Agent Framework and End-to-End systems. All agents use a search tool and a webpage reader. We intentionally use naive agent architectures with minimal heuristic design to test the models' core capabilities. Appendix B lists all model API identifiers. The details of the agents are shown in Appendix G.

**Single Agent.** We first benchmark the intrinsic problem-solving abilities of individual LLMs. In this baseline setup, a single LLM is responsible for the entire task. The evaluated models are: DeepSeek-R1 (Guo et al., 2025), Doubao-Seed-1.6 (Thinking and Non-Thinking) (Seed et al., 2025), Claude Sonnet 4 (Thinking) (Anthropic, 2025), Gemini 2.5 Pro (Google Gemini, 2025b), Kimi K2 (Team et al., 2025), Qwen3-235B-A22B-Thinking-2507 (Yang et al., 2025), OpenAI GPT-5 (OpenAI, 2025a), and OpenAI o3 (OpenAI, 2025b).

**Multi-Agent Framework.** To test a "divide-and-conquer" strategy, we evaluate the same foundation models within a multi-agent framework. A main agent decomposes the query and aggregates results from multiple sub-agents that execute sub-tasks in parallel. This allows a direct comparison of the architectural impact versus the single-agent paradigm.

**End-to-End System.** To contextualize performance, we benchmark against the integrated web-browsing modes of leading commercial systems: Gemini 2.5 Pro, Claude Sonnet 4 (Thinking), and OpenAI o3. We manually get the results from the web interface.

**Human Evaluation.** Annotating the ground truth for WideSearch is an arduous process. To gauge human performance, we had 10 annotators each solve two randomly selected questions (one in Chinese, one in English) from a set of 20. Participants were given unlimited time and could use any tools, including AI assistants, to ensure their answers were complete.

### 3.2 MAIN RESULTS

We report the main experiment results in Table 1 and the complete results in Table 6. We provide two typical example trajectories in Appendix K. The conclusions are obtained as follows:

**Existing models still lack the advanced agentic abilities.** Current advanced large language models show fundamental weaknesses when performing large-scale information-seeking tasks, with failures stemming from fundamental cognitive deficits beyond simple search inaccuracies. They exhibit poor planning by struggling to break down complex questions into comprehensive sub-queries, which leads to incomplete information seeking. Furthermore, they lack reflection and fail to dynamically adjust their search strategy when initial attempts are unsuccessful, often giving up or answering with insufficient data instead of trying new methods. Even when they successfully find relevant information, they demonstrate faulty evidence use by misinterpreting or misattributing the content. These basic deficiencies in planning, dynamic adjustment, and reasoning are the primary reasons for their extremely low success rates on such complex tasks.

**Multi-agent mode outperforms the single-agent mode on WideSearch.** The multi-agent framework, using a "divide-and-conquer" strategy, consistently and significantly outperforms the single-agent mode on WideSearch tasks by more effectively addressing their inherent breadth. Although absolute success rates are low for both, the multi-agent system shows a distinct advantage in F1 scores, which measure partial correctness. This superior performance is due to its architecture, where a planner decomposes a broad query into parallel sub-tasks assigned to different agents. This parallel search and division of labor not only improves the breadth and efficiency of information seeking but also mimics the specialized, collaborative process of human expert teams, making the framework better suited for complex, wide-ranging searches.

**Current commercial AI assistants cannot yet seek information at a large scale.** Although top commercial AI models have some information retrieval capabilities in their integrated web Browse

Table 1: Main results on the WideSearch benchmark. We report Success Rate (SR), Row-level F1, and Item-level F1 for all evaluated systems. All scores are reported as percentages (%). We use **boldface** to indicate the highest-performing metric in each mode, and underlining to mark the second-highest-performing metric.

| Model / System | Success Rate (%) | | Row F1 Score (%) | | Item F1 Score (%) | | # Tool Calls | |
|---|---|---|---|---|---|---|---|---|
| | Avg@4 | Pass@4 | Avg@4 | Max@4 | Avg@4 | Max@4 | Search | Web Browse |
| *Single Agent* | | | | | | | | |
| GPT-5 | **6.9** | **13.5** | **37.9** | **52.2** | 54.0 | **68.2** | **25.64** | **14.27** |
| OpenAI o3 | 4.5 | 9.0 | 34.0 | 44.1 | 52.6 | 62.3 | 13.26 | 5.75 |
| Claude Sonnet 4 (Thinking) | 2.3 | 5.0 | 31.7 | 41.9 | **57.9** | **66.7** | 8.20 | 3.42 |
| Gemini 2.5 Pro | 1.5 | 5.0 | 30.0 | 41.4 | 51.0 | 63.6 | 7.48 | 1.58 |
| Kimi K2 | 1.1 | 3.5 | 29.7 | 41.4 | 54.4 | 65.1 | 10.78 | 2.22 |
| Qwen3-235B (Thinking) | 0.5 | 1.5 | 20.6 | 30.8 | 44.9 | 55.5 | 3.01 | 2.61 |
| DeepSeek-R1 | 0.4 | 1.5 | 20.7 | 31.7 | 41.3 | 55.1 | 2.91 | 1.40 |
| Doubao-Seed-1.6 (Thinking) | 2.6 | 5.0 | 30.0 | 44.1 | 48.3 | 63.9 | 22.08 | 1.14 |
| Doubao-Seed-1.6 (Non-Thinking) | 1.0 | 3.5 | 27.2 | 39.9 | 49.0 | 62.0 | 8.01 | 1.82 |
| *Multi-Agent Framework* | | | | | | | | |
| GPT-5 | **7.3** | **12.0** | **42.5** | **54.2** | 59.7 | **74.5** | 37.50 | **22.00** |
| OpenAI o3 | 5.1 | 9.5 | 37.8 | 50.5 | 57.3 | 68.9 | 26.72 | 16.29 |
| Claude Sonnet 4 (Thinking) | 3.6 | 6.5 | 38.5 | 52.2 | **62.2** | 73.1 | 27.64 | 11.60 |
| Gemini 2.5 Pro | 2.0 | 6.5 | 33.5 | 44.6 | 57.4 | 66.3 | 20.73 | 4.72 |
| Kimi K2 | 3.0 | 6.5 | 36.2 | 49.6 | 61.2 | 70.7 | 28.79 | 8.85 |
| Qwen3-235B (Thinking) | 0.6 | 2.0 | 25.8 | 37.4 | 51.9 | 62.6 | 13.72 | 14.52 |
| DeepSeek-R1 | 0.8 | 3.0 | 22.9 | 36.6 | 44.3 | 60.3 | 11.81 | 7.02 |
| Doubao-Seed-1.6 (Thinking) | 2.5 | 5.5 | 34.0 | 48.9 | 54.6 | 69.7 | **52.34** | 6.44 |
| Doubao-Seed-1.6 (Non-Thinking) | 2.1 | 4.5 | 29.7 | 42.7 | 52.8 | 65.1 | 14.83 | 5.18 |
| *End-to-End System* | | | | | | | | |
| OpenAI o3 | 3.0 | 5.5 | 23.9 | 36.0 | 45.5 | 56.5 | - | - |
| Claude Sonnet 4 (Thinking) | 2.5 | 5.0 | 24.1 | 33.5 | 48.4 | 58.5 | - | - |
| Gemini 2.5 Pro | **4.3** | **8.0** | **36.6** | **45.4** | **59.1** | **67.2** | - | - |
| Human | 20.0 | | 69.2 | | 82.4 | | - | |

modes, the experiment results show that they still struggle with WideSearch. Several leading commercial models tested in the experiment, including Gemini 2.5 Pro, Claude Sonnet 4, and OpenAI o3, hover around a 5% table-level success rate. Furthermore, in the early stages of the experiment, we found that some specialized DeepResearch systems tend to generate lengthy reports rather than the single, well-formatted table required by the task. It demonstrates that the design of current mainstream AI assistants has not yet been optimized for large-scale information gathering, and they lack the stability and precision required to become reliable productivity tools.

**Even humans cannot achieve a high success rate in single-player mode.** Experimental results show that even when given ample time and access to any tools, the success rate for a single individual completing the task independently is merely 20%. This outcome highlights the inherent difficulty of the task itself. A key characteristic of WideSearch tasks is the extreme density of data points, which means a complete answer may contain thousands of individual facts. Under these circumstances, any minor error (an extra, a missing, or an incorrect piece of data) results in the failure of the entire task. The creation of the ground truth is an arduous process requiring multiple annotators and extensive cross-validation. This high standard confirms that WideSearch is a challenging yet reasonable benchmark for evaluating the robustness and completeness of search agents.

For a detailed domain-level performance analysis, please refer to Appendix J.

### 3.3 CONSISTENCY WITH HUMAN EVALUATION

To validate the stability and reliability of our proposed automated evaluation pipeline, we conduct a consistency analysis against human assessment. For this analysis, we curate an evaluation set of 200 responses by randomly selecting one output from the pool of commercial agentic systems for each task in WideSearch.

Table 2: Consistency between our evaluation pipeline using different judge models and human evaluation. (%)

| Judge Model | Consistency Score |
|---|---|
| OpenAI o4-mini | 98.3 |
| Gemini 2.5 Pro | 98.1 |
| GPT-4.1 | 98.0 |
| Doubao-Seed-1.6 (Non-Thinking) | 97.8 |

These selected responses are then meticulously annotated by human experts to determine their item-level correctness against the ground truth.

Subsequently, we utilize our evaluation pipeline, employing different models as judges, to assess the same set of responses. The primary objective is to measure the degree of agreement between our automated pipeline's judgments and the human-annotated labels. The results of this comparison are presented in Table 2. As shown, the consistency between our pipeline's evaluation and human assessment is exceptionally high, exceeding 97.8% for all tested judge models, including both thinking and non-thinking variants. This high level of correlation underscores the effectiveness and reliability of our proposed evaluation methodology. Furthermore, it reinforces the objective nature of the WideSearch benchmark, demonstrating that performance can be assessed accurately and consistently without being subject to the variability of human evaluation.

The prompt of LLM-Judge is designed through multiple rounds of iteration. After manual inspection, the consistency between the evaluations of LLM-Judge and human evaluations has reached a very high level. Unfortunately, there are still some inconsistent judgments between the LLM and human evaluators. For example, regarding "carlosslimhelu family" and "carlosslimhelú", human annotators consider them consistent and believe the score should be 1. However, when the LLM conducted the evaluation, it output: "Standard answer: 'carlosslimhelufamily'; Answer provided: 'carlosslimhelú'. The word 'family' is missing, making it insufficient to determine factors such as children. Score: 0". In the current practice of LLM-Judge, such cases are usually unavoidable.

## 4    ANALYSIS

To gain a deeper understanding of the core challenges that current models face on the WideSearch benchmark, we conduct a systematic analysis of the experimental data and failure cases. We categorize the primary failure modes into two main groups: 1) Challenges in Advanced Agentic Capabilities, which reflect fundamental deficiencies in complex cognitive skills such as planning, reasoning, and synthesis; and 2) Basic Failure Modes, which arise from the model's inability to reliably execute explicit instructions or tool-use protocols. This classification not only highlights the technical bottlenecks of current search agents but also provides clear directions for future research.

### 4.1    CHALLENGES IN ADVANCED AGENTIC CAPABILITIES

In large-scale information gathering scenarios (i.e., WideSearch), the balance between Precision and Recall remains a core challenge, which is consistent with the challenges faced in traditional information retrieval tasks. Experiments indicate that the model's performance is far from optimal, both at the row-level and item-level evaluations. A particularly prominent phenomenon is that Recall is significantly lower than Precision across all test subsets, as shown in Table 6. This finding reveals a critical deficiency in the current model's ability to capture comprehensive information, identifying that "inadequate recall" is the primary bottleneck. Through an in-depth review of the Agent's reasoning process, we have identified the following key failure patterns:

**Incomplete Query Decomposition.** Agents often fail to fully break down complex user intents into sufficient sub-queries. This incomplete decomposition causes them to miss key information, highlighting a weakness in complex task planning. We provide an example in Figure I.1.

**Lack of Reflection and Iterative Refinement.** Following a failed tool call, agents tend to give up rather than reflect and adjust their search strategy (e.g., by rewriting queries). They answer with incomplete data, showing a deficiency in adaptive planning. We provide an example in Figure I.2.

**Failure in Evidence Utilization.** Agents struggle to ground final answers in retrieved evidence. They may misinterpret, ignore, or misapply sources, revealing a critical gap between information retrieval and generation. We provide an example in Figure I.3.

**Knowledge Hallucination and Factual Inconsistency.** When a search returns no results, agents sometimes turn to internal knowledge (which might be outdated or incorrect), leading to hallucinations and factual errors. It means the model may fabricate non-existent facts. This underscores the challenge of strictly grounding outputs in verifiable sources. We provide an example in Figure I.4.

We manually examined the trajectories that ultimately succeeded in generating result tables but produced tables inconsistent with the ground truth tables. The results showed that nearly all unsuc-

cessful trajectories exhibited deficiencies in the four aforementioned advanced agent capabilities, particularly in tasks involving large volumes of atomic information.

## 4.2 BASIC FAILURE MODES

In addition to the sophisticated agentic deficiencies described above, we also catalog a series of more basic failures where the model failed to generate the desired output. These errors often lead directly to the termination of the task workflow. We have statistically analyzed the distribution of basic error patterns and presented it in the Table 7.

**Tool Invocation Error.** This is one of the most common failures, typically caused by the model generating incorrect parameter formats, omitting necessary arguments, or attempting to call a non-existent tool, leading to an API call failure.

**Output Formatting Error.** Some models fail to strictly adhere to the output format specified in the instructions, such as failing to generate a Markdown table or producing a malformed one. It sometimes generates the final results in json format or unstructured format.

**Context Length Exceedance.** The task is prematurely terminated because the model generated overly verbose intermediate steps or became trapped in ineffective loops, causing the total input to exceed the model's maximum context window. The core reason is overthinking.

**Response Refusal.** The model exhibits refusal behaviors for some queries. We have identified two primary patterns of refusal: 1. The model perceives ambiguity in the user's question and consequently requests further clarification to narrow the scope of the inquiry. 2. The model deems the required information too extensive to be presented in a single output, leading to a direct refusal.

## 5 TEST-TIME SCALING

Allocating more compute resources during testing is a common method for probing the upper limits of a model's performance. In the experiments in this section, we use Kimi K2 as the foundation model, equipped with Search and Web Browse tools. Based on the single-agent mode, we attempt each question N times (where N is expanded from 1 to 128) and record the Success Rate (Pass@N), Row-level F1 Score (Max@N), and Item-level F1 Score (Max@N).

As shown in Figure 4, the Item-level F1 score shows a significant improvement as the number of attempts increases. With the compute volume of 128 attempts, it even reaches a level close to 80 points. This fully indicates that a single information-seeking action within WideSearch is not a particularly difficult task. Corresponding to real-world human scenarios, finding a piece of basic information is not an exceedingly difficult task for a person, provided enough time is spent. Quite the opposite, even with 128 attempts, the table-level score only reaches a level below 20 points. This strongly suggests that large-scale information retrieval on a fixed topic is an extremely difficult task. It requires not only comprehensive but also accurate search results. For a task with 5,000 atomic pieces of information, even if you find 4,999

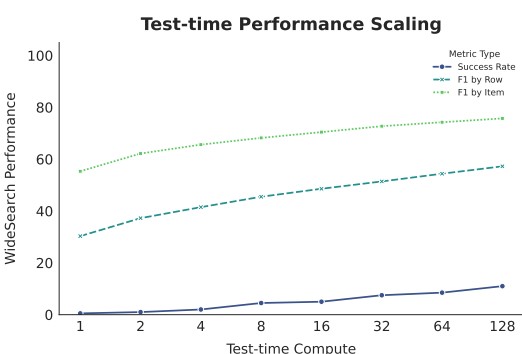

Figure 4: Test-time scaling experiments. We report the Pass@N for Success Rate, Max@N for Row-level, and Item-level F1 score.

correct pieces, the entire task is considered a failure if you retrieve one extra, one fewer, or one incorrect piece of information. Even for humans, completing this task requires multiple annotators to perform several rounds of repeated revisions, which is how the ground truth table for each question was annotated).

Given the characteristics of the WideSearch, we view the optimization of a multi-agent architecture as an important future research direction. Multiple agents can conduct parallel searches and perform mutual cross-validation, a process that aligns highly with the cognitive process of human annotation.

## 6 RELATED WORK

### 6.1 BENCHMARKS FOR SEARCH AGENTS

The evaluation of search agents has evolved from simple fact-retrieval (Joshi et al., 2017; Kwiatkowski et al., 2019) and multi-hop question answering (Yang et al., 2018; Ho et al., 2020; Trivedi et al., 2022) to more complex benchmarks that require deep, non-linear investigation of single topics or comprehensive report generation (Lei et al., 2025; Mialon et al., 2023; Chen et al., 2025; Wei et al., 2025; Zhou et al., 2025; Du et al., 2025). Existing benchmarks primarily test an agent's deep reasoning on query that seeks a hard-to-find information point. In contrast, our WideSearch benchmark is the first to evaluate the ability to gather broad information across multiple entities. We provided a more detailed comparison table for different search agent benchmarks in Table 5.

### 6.2 SEARCH AGENTS

The development of search agents includes powerful but closed-source proprietary systems (OpenAI, 2024; Google Gemini, 2025a; x.ai, 2025; Moonshot AI, 2025). In parallel, open-source efforts have focused on model-centric optimization (e.g., using Reinforcement Learning, simulated environments, and synthetic data) and workflow orchestration with multi-agent systems (Song et al., 2025a; Jin et al., 2025; Zheng et al., 2025; Sun et al., 2025; Song et al., 2025b; Huang et al., 2025; Wu et al., 2025a; Li et al., 2025a;b; Qiu et al., 2025). However, the performance of these agents on broad, multi-entity information-seeking tasks is underexplored. WideSearch is the first benchmark specifically designed to assess this capability.

## 7 CONCLUSION

We introduce WideSearch, a benchmark designed to evaluate LLM-Agents on "wide information seekin" tasks. Our evaluation of over 10 leading agent systems reveals significant shortcomings. Even the top performer achieved only a 7% success rate on table-level tasks, with most scoring near 0%. In-depth analysis shows the primary cause of failure is not finding individual data points (item-level F1 scores can be high), but the inability to gather all atomic information comprehensively and accurately. We identify core deficiencies in agentic capabilities: poor problem decomposition into sub-queries, a lack of reflection and iteration on failures, and improper use of retrieved evidence. In summary, current search agents are critically flawed for large-scale, high-fidelity information gathering. Our findings suggest a need for more sophisticated agent architectures, with multi-agent systems that simulate human collaboration—such as parallel search and cross-validation, representing a promising future direction.

## ETHICS STATEMENT

We confirm that this work adheres to the ICLR Code of Ethics. The WideSearch benchmark is derived from real-world user queries that were rigorously anonymized to protect user privacy. Each task is constructed around publicly verifiable and objective facts. All human annotators involved in the data curation and validation process were fairly compensated for their contributions. The authors are fully responsible for the content and integrity of this research.

## REPRODUCIBILITY STATEMENT

We are committed to ensuring the reproducibility of our research. To facilitate this, we provide a ZIP file in the supplementary materials that contains all necessary components to reproduce our findings. This file includes: (1) the complete and anonymized WideSearch benchmark dataset, comprising all 200 tasks and their ground-truth tables; and (2) the full source code for our experimental framework and automated evaluation pipeline. The experimental setup is described in detail in Section 3.1. For precise replication, the specific model API identifiers used in our experiments are listed in Appendix B. The exact system prompts and tool definitions for the agent frameworks are provided in Appendix G, and the prompts used within our evaluation pipeline, such as for the LLM-as-a-judge, are detailed in Appendix H.

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

## A   LLM USAGE DISCLOSURE

We used large language models to assist in the writing process, for purposes of checking grammar and improving language clarity. We did not use any large language models to generate research ideas, experimental code, or to analyze results. Authors of this paper are responsible for the content of the paper.

## B   MODELS AND API IDENTIFIERS

The models used in this benchmark and their corresponding API identifiers (which often include version or date information) are listed in Table 3. The "Benchmark Alias" is the shorthand name used to refer to the model within our paper.

It is worth noting that `Doubao-Seed-1.6 (Thinking)` and `Doubao-Seed-1.6 (Non-Thinking)` share the same API identifier but are configured with different generation parameters (with the "thinking" feature enabled and disabled, respectively). Similarly, `Claude Sonnet 4 (Thinking)` represent specific configurations with the "thinking" feature enabled.

Table 3: Correspondence between Benchmark Aliases and API Identifiers

| Benchmark Alias | API Identifier |
|---|---|
| Kimi K2 | kimi-k2-250711 |
| Doubao-Seed-1.6 (Thinking) | doubao-seed-1-6-250615 |
| Doubao-Seed-1.6 (Non-Thinking) | doubao-seed-1-6-250615 |
| DeepSeek-R1 | deepseek-r1-0528 |
| Claude Sonnet 4 (Thinking) | claude-sonnet-4-20250514 |
| OpenAI o3 | o3-2025-04-16 |
| Gemini 2.5 Pro | gemini-2.5-pro-preview-06-05 |
| GPT-4.1 | gpt-4.1-2025-04-14 |
| OpenAI o4-mini | o4-mini-2025-04-16 |
| GPT-5 | gpt-5-2025-08-07 |
| Qwen3-235B (Thinking) | qwen3-235b-a22b-thinking-2507 |

## C   TASK EXAMPLE

We provide a visually enhanced example of a task from our benchmark for better illustration. The task is separated into a styled **Task Prompt** box, a **Ground-Truth** box, and an **Evaluation Criteria** box.

---

**Task Prompt**

Could you list every single concert on Taylor Swift's tour from January 1, 2010, to May 1, 2025, including the specific date, the concert's English name, the country, the city, and the venue? Each show should be on its own line, in chronological order from earliest to latest. Please organize the results in one Markdown table with the following columns: Date, Concert's English Name, Host Country, Host City, Host Venue

Notes: Do not use date ranges for Date, list it in the format of "Day Month, Year", for example: 4th June, 2011

The output format is ```markdown\n{data_content}\n```.

---

**Ground-Truth**

| Date | Concert's English Name | Host Country | Host City | Host Venue |
|---|---|---|---|---|
| 4th February, 2010 | Fearless Tour | Australia | Brisbane | Brisbane Entertainment Centre |
| 6th February, 2010 | Fearless Tour | Australia | Sydney | Acer Arena |
| ⋮ | ⋮ | ⋮ | ⋮ | ⋮ |
| 7th December, 2024 | The Eras Tour | Canada | Vancouver | BC Place |
| 8th December, 2024 | The Eras Tour | Canada | Vancouver | BC Place |

*(Full table contains 533 entries and is truncated for clarity)*

**Evaluation Criteria**

**Unique Columns:** `["Date"]`

**Required Columns:** `["Date", "Concert's English Name", "Host Country", "Host City", "Host Venue"]`

**Evaluation Pipeline:**
- **Date**:
  - *Pre-process*: `["norm_str"]`
  - *Metric*: `["exact_match"]`
- **Concert's English Name**:
  - *Pre-process*: `["norm_str"]`
  - *Metric*: `["exact_match"]`
- **Host Country**:
  - *Pre-process*: `["norm_str"]`
  - *Metric*: `["exact_match"]`
- **Host City**:
  - *Pre-process*: `["norm_str"]`
  - *Metric*: `["llm_judge"]`
  - *Criterion*: It is sufficient if the semantics are approximately the same as the reference answer or if they point to the same entity. There is no need for a word-for-word correspondence.
- **Host Venue**:
  - *Pre-process*: `["norm_str"]`
  - *Metric*: `["llm_judge"]`
  - *Criterion*: It is sufficient if the semantics are approximately the same as the reference answer or if they point to the same entity. There is no need for a word-for-word correspondence.

# D  BENCHMARK COMPOSITION AND STATISTICS

The rigorous curation pipeline culminates in the final WideSearch benchmark, which comprises 200 high-quality tasks. For robust cross-lingual evaluation, these tasks are distributed equally between English and Chinese (100 tasks per language). Furthermore, to ensure broad applicability and mitigate domain-specific biases, the tasks are methodically balanced across 18 diverse topics, as detailed in Figure 5.

To quantitatively substantiate the complexity inherent in our benchmark, we conduct a detailed human annotation study with 30 participants. This evaluation is performed on a representative subset of 100 tasks, drawn equally from the Chinese and English pools (50 tasks each). Annotators are given ample time and instructed to complete each task independently to achieve the highest possible accuracy. However, we acknowledge that due to the numerous data points required for each complex task, even a diligent human annotator may commit inadvertent errors in a single session. To mitigate the impact of such potential errors and establish a robust performance ceiling, we implement a dual-annotation protocol. Each task is independently completed by two annotators, and we exclusively utilize the data from the annotator who achieved higher accuracy. This rigorous methodology ensures that our complexity metrics are grounded in high-quality, successful task completions.

Our analysis focuses on several key indicators. The first two metrics, derived from the human study, measure the procedural effort. As illustrated in Figure 6a, we report the distribution of human completion times. It directly reflects the significant cognitive and temporal investment demanded

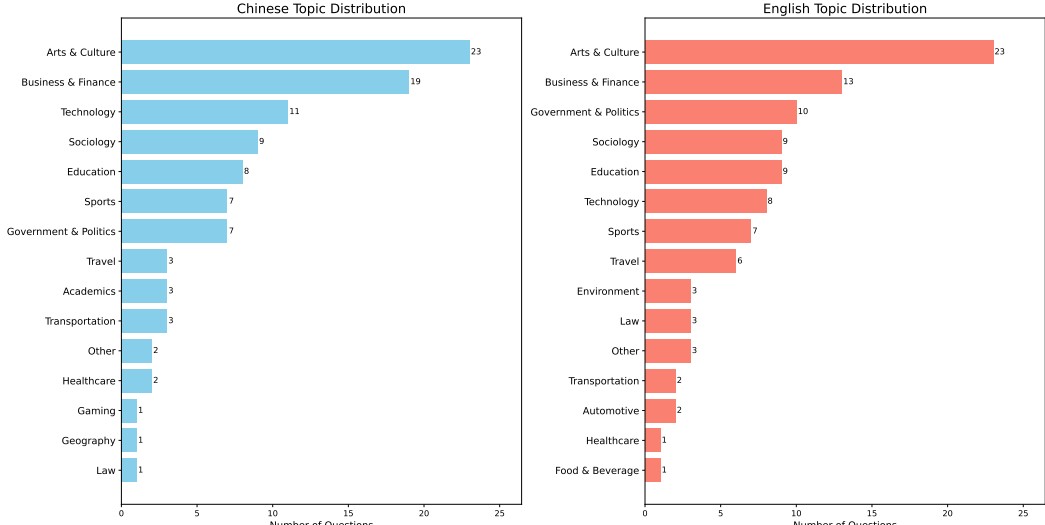

Figure 5: Distribution of the 18 distinct topics across the 200 tasks in the WideSearch benchmark, ensuring broad domain coverage.

by each task, with an overall average completion time of 2.33 hours. This is remarkably consistent across languages, with English tasks averaging 2.29 hours and Chinese tasks averaging 2.37 hours. This metric is comprehensive, encapsulating the entire workflow from initial query comprehension, through multi-step searching and information synthesis, to final result validation.

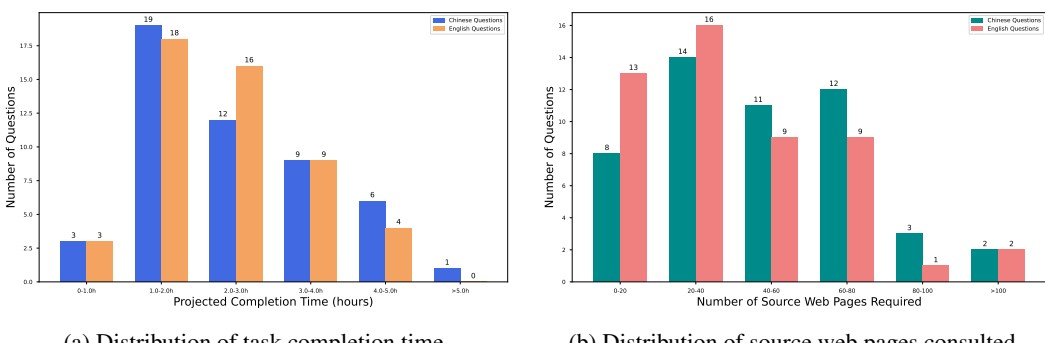

(a) Distribution of task completion time.

(b) Distribution of source web pages consulted.

Figure 6: Statistical distributions of key complexity metrics from our human annotation study. Both charts compare performance on Chinese and English tasks, showing (a) the time required for completion and (b) the breadth of research needed.

Furthermore, to quantify the procedural depth, Figure 6b shows the number of unique source web pages that annotators consulted. The breadth of research required is extensive; on average, annotators need to consult 44.10 unique web pages per task (48.74 for Chinese and 39.46 for English). Annotators are not limited in their choice of search tools. Crucially, this number represents not a theoretical minimum, but the actual breadth of research performed, including the cross-verification of facts across multiple sources to ensure accuracy. It therefore serves as a strong proxy for the non-trivial nature of the information-seeking process.

Finally, to characterize the informational scope across the *entire* benchmark, Table 4 presents the distribution of answer data volume. This metric reflects the amount of factual information that must be synthesized and structured to provide a complete and correct solution.

Table 4: Projected Distribution of Answer Data Volume. Data volume is defined as the number of discrete factual data points (e.g., rows multiplied by columns in a result table) required for a complete answer.

| Data Volume Range | Chinese Questions | English Questions |
|---|---|---|
| 0 - 100 | 0 | 2 |
| 100 - 1000 | 59 | 77 |
| 1000 - 2000 | 17 | 13 |
| 2000 - 3000 | 9 | 5 |
| 3000 - 4000 | 2 | 0 |
| 4000 - 5000 | 7 | 0 |
| >5000 | 6 | 3 |
| **Average Volume** | **2001.2** | **938.6** |

Table 5: Extended comparison with existing benchmarks, highlighting core task dimensions. The **Task Archetype**, **Output Format**, and **Core Model Challenge** columns are derived from our definitions in Table 2. Classifications for existing benchmarks are inferred based on their evaluation methods and task goals.

| Benchmark | Horizon | # of Tasks | Time-Varying | Task Archetype | Output Format | Evaluation | Core Model Challenge |
|---|---|---|---|---|---|---|---|
| WebWalkerQA (Wu et al., 2025b) | Short | 680 | × | DeepSearch | Single/few facts | Answer Match | Deep reasoning |
| GAIA (Mialon et al., 2023) | Medium | 466 | × | DeepSearch | Single/few facts | Answer Match | Deep reasoning |
| BrowseComp (Wei et al., 2025) | Long | 1,266 | × | DeepSearch | Single/few facts | Answer Match | Deep reasoning |
| BrowseComp-ZH (Zhou et al., 2025) | Long | 289 | × | DeepSearch | Single/few facts | Answer Match | Deep reasoning |
| DeepResearch Bench (Du et al., 2025) | Long | 100 | ✓ | DeepResearch | Long-form report | LLM-as-a-Judge | Information synthesis, logical organization |
| Deep Research Bench (Bosse et al., 2025) | Long | 89 | ✓ | DeepResearch | Few facts / Long-form report | LLM-as-a-Judge / Answer Match | Information synthesis, logical organization |
| **WideSearch (Ours)** | **Long** | **200** | **×** | **WideSearch** | **A structured table of entities** | **LLM-as-a-Judge / Answer Match** | **Exhaustiveness, and fidelity of Information** |

# E  DETAILED COMPARISON WITH PREVIOUS WORK

# F  DETAILED EXPERIMENTS

We present detailed experimental data in Table 6, including Success Rate, Row-level Precision, Row-level Recall, Row-level F1, Item-level Precision, Item-level Recall, and Item-level F1 on different subsets of WideSearch.

Table 6: Detailed experiments results on the WideSearch benchmark.

| Model / System | Success Rate | | Row Precision | | Row Recall | | Row F1 | | Item Precision | | Item Recall | | Item F1 | |
|---|---|---|---|---|---|---|---|---|---|---|---|---|---|---|
| | Avg@4 | Pass@4 | Avg@4 | Max@4 | Avg@4 | Max@4 | Avg@4 | Max@4 | Avg@4 | Max@4 | Avg@4 | Max@4 | Avg@4 | Max@4 |
| *Single Agent on WideSearch-zh* | | | | | | | | | | | | | | |
| GPT-5 | 4.75 | 9.00 | 50.92 | 66.25 | 38.45 | 49.26 | 40.79 | 51.89 | 68.54 | 81.54 | 51.09 | 61.63 | 54.64 | 65.18 |
| Qwen3-235B (Thinking) | 0.25 | 1.00 | 28.35 | 44.56 | 18.36 | 28.60 | 20.46 | 31.34 | 61.02 | 73.93 | 37.98 | 48.30 | 42.51 | 52.84 |
| Claude Sonnet 4 (Thinking) | 0.25 | 1.00 | 35.91 | 48.63 | 29.08 | 38.11 | 30.19 | 39.73 | 65.05 | 75.99 | 50.84 | 60.51 | 53.76 | 63.19 |
| Gemini 2.5 Pro | 1.00 | 3.00 | 33.98 | 48.04 | 25.68 | 35.13 | 26.95 | 36.96 | 60.49 | 73.46 | 42.35 | 54.45 | 45.57 | 57.26 |
| OpenAI o3-high | 2.00 | 5.00 | 37.10 | 50.97 | 27.02 | 36.23 | 29.30 | 39.31 | 57.31 | 71.48 | 41.45 | 50.19 | 45.19 | 54.46 |
| K2 | 0.25 | 1.00 | 35.53 | 49.23 | 25.89 | 36.68 | 27.79 | 39.03 | 63.15 | 75.05 | 45.05 | 56.07 | 48.81 | 59.64 |
| DeepSeek-R1-0528 | 0.25 | 1.00 | 28.12 | 44.39 | 16.33 | 25.40 | 18.44 | 28.35 | 54.26 | 71.91 | 29.55 | 43.35 | 33.95 | 47.83 |
| Doubao-1.6 | 1.75 | 4.00 | 37.20 | 54.25 | 27.13 | 39.62 | 29.25 | 42.08 | 58.29 | 77.74 | 40.00 | 54.87 | 43.72 | 58.84 |
| Doubao-1.6-non-thinking | 0.50 | 2.00 | 34.48 | 52.17 | 23.67 | 35.00 | 25.56 | 37.41 | 59.97 | 76.31 | 39.14 | 52.01 | 42.87 | 55.79 |
| *Single Agent on WideSearch-en* | | | | | | | | | | | | | | |
| GPT-5 | 9.00 | 18.00 | 38.29 | 57.64 | 34.10 | 51.14 | 34.93 | 52.56 | 59.05 | 78.08 | 51.82 | 69.68 | 53.37 | 71.27 |
| Qwen3-235B (Thinking) | 0.75 | 2.00 | 27.47 | 42.40 | 18.89 | 27.80 | 20.65 | 30.21 | 62.81 | 74.82 | 43.36 | 54.74 | 47.23 | 58.21 |
| Claude Sonnet 4 (Thinking) | 4.25 | 9.00 | 37.01 | 50.55 | 31.97 | 42.69 | 33.18 | 44.08 | 69.83 | 79.13 | 59.67 | 68.32 | 62.02 | 70.27 |
| Gemini 2.5 Pro | 2.00 | 7.00 | 37.66 | 52.62 | 31.37 | 43.74 | 33.05 | 45.82 | 65.57 | 81.19 | 53.67 | 67.38 | 56.38 | 69.97 |
| OpenAI o3-high | 7.00 | 13.00 | 43.04 | 55.30 | 37.48 | 47.26 | 38.70 | 48.84 | 68.28 | 79.30 | 58.07 | 68.31 | 60.03 | 70.08 |
| K2 | 2.00 | 6.00 | 34.71 | 48.42 | 30.36 | 42.34 | 31.54 | 43.68 | 67.08 | 79.51 | 57.63 | 68.42 | 59.91 | 70.52 |
| DeepSeek-R1-0528 | 0.50 | 2.00 | 28.05 | 44.39 | 21.10 | 32.53 | 22.88 | 35.03 | 60.35 | 76.06 | 44.89 | 59.16 | 48.58 | 62.36 |
| Doubao-1.6 | 3.50 | 6.00 | 35.75 | 54.23 | 29.02 | 43.99 | 30.56 | 46.16 | 63.25 | 81.62 | 49.71 | 65.68 | 52.82 | 68.88 |
| Doubao-1.6-non-thinking | 1.50 | 5.00 | 35.84 | 51.98 | 26.90 | 39.88 | 28.86 | 42.31 | 69.29 | 82.92 | 51.23 | 64.86 | 55.06 | 68.17 |
| *Single Agent on WideSearch-all* | | | | | | | | | | | | | | |
| GPT-5 | 6.88 | 13.50 | 44.60 | 61.95 | 36.28 | 50.20 | 37.86 | 52.22 | 63.79 | 79.81 | 51.46 | 65.65 | 54.01 | 68.23 |
| Qwen3-235B (Thinking) | 0.50 | 1.50 | 27.91 | 43.48 | 18.63 | 28.20 | 20.56 | 30.78 | 61.91 | 74.37 | 40.67 | 51.52 | 44.87 | 55.52 |
| Claude Sonnet 4 (Thinking) | 2.25 | 5.00 | 36.46 | 49.59 | 30.52 | 40.40 | 31.69 | 41.90 | 67.44 | 77.56 | 55.26 | 64.41 | 57.89 | 66.73 |
| Gemini 2.5 Pro | 1.50 | 5.00 | 35.82 | 50.33 | 28.52 | 39.44 | 30.00 | 41.39 | 63.03 | 77.32 | 48.01 | 60.92 | 50.98 | 63.62 |
| OpenAI o3-high | 4.50 | 9.00 | 40.07 | 53.13 | 32.25 | 41.74 | 34.00 | 44.07 | 62.79 | 75.39 | 49.76 | 59.25 | 52.61 | 62.27 |
| K2 | 1.12 | 3.50 | 35.12 | 48.82 | 28.12 | 39.51 | 29.67 | 41.35 | 65.12 | 77.28 | 51.34 | 62.24 | 54.36 | 65.08 |
| DeepSeek-R1-0528 | 0.37 | 1.50 | 28.09 | 44.39 | 18.72 | 28.97 | 20.66 | 31.69 | 57.31 | 73.98 | 37.22 | 51.26 | 41.26 | 55.09 |
| Doubao-1.6 | 2.63 | 5.00 | 36.47 | 54.24 | 28.07 | 41.80 | 29.90 | 44.12 | 60.77 | 79.68 | 44.86 | 60.28 | 48.27 | 63.86 |
| Doubao-1.6-non-thinking | 1.00 | 3.50 | 35.16 | 52.07 | 25.29 | 37.44 | 27.21 | 39.86 | 64.63 | 79.62 | 45.18 | 58.44 | 48.97 | 61.98 |
| *Multi-Agent Framework on WideSearch-zh* | | | | | | | | | | | | | | |
| GPT-5 | 4.00 | 7.00 | 48.90 | 65.20 | 39.65 | 50.04 | 40.86 | 51.81 | 65.36 | 79.21 | 51.88 | 63.14 | 54.04 | 65.10 |
| Qwen3-235B (Thinking) | 0.00 | 0.00 | 35.53 | 51.84 | 24.68 | 34.66 | 26.94 | 37.65 | 65.80 | 77.87 | 44.48 | 55.16 | 48.77 | 59.16 |
| Claude Sonnet 4 (Thinking) | 2.75 | 6.00 | 41.27 | 58.42 | 35.88 | 50.25 | 36.85 | 51.46 | 64.73 | 79.80 | 55.14 | 67.62 | 57.13 | 69.53 |
| Gemini 2.5 Pro | 1.00 | 4.00 | 36.46 | 50.62 | 29.43 | 40.41 | 30.93 | 42.21 | 62.82 | 73.96 | 48.59 | 58.16 | 51.79 | 60.87 |
| OpenAI o3-high | 2.75 | 6.00 | 41.11 | 59.03 | 32.17 | 45.29 | 33.83 | 47.85 | 62.64 | 79.05 | 47.16 | 59.24 | 50.35 | 63.06 |
| K2 | 1.25 | 3.00 | 40.37 | 56.14 | 33.21 | 46.30 | 34.74 | 48.01 | 67.39 | 78.04 | 53.74 | 64.72 | 56.86 | 66.75 |
| DeepSeek-R1-0528 | 0.50 | 2.00 | 28.67 | 47.38 | 19.37 | 32.63 | 21.17 | 35.08 | 53.66 | 72.11 | 33.79 | 49.38 | 37.66 | 53.15 |
| Doubao-1.6 | 2.25 | 6.00 | 39.22 | 57.11 | 30.84 | 45.21 | 32.83 | 47.49 | 60.34 | 78.62 | 45.51 | 61.32 | 48.79 | 64.43 |
| Doubao-1.6-non-thinking | 0.50 | 1.00 | 36.78 | 56.15 | 24.76 | 37.75 | 26.93 | 40.30 | 64.13 | 79.27 | 42.54 | 56.06 | 46.52 | 59.63 |
| *Multi-Agent Framework on WideSearch-en* | | | | | | | | | | | | | | |
| GPT-5 | 10.75 | 17.00 | 47.94 | 62.51 | 43.37 | 55.16 | 44.13 | 56.52 | 71.02 | 83.56 | 63.96 | 74.07 | 65.25 | 75.86 |
| Qwen3-235B (Thinking) | 1.25 | 4.00 | 30.55 | 45.65 | 22.95 | 35.20 | 24.57 | 37.21 | 66.81 | 78.19 | 51.64 | 63.01 | 54.94 | 66.09 |
| Claude Sonnet 4 (Thinking) | 4.50 | 7.00 | 43.35 | 57.86 | 39.24 | 51.76 | 40.13 | 52.91 | 72.48 | 82.49 | 65.83 | 75.80 | 67.21 | 76.72 |
| Gemini 2.5 Pro | 3.00 | 9.00 | 39.35 | 52.54 | 34.87 | 45.68 | 36.00 | 47.06 | 70.41 | 82.19 | 60.96 | 70.19 | 63.06 | 71.75 |
| OpenAI o3-high | 7.50 | 13.00 | 46.30 | 59.86 | 40.11 | 51.16 | 41.78 | 53.20 | 71.80 | 82.84 | 61.87 | 73.16 | 64.27 | 74.80 |
| K2 | 4.75 | 10.00 | 40.33 | 54.90 | 36.90 | 50.43 | 37.71 | 51.20 | 70.91 | 80.53 | 63.83 | 73.79 | 65.44 | 74.68 |
| DeepSeek-R1-0528 | 1.00 | 4.00 | 29.28 | 45.93 | 23.17 | 36.17 | 24.57 | 38.10 | 60.58 | 77.93 | 48.19 | 65.01 | 50.91 | 67.54 |
| Doubao-1.6 | 2.75 | 5.00 | 39.49 | 57.64 | 33.89 | 48.73 | 35.14 | 50.38 | 68.65 | 83.90 | 58.18 | 73.38 | 60.39 | 74.87 |
| Doubao-1.6-non-thinking | 3.75 | 8.00 | 38.51 | 52.54 | 31.05 | 43.42 | 32.38 | 44.99 | 70.74 | 81.92 | 56.27 | 68.24 | 58.97 | 70.62 |
| *Multi-Agent Framework on WideSearch-all* | | | | | | | | | | | | | | |
| GPT-5 | 7.37 | 12.00 | 48.42 | 63.85 | 41.51 | 52.60 | 42.50 | 54.16 | 68.19 | 81.39 | 57.92 | 68.61 | 59.65 | 70.48 |
| Qwen3-235B (Thinking) | 0.63 | 2.00 | 33.04 | 48.75 | 23.82 | 34.93 | 25.76 | 37.43 | 66.31 | 78.03 | 48.06 | 59.09 | 51.86 | 62.62 |
| Claude Sonnet 4 (Thinking) | 3.62 | 6.50 | 42.31 | 58.14 | 37.56 | 51.01 | 38.49 | 52.19 | 68.60 | 81.15 | 60.48 | 71.71 | 62.17 | 73.13 |
| Gemini 2.5 Pro | 2.00 | 6.50 | 37.90 | 51.58 | 32.15 | 43.04 | 33.47 | 44.64 | 66.62 | 77.07 | 54.77 | 64.18 | 57.42 | 66.31 |
| OpenAI o3-high | 5.12 | 9.50 | 43.70 | 59.44 | 36.14 | 48.23 | 37.80 | 50.52 | 67.22 | 80.95 | 54.51 | 66.20 | 57.31 | 68.93 |
| K2 | 3.00 | 6.50 | 40.35 | 55.52 | 35.06 | 48.36 | 36.22 | 49.60 | 69.15 | 79.28 | 58.78 | 69.26 | 61.15 | 70.72 |
| DeepSeek-R1-0528 | 0.75 | 3.00 | 28.97 | 46.66 | 21.27 | 34.40 | 22.87 | 36.59 | 57.12 | 75.02 | 40.99 | 57.20 | 44.28 | 60.34 |
| Doubao-1.6 | 2.50 | 5.50 | 39.35 | 57.38 | 32.36 | 46.97 | 33.98 | 48.93 | 64.49 | 81.26 | 51.85 | 67.35 | 54.59 | 69.65 |
| Doubao-1.6-non-thinking | 2.12 | 4.50 | 37.64 | 54.34 | 27.91 | 40.58 | 29.65 | 42.65 | 67.43 | 80.59 | 49.41 | 62.15 | 52.75 | 65.13 |
| *End-to-End System on WideSearch-zh* | | | | | | | | | | | | | | |
| Claude | 0.00 | 0.00 | 27.91 | 38.94 | 19.29 | 27.30 | 20.84 | 28.92 | 57.15 | 67.96 | 40.25 | 49.54 | 43.51 | 52.14 |
| Gemini | 1.50 | 4.00 | 37.25 | 47.95 | 31.28 | 39.12 | 32.32 | 40.52 | 64.06 | 73.90 | 50.07 | 58.17 | 52.92 | 60.44 |
| OpenAI o3 | 3.00 | 5.00 | 35.41 | 52.21 | 25.91 | 36.29 | 27.40 | 38.34 | 62.29 | 75.94 | 42.73 | 53.53 | 46.03 | 56.51 |
| *End-to-End System on WideSearch-en* | | | | | | | | | | | | | | |
| Claude | 5.00 | 10.00 | 30.49 | 43.85 | 26.48 | 36.78 | 27.39 | 38.07 | 60.23 | 74.22 | 51.16 | 62.83 | 53.29 | 64.81 |
| Gemini | 7.00 | 12.00 | 44.43 | 54.58 | 39.75 | 49.13 | 40.95 | 50.29 | 71.39 | 80.43 | 63.30 | 72.19 | 65.18 | 73.90 |
| OpenAI o3 | 3.00 | 6.00 | 24.38 | 40.65 | 19.38 | 31.88 | 20.42 | 33.72 | 55.32 | 67.22 | 42.45 | 53.73 | 45.02 | 56.47 |
| *End-to-End System on WideSearch-all* | | | | | | | | | | | | | | |
| Claude | 2.50 | 5.00 | 29.20 | 41.40 | 22.89 | 32.04 | 24.11 | 33.49 | 58.69 | 71.09 | 45.70 | 56.18 | 48.40 | 58.47 |
| Gemini | 4.25 | 8.00 | 40.84 | 51.27 | 35.51 | 44.13 | 36.63 | 45.41 | 67.72 | 77.16 | 56.69 | 65.18 | 59.05 | 67.17 |
| OpenAI o3 | 3.00 | 5.50 | 29.90 | 46.43 | 22.64 | 34.08 | 23.91 | 36.03 | 58.81 | 71.58 | 42.59 | 53.63 | 45.52 | 56.49 |

# G   AGENT FRAMEWORK DETAILS

We provide the system prompt of the Single-Agent framework, the Multi-Agent framework, and the tools prompt for the agents as follows. The "Create Sub-Agent" is only provided for the Multi-Agent framework.

> SINGLE AGENT PROMPT
>
> # Role
> You are an expert in online search. Your task is gathering relevant information using advanced online search tools based on the user's query, and providing accurate answers according to the search results.

# Task Description
Upon receiving the user's query, you must thoroughly analyze and understand the user's requirements. In order to effectively address the user's query, you should make the best use of the provided tools to acquire comprehensive and reliable information and data. Below are the principles you should adhere to while performing this task:
- Fully understand the user's needs: Analyze the user's query, if necessary, break it down into smaller components to ensure a clear understanding of the user's primary intent.
- Flexibly use tools: After fully comprehending the user's needs, employ the provided tools to retrieve the necessary information.If the information retrieved previously is deemed incomplete or inaccurate and insufficient to answer the user's query, reassess what additional information is required and invoke the tool again until all necessary data is obtained.

MULTI AGENT PROMPT

# Role
You are a professional and meticulous expert in information collection and collation. You can fully understand users' needs, skillfully use search tools, and complete the tasks assigned by users with the highest efficiency.
# Task Description
After receiving users' questions, you need to fully understand their needs and think about and plan how to complete the tasks assigned by users efficiently and quickly. To help you complete tasks better and faster, I have provided you with three tools:
1. Search tool: You can use the search engine to retrieve information;
2. Link reading tool: a link reading tool that can open links (which can be web pages, PDFs, etc.) and summarize all relevant information on the page according to the requirement description.
3. Sub Agent: The Sub Agent can complete various types of tasks according to the prompt you input. The Sub Agent itself can also use the search tool and the link reading tool. You can split your tasks into multiple sub-tasks according to your own needs, and then create one or more Agents to help you complete these sub-tasks in parallel.

TOOL DESCRIPTION

SEARCH TOOL

Description:
This is a search tool. Enter search queries, and it will return a list of web pages along with their corresponding summary information. Search queries should be concise and clear; complex questions should be broken down into multiple steps and searched step by step. If no useful pages are found, you can adjust the question description (such as reducing qualifiers or changing the search approach) and search again. The quality of search results is related to the language: for Chinese resources, you can try entering Chinese queries; for non-Chinese resources, you can try using English or the corresponding language.
Parameters: query, count, summary_type, use_english

TEXT BROWSER VIEW

Description:
This is a link reading tool that can open links (which can be web pages, PDFs, etc.) and summarize all relevant information on the page according to the requirement description. This tool can be called to obtain information for all valuable links. Valuable links include but are not limited to the following types: 1. URLs explicitly provided in the task; 2. URLs with relevant summaries provided in search results; 3. URLs contained in the content returned by previous calls to TextBrowserView that are judged to potentially contain useful information. Please try to avoid constructing links out of thin air by yourself.
Parameters: url, description

CREATE SUB-AGENT

Description:
Creates sub-agents that can perform specific tasks based on the input prompt.
Parameters: sub_agents: $[(\text{prompt}_0, \text{index}_0), ..., (\text{prompt}_N, \text{index}_N)]$

## H   EVALUATION DETAILS

Sometimes, the column names in the markdown table generated by the model are completely semantically consistent with those in the ground truth, but the surface strings are not entirely the same. We believe that such cases should be considered as correct predictions by the model, so we use the following mapping prompt to map column names with the same semantics to the same string. Similarly, the inner join on the primary key between the two tables also relies on string matching. We also use the mapping prompt to map entities with the same semantics under the corresponding primary keys in the two tables to the same string. We present an example of the mapping below.

MAPPING PROMPT

Your task is to align two vocabularies. The inputs are the vocabulary to be aligned and the reference vocabulary, respectively. Note that you need to perform semantic alignment (not positional alignment). If two strings are the same, they must correspond to each other. These two strings are supposed to represent the same entity, with differences only in the expression forms and formats.
The vocabulary to be aligned is as follows:
{response}
The reference vocabulary is as follows:
{reference}
The alignment rules are as follows:
List the values in the vocabulary to be aligned one by one. If there is a value in the reference vocabulary that has the same meaning as this value, 'transform' should be represented as the value from the reference vocabulary; otherwise, 'transform' should be represented as the original value from the vocabulary to be aligned.
Note that 'origin' must be taken from the vocabulary to be aligned, keeping the original format, and 'transform' must be taken from the reference vocabulary. For example: Some words in the vocabulary to be aligned might be the words in the reference vocabulary with Markdown formatting added, keep the to be aligned format in 'origin' and the reference format in 'transform'.
For the 'origin', first find the 'transform' that is the closest in meaning, and then judge whether they correspond to each other. Those entities not correspond to each other cannot output.
Please output the alignment results in the following format:
```json
{
"origin_str1": "transform_str1",
"origin_str2": "transform_str2"
}
```

MAPPING OUTPUT EXAMPLE

```json
"gemini-2.5-pro (thinking)": "Gemini 2.5 Pro",
"gemini-2.5-flash (thinking)": "Gemini 2.5 Flash-thinking",
"gemini-2.5-flash (non-thinking)": "Gemini 2.5 Flash-non thinking",
"gemini-2.5-flash-lite (thinking)": "Gemini 2.5 Flash-Lite-thinking",
"gemini-2.5-flash-lite (non-thinking)": "Gemini 2.5 Flash-Lite-non thinking",
"claude-3-7-sonnet": "Claude 3.7 sonnet",
"claude-opus-4": "Claude Opus 4",
```

"claude-sonnet-4": "Claude Sonnet 4",
"o3 (high)": "O3",
"o3-mini (high)": "O3 mini",
"o4-mini (high)": "o4-mini",
"doubao-1.5-pro-thinking": "Seed-Thinking v1.5",
"doubao-1.6": "Seed-Thinking v1.6",
"deepseek-v3": "DeepSeek-V3",
"deepseek-r1-0528": "DeepSeek-R1-0528"
"'

For non-primary key columns, some columns cannot be evaluated using rule-based methods and must be evaluated using the llm-as-judge approach. During the data annotation process, we annotated the LLM-as-judge criteria for these columns. To improve the efficiency of the judge, we perform evaluations column by column. The specific llm-as-judge prompt and evaluation examples are as follows:

---

**LLM-AS-JUDGE PROMPT**

You are an expert in grading answers. Your task is to score the responses to a certain question. Below, you will be provided with a set of standard answers, a set of responses to be graded, and specific grading criteria.

Each answer and each response has an index. Please score each pair of answers and responses in this set according to the following methods:

1. The scoring range is from 0 to 1. A score of 1 indicates a completely correct answer. For deduction items, please refer to the specific grading criteria section.

2. After reading the standard answers, responses to be graded, and grading criteria, please first analyze and judge them item by item according to the grading criteria.

3. The score can only be an integer of 0 or 1.

4. After the analysis and judgment, please provide the final scoring results. Each pair should have a score. Output in Markdown JSON format, as shown below:

```json
{
"idx_xxx": score,
"idx_yyy": score,

...
}
```

====== criterion-start ======
{criterion}
====== criterion-end ======
====== response-start ======
{response}
====== response-end ======
Now start scoring. Please make sure to analyze each item step by step before providing the final scoring results.

---

**LLM-AS-JUDGE OUTPUT EXAMPLE**

Let's analyze each answer and response according to the criterion: semantic equivalence is sufficient; answers do not have to correspond word-for-word.
- idx_0: response = '1mtokens', target = '1m'
- "1mtokens" clearly refers to "1 million tokens" which semantically matches "1m" (1 million).
- Score: 1
- idx_1: response = '1mtokens', target = '1m'
- Same as above.
- Score: 1

- idx_2: response = '1mtokens', target = '1m'
- Same as above.
- Score: 1
- idx_3: response = '128ktokens', target = '200k'
- ""128ktokens"" (128,000 tokens) does not semantically match ""200k"" (200,000 tokens).
- Score: 0
- idx_4: response = '200ktokens', target = '200k'
- ""200ktokens"" corresponds directly to ""200k"".
- Score: 1
- idx_5: response = '128ktokens', target = '200k'
- ""128ktokens"" does not correspond to ""200k"".
- Score: 0
- idx_6: response = 'nan', target = '200k'
- ""nan"" means there is no answer, which does not match ""200k"".
- Score: 0
- idx_7: response = '32ktokens', target = '32k'
- ""32ktokens"" refers to ""32,000 tokens"" and matches ""32k"".
- Score: 1
- idx_8: response = 'nan', target = '256k'
- ""nan"" is no answer, does not match ""256k"".
- Score: 0
- idx_9: response = '128ktokens', target = '64k'
- ""128ktokens"" does not correspond to ""64k"".
- Score: 0
- idx_10: response = '128ktokens', target = '64k'
- Same as above.
- Score: 0
- idx_11: response = '128ktokens', target = '64k'
- Same as above.
- Score: 0
Here is the final scoring according to the criteria:
```json
{
""idx_0"": 1, ""idx_1"": 1, ""idx_2"": 1, ""idx_3"": 0, ""idx_4"": 1, ""idx_5"": 0,
""idx_6"": 0, ""idx_7"": 1, ""idx_8"": 0, ""idx_9"": 0, ""idx_10"": 0, ""idx_11"": 0
}
```

## I   ERROR ANALYSIS AND EXAMPLES

This section provides detailed examples for each of the failure modes identified in Section 4. We use the trajectories of Gemini-2.5-pro in single-agent mode as examples. Each case includes the user's task, the agent's actions (e.g., search queries and retrieved evidence), and an analysis of the specific error.

**Case 1: Incomplete Query Decomposition** Figure I.1 illustrates a case of incomplete query decomposition. The agent was tasked with compiling a comprehensive table on the top five universities across five subject areas from the QS 2025 rankings. The required information included not only the rankings but also logistical details such as university websites, application deadlines, and fees. The agent successfully decomposed the initial part of the task, generating specific queries to identify the top universities in each subject area (e.g., "Top 5 universities in Arts and Humanities QS World University Rankings by Subject 2025"). However, it failed to generate the necessary subsequent queries to gather the additional required data points for each university. This omission demonstrates a failure in comprehensive task planning, as the agent did not create a complete set of subtasks needed to fulfill all aspects of the user's request, resulting in a final output with significant information gaps.

**Case 2: Lack of Reflection and Iterative Refinement** Figure I.2 provides an example of the agent's failure to reflect and refine its strategy. The task was to find the annual GDP growth rates for all sub-Saharan African countries from 2022 to 2024, using data from the World Bank. The

---

**Error: Incomplete Query Decomposition**

**Task:** Please compile a Markdown table summarizing the top five universities in five subject areas according to the QS 2025 rankings. The table should include the following columns: subject, university, QS subject ranking 2025, QS overall ranking 2025, THE overall ranking 2025, university website, regular application deadline, and application fee.

**Search Queries:**

"QS World University Rankings by Subject 2025 five broad subjects"

"Top 5 universities in Arts and Humanities QS World University Rankings by Subject 2025",

"Top 5 universities in Engineering and Technology QS World University Rankings by Subject 2025",

"Top 5 universities in Life Sciences and Medicine QS World University Rankings by Subject 2025",

"Top 5 universities in Natural Sciences QS World University Rankings by Subject 2025",

"Top 5 universities in Social Sciences and Management QS World University Rankings by Subject 2025"

**Analysis:** This query decomposition is incomplete as it only identifies the top universities based on the QS subject rankings. It omits the necessary subsequent queries to gather the other required data points, such as the overall rankings, university websites, application deadlines, and fees for each institution.

---

Figure I.1: An illustration of **Incomplete Query Decomposition** error. The agent identifies the universities but fails to generate subsequent queries for other required details like application deadlines and fees. Please note that the task description in the figure has been simplified for visualization purposes.

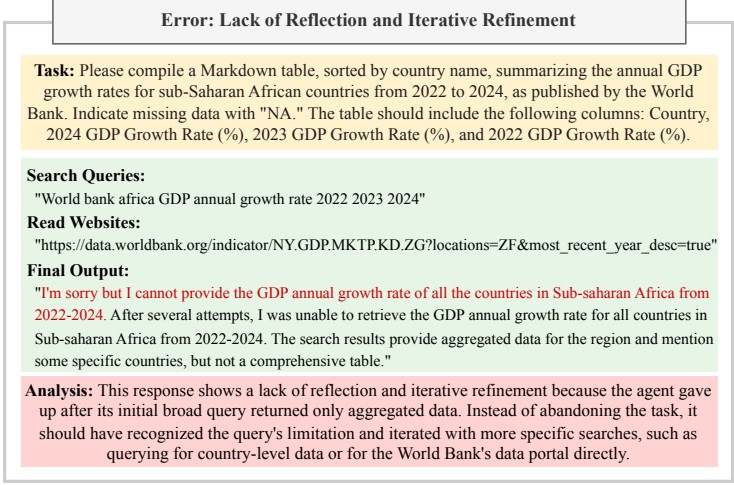

**Error: Lack of Reflection and Iterative Refinement**

**Task:** Please compile a Markdown table, sorted by country name, summarizing the annual GDP growth rates for sub-Saharan African countries from 2022 to 2024, as published by the World Bank. Indicate missing data with "NA." The table should include the following columns: Country, 2024 GDP Growth Rate (%), 2023 GDP Growth Rate (%), and 2022 GDP Growth Rate (%).

**Search Queries:**
"World bank africa GDP annual growth rate 2022 2023 2024"
**Read Websites:**
"https://data.worldbank.org/indicator/NY.GDP.MKTP.KD.ZG?locations=ZF&most_recent_year_desc=true"
**Final Output:**
"I'm sorry but I cannot provide the GDP annual growth rate of all the countries in Sub-saharan Africa from 2022-2024. After several attempts, I was unable to retrieve the GDP annual growth rate for all countries in Sub-saharan Africa from 2022-2024. The search results provide aggregated data for the region and mention some specific countries, but not a comprehensive table."

**Analysis:** This response shows a lack of reflection and iterative refinement because the agent gave up after its initial broad query returned only aggregated data. Instead of abandoning the task, it should have recognized the query's limitation and iterated with more specific searches, such as querying for country-level data or for the World Bank's data portal directly.

Figure I.2: An example of the **Lack of Reflection and Iterative Refinement** error. After an initial broad query returns aggregated data, the agent gives up instead of refining its search to find country-specific information. Please note that the task description in the figure has been simplified for visualization purposes.

agent initiated the search with a very broad query: "World bank africa GDP annual growth rate 2022 2023 2024". The search results provided only aggregated data for the entire region, not the country-specific data required. Instead of recognizing this limitation and adapting its approach—for instance, by iterating through individual countries or searching for the specific World Bank data portal—the agent abandoned the task. It concluded that it was unable to retrieve the information, demonstrating a lack of a critical reflective mechanism to diagnose search failures and dynamically adjust its plan.

---

**Error: Failure in Evidence Utilization**

**Task:** Please compile a Markdown table, sorted by country, summarizing the minimum GPA requirements for Master of Civil Engineering programs at Ivy League and Group of Eight universities. Use 'no minimum GPA specified' for unstated requirements. The table should include the following columns: Country, University, Alliance, and Minimum GPA Requirement.

**Search Queries:**
...
"Harvard University Master's in Civil Engineering GPA requirement"

**Retrieved Information:**
"... Title: Civil Engineering Program — Admission Requirements(cive.uh.edu)\nAbstract: a grade point average (gpa) of at least 3.0/4.0 (b-average grade) on the last 60 hours attempted.\nURL:https://www.cive.uh.edu/admissions/civil-engineering\n ..."

**Final Output:**
| USA | Harvard University | Ivy League | 3.0/4.0 |

**Analysis:** This case exemplifies a failure in source validation. The agent correctly extracts the GPA data point but fails to recognize that the evidence comes from an irrelevant source (the University of Houston), thus critically misattributing the information to Harvard University.

---

Figure I.3: An illustration of the **Failure in Evidence Utilization** error. The agent correctly extracts a GPA requirement but fails to validate its source, misattributing information from the University of Houston to Harvard University. Please note that the task description in the figure has been simplified for visualization purposes.

**Case 3: Failure in Evidence Utilization** Figure I.3 illustrates a critical failure in evidence utilization, specifically in the area of source validation. The agent was asked to find the minimum GPA requirement for a Master of Civil Engineering at Harvard University. It issued a correct query and retrieved a snippet of text stating a GPA requirement of "3.0/4.0". However, the source of this information was cive.uh.edu, the website for the University of Houston, not Harvard. The agent failed to validate the context and relevance of the retrieved evidence, incorrectly attributing the GPA requirement from the University of Houston to Harvard University in its final table. This error highlights a fundamental gap between information retrieval and faithful generation, where the agent does not properly ground its answer in correctly attributed evidence.

**Case 4: Knowledge Hallucination and Factual Inconsistency** Figure I.4 presents a clear case of knowledge hallucination. The task was to find the entrance fee for U.S. National Parks for June 2025. The agent's search for this future-dated information correctly returned no results. However, when generating the final table for Everglades National Park, instead of indicating that the information was unavailable (e.g., with "NA"), the agent fabricated a specific fee of "$15". This response was not based on any retrieved evidence but was instead generated from the model's internal parameters, resulting in a factual inconsistency. This case underscores the challenge of ensuring that LLMs remain strictly grounded in external sources, especially when those sources do not contain the requested information.

Figure I.4: An example of **Knowledge Hallucination and Factual Inconsistency** error. When a search for a future entrance fee returns no results, the agent invents an incorrect value ($15) instead of stating the information is unavailable. Please note that the task description in the figure has been simplified for visualization purposes.

Table 7: Basic failure modes distribution.

| Model | Tool Call Error | Context Length Exceeded | Output Format Error | Response Refusal |
|---|---|---|---|---|
| **Single-Agent Mode Error Distribution** | | | | |
| Deepseek-R1 | 4.12% | 0.12% | 0.50% | 3.00% |
| Doubao-Seed-1.6 (Thinking) | 3.62% | 12.62% | 0.25% | 1.62% |
| Doubao-Seed-1.6 (Non-Thinking) | 4.00% | 3.88% | 0.25% | 1.12% |
| Claude Sonnet 4 (Thinking) | 0.62% | 1.25% | 0.00% | 1.50% |
| Gemini 2.5 Pro | 3.00% | 0.00% | 0.38% | 7.25% |
| GPT-5 | 13.12% | 6.75% | 0.00% | 10.75% |
| Kimi K2 | 0.50% | 0.50% | 0.38% | 2.38% |
| OpenAI o3 | 1.88% | 0.38% | 0.00% | 16.00% |
| Qwen3-235B (Thinking) | 1.50% | 0.50% | 2.00% | 0.25% |
| **Multi-Agent Mode Error Distribution** | | | | |
| Deepseek-R1 | 22.75% | 0.00% | 0.00% | 4.00% |
| Doubao-Seed-1.6 (Thinking) | 29.88% | 4.25% | 0.12% | 1.38% |
| Doubao-Seed-1.6 (Non-Thinking) | 9.88% | 0.75% | 0.25% | 1.12% |
| Claude Sonnet 4 (Thinking) | 9.12% | 0.25% | 0.00% | 0.88% |
| Gemini 2.5 Pro | 6.75% | 0.00% | 0.00% | 3.38% |
| GPT-5 | 23.75% | 1.38% | 0.00% | 10.62% |
| Kimi K2 | 1.62% | 0.00% | 0.38% | 0.38% |
| OpenAI o3 | 4.38% | 0.62% | 0.00% | 10.12% |
| Qwen3-235B (Thinking) | 6.00% | 0.50% | 0.75% | 0.12% |

## J    DOMAIN-SPECIFIC PERFORMANCE ANALYSIS

To move beyond aggregate performance metrics, we conduct a granular analysis to understand how different models and frameworks perform across a variety of specific domains. This approach allows us to identify the strengths and weaknesses of each model and to assess the impact of the multi-agent framework on tasks requiring topic-specific knowledge. The evaluation is performed on both English and Chinese datasets to capture any language-dependent variations in performance.

Figure J.1 presents a detailed heatmap of the row-level F1 scores from this analysis. The figure is structured to facilitate two key comparisons: the performance of different models against each other, and the effectiveness of the single-agent framework (top half) versus the multi-agent framework (bottom half). The domains are grouped by language, with English results on the left and Chinese on the right, separated by a distinct vertical line. As the distribution of questions per domain is

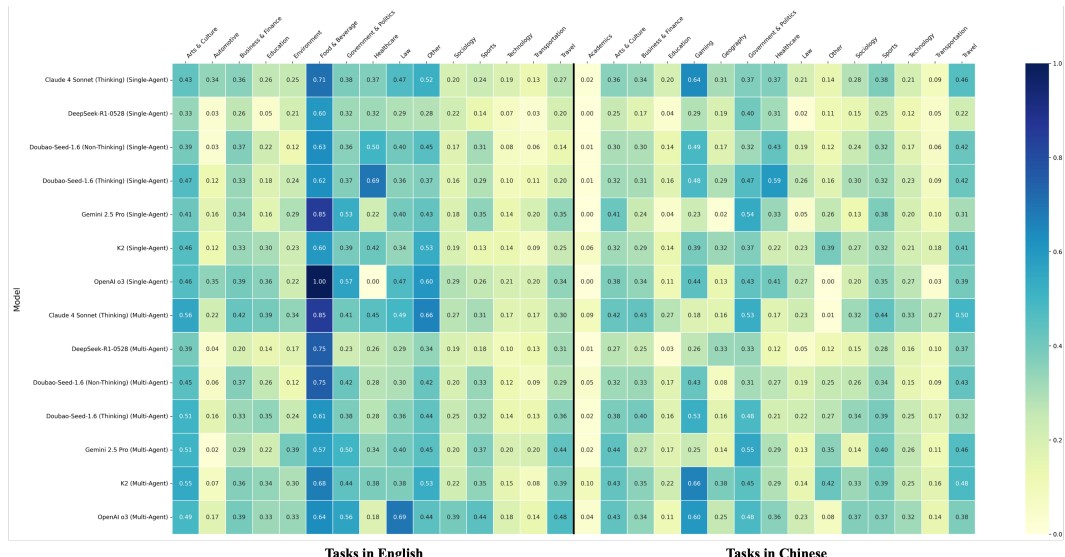

Figure J.1: Model performance (Row-level F1 Score) across domains and languages. The heatmap compares F1 scores for various models under single-agent and multi-agent frameworks. Models are evaluated on English and Chinese datasets across multiple domains using an automated pipeline. Darker blue indicates a higher score.

uneven, the heatmap serves primarily to identify general patterns rather than to provide a precise statistical comparison.

From the figure, we can draw several key observations. First, the multi-agent framework provides a consistent performance improvement across nearly all models and domains. This suggests that the collaborative approach is effective at enhancing the quality and relevance of the search results, regardless of the topic. Second, certain domains, such as "Academics" and "Transportation," prove to be more challenging for all models, likely due to the need for highly specialized and nuanced information. Finally, the results reveal model-specific aptitudes; for instance, some models show a clear advantage in domains like "Healthcare," while others excel in topics like "Automotive" or "Law." This domain-level insight is crucial for selecting the optimal model and framework for specific real-world applications.

## K    EXAMPLE TRAJECTORIES

In this section, we show the agent trajectory examples. For Single-Agent, we show an complete example as follows. For Multi-Agent, we show the trajectory of the main agent and omit the ones of sub-agent. The trajectories of sub-agent is like the trajectories of Single-Agent mode.

---

**Case Trajectory for Single-Agent**

**Question:** My son is about to start his university applications but he's still uncertain about both his major and which universities to apply to. Could you help me find the top five universities in each of the five broad subjects from the QS World University Rankings by Subject 2025, and also check their standings in the QS World University Rankings 2025 and the Times Higher Education World University Rankings 2025? And I need the home page of the university's official website, standard application deadline for regular decison as well as the application fee without the fee wavier.Please organize the results in one Markdown table with the following columns:
Subject, University, QS World University Rankings by Subject 2025, QS World University Rankings 2025, Times Higher Education World University Rankings 2025, Home Page, Application Deadline, Application Fee
Please use the universities' full official names in English.
Use only Arabic numerals in the ranking, for example: 1.
The output format is "'markdown
{data_content}
"'.

**Step 1**

**Reasoning Content**
The user's child is preparing for university applications and needs information about the top five universities in five major subjects based on the QS World University Rankings by Subject 2025. Additionally, they require each university's overall QS and THE rankings for 2025, official website URLs, regular decision application deadlines, and application fees (excluding fee waivers). All information must be organized into a Markdown table with specified columns.
First, the user's needs must be clarified. The key points include:
1. Five major subjects (but the user did not specify which ones; this is a problem).
2. Top five universities per subject in QS Subject Rankings 2025.
3. Each university's 2025 QS overall ranking and THE overall ranking.
4. Official website homepage, regular application deadline, and application fee.
The critical issue here is that the user mentioned "five broad subjects" but did not specify which ones. Common broad subjects might include Engineering, Medicine, Business & Management, Computer Science, Social Sciences, etc. However, since the user did not specify, I need to confirm. But since this is a search task and I cannot ask the user for clarification, I must infer or select typical subjects. For example, QS usually includes subjects like Engineering and Technology, Life Sciences and Medicine, Natural Sciences, Social Sciences and Management, Arts and Humanities. Alternatively, more specific subjects like Computer Science, Business, etc. However, this may affect the accuracy of the results. Since the user requires "broad subjects," perhaps choosing the five main subject areas from QS rankings would be appropriate. For example:
1. Engineering and Technology
2. Life Sciences and Medicine
3. Natural Sciences
4. Social Sciences and Management
5. Arts and Humanities
But I need to verify whether QS's 2025 subject rankings use these broad categories or more specific ones. For example, QS often has rankings for specific disciplines like Electrical Engineering, rather than broad "Engineering." This requires checking the structure of QS Subject Rankings 2025. Since the current time is April 2025, the 2025 rankings should already be released, but I need to confirm the specific subject divisions.
Next steps:

1. Determine the five broad subjects as per QS 2025 rankings. This requires searching for "QS World University Rankings by Subject 2025 broad subject categories" to identify the main subject areas.
2. For each of these five subjects, retrieve the top five universities.
3. For each university, collect:
- QS overall ranking 2025
- THE overall ranking 2025
- Official website homepage
- Regular decision application deadline (for undergraduate programs, assuming)
- Application fee (excluding waivers)
Challenges include:
- Ensuring the 2025 rankings are available. Since it's April 2025, they should be, but confirmation is needed.
- Application deadlines and fees vary by program and country. The user likely refers to undergraduate deadlines, but consistency must be maintained (e.g., for universities in the US, UK, etc.).
- Some universities may have different deadlines for different programs; the "regular decision" deadline for undergraduate studies is required.
- Application fees also vary by program and nationality; the user wants the standard fee without waivers, possibly for international students.
Now, step-by-step planning:
**Step 1: Confirm QS 2025 Subject Rankings' broad categories**
Search query: "QS World University Rankings by Subject 2025 broad subject areas" or check the official QS website to identify the five main subject groups. For example, QS typically divides into five broad areas: Arts & Humanities, Engineering & Technology, Life Sciences & Medicine, Natural Sciences, Social Sciences & Management. Assuming these five, proceed, but verification is needed via search.
**Step 2: For each of the five subjects, find top 5 universities**
For example, search "QS World University Rankings by Subject 2025 Engineering and Technology top 5" and repeat for each subject.
**Step 3: For each university, collect overall QS and THE 2025 rankings**
Search "QS World University Rankings 2025" and "Times Higher Education World University Rankings 2025" to find each university's overall rank.
**Step 4: Collect official website, application deadline, and fee**
For each university, visit their official admissions page to find the regular decision deadline and application fee for international undergraduate students. Note that deadlines may vary by program and country; the user likely wants the undergraduate regular deadline (e.g., for US universities, usually January/February, while UK universities may have earlier deadlines like January 15 for UCAS). Application fees also vary (e.g., US universities often charge $75-$100, UK may be £20-£25 for UCAS, but individual universities may differ).
Potential issues:
- Some universities may not be in the top 5 of multiple subjects, but the user requires top 5 per subject, so each subject will have five universities, possibly overlapping across subjects.
- For THE rankings, need to confirm if the 2025 edition is available. THE usually releases rankings in September, so as of April 2025, the 2025 rankings should exist (e.g., THE 2025 released in September 2024).
- Application deadlines and fees for 2025 entry: For example, US universities' 2025 fall entry deadlines may be in late 2024 (e.g., January 2025), but the user is asking in April 2025, which may be after the deadline. However, the user may need deadlines for the next intake (e.g., 2026 entry), but the query specifies "standard application deadline for regular decision" without a year. This requires clarification, but since the user mentions "about to start his university applications," it's likely for 2025 entry, though deadlines may have passed. Alternatively, 2026 entry deadlines may not yet be published. This is a potential ambiguity, but the user may expect the most recent available deadlines (e.g., 2025 entry deadlines, even if passed) or the upcoming 2026 deadlines. Need to handle carefully during searches.
**Implementation Plan:**

1. **Search for QS 2025 Subject Rankings' broad categories**
- Query: "QS World University Rankings by Subject 2025 broad subject categories"
- Aim: Confirm the five main subject areas. Assume they are Arts & Humanities, Engineering & Technology, Life Sciences & Medicine, Natural Sciences, Social Sciences & Management.
2. **For each subject, retrieve top 5 universities**
- Example queries: "QS World University Rankings by Subject 2025 Engineering and Technology top 5"
- Repeat for each of the five subjects.
3. **Collect QS and THE 2025 overall rankings for each university**
- Query: "QS World University Rankings 2025" and "Times Higher Education World University Rankings 2025"
- For each university, find their overall ranks.
4. **For each university, find official website, deadline, and fee**
- Official website: Usually straightforward (e.g., "Harvard University official website").
- Deadline and fee: Search "[University name] undergraduate regular decision application deadline 2025" and "[University name] undergraduate application fee international students".
**Challenges and Solutions:**
- **Subject selection uncertainty**: If QS 2025 uses different broad categories, adjust accordingly. For example, if "Computer Science" is a separate category instead of under Engineering, need to adapt.
- **Overlapping universities**: The same university may appear in multiple subjects (e.g., MIT in Engineering and Natural Sciences), requiring duplicate entries in the table with different subjects.
- **Missing THE rankings**: If a university isn't ranked by THE, note as "N/A" or check if data is available.
- **Deadline/fee variability**: Some universities have different deadlines for different programs; use the undergraduate regular decision deadline for the main campus. For fees, use the standard international undergraduate application fee without waivers.
**Search Steps:**
First, confirm the five broad subjects in QS 2025 Subject Rankings.
**Search 1: QS 2025 Subject Rankings broad categories**
Query: "QS World University Rankings by Subject 2025 broad subject areas"
Assume results confirm the five categories: Arts & Humanities, Engineering & Technology, Life Sciences & Medicine, Natural Sciences, Social Sciences & Management.
**Next, for each subject, search top 5 universities:**
**Subject 1: Engineering and Technology**
Query: "QS World University Rankings by Subject 2025 Engineering and Technology top 5"
Assume results: MIT (1), Stanford (2), Cambridge (3), Berkeley (4), Oxford (5) (hypothetical; actual data needed via search).
**Subject 2: Life Sciences and Medicine**
Query: "QS World University Rankings by Subject 2025 Life Sciences and Medicine top 5"
Example results: Harvard, Oxford, Cambridge, Stanford, MIT.
**Subject 3: Natural Sciences**
Query: "QS World University Rankings by Subject 2025 Natural Sciences top 5"
**Subject 4: Social Sciences and Management**
Query: "QS World University Rankings by Subject 2025 Social Sciences and Management top 5"
**Subject 5: Arts and Humanities**
Query: "QS World University Rankings by Subject 2025 Arts and Humanities top 5"
Once the top 5 universities per subject are identified, collect their QS and THE overall rankings.
**Search QS 2025 overall rankings:**
Query: "QS World University Rankings 2025 list"
**Search THE 2025 overall rankings:**

Query: "Times Higher Education World University Rankings 2025 list"
For each university, extract their ranks.
Next, for each university, find the official website (usually the main domain, e.g., www.harvard.edu), application deadline, and fee.
**Example for Harvard University:**
- Official website: https://www.harvard.edu/
- Deadline: January 1 (for 2025 entry, regular decision)
- Application fee: $85 (international students)
**Potential Issues During Search:**
- **Rankings not released**: If QS 2025 subject rankings are unavailable (though April 2025 should have them), use 2024 rankings and note, but the user specifically wants 2025.
- **Deadline discrepancies**: Some universities may have rolling admissions or different deadlines for different countries.
- **Application fee waivers**: Ensure the fee is the standard one without waivers.
**Data Organization:**
The final table will have rows for each university per subject. For example:
— Subject — University — QS Subject Rank — QS Overall — THE Overall — Home Page — Deadline — Fee —
——————————————————————————————————————————————————————————————————————————
————————————————————————
— Engineering & Tech — MIT — 1 — 1 — 2 — https://mit.edu/ — Jan 5 — $75 —
— ... — ... — ... — ... — ... — ... — ... — ... —
**Execution Steps:**
1. Confirm QS 2025 broad subjects via search.
2. For each subject, retrieve top 5 universities.
3. For each university, get QS and THE 2025 overall ranks.
4. Collect website, deadline, fee for each university.
5. Compile into the required Markdown table.
Now, proceed to simulate the search steps and data collection.
**Tool Call**

```
{
  "tool_name": "search_global",
  "arguments": "{\"query\":\"QS World University Rankings by
      ↪ Subject 2025 broad subject categories\",\"count\":5,\"
      ↪ summary_type\":\"short\"}",
  "tool_call_id": "call_tqp4ug60z97geyrce854oniz"
}

{
  "tool_name": "search_global",
  "arguments": "{\"query\":\"QS World University Rankings 2025\",\"
      ↪ count\":10,\"summary_type\":\"short\"}",
  "tool_call_id": "call_plyb3hwvjkkkbbb49j2pllqh"
}

{
  "tool_name": "search_global",
  "arguments": "{\"query\":\"Times Higher Education World
      ↪ University Rankings 2025\",\"count\":10,\"summary_type\":\"
      ↪ short\"}",
  "tool_call_id": "call_ozbw0cez7hqriym843bx5acv"
}
```

**Tool Response**

```
[index] 0
[siteName] qs.com
...
(search content omitted due to length limit)
```

**Step 2**

**Reasoning Content**

I now need to continue analyzing the user's query. The user wants the top five universities in each of the five broad subjects from the QS World University Rankings by Subject 2025, along with their overall rankings in QS and THE 2025, official website homepage, regular decision application deadline, and application fee. These details need to be organized into a markdown table.

First, reviewing previous search results: The first search (ID 0) confirmed that the five broad subjects in QS 2025 are Arts and Humanities, Engineering and Technology, Life Sciences and Medicine, Natural Sciences, and Social Sciences and Management. However, specific rankings for each subject were not provided. The second search (ID 1) retrieved QS overall rankings but only mentioned the number of institutions and countries without specific university rankings. The third search (ID 2) for THE rankings only provided limited information, such as the University of Melbourne at 47th place and Johns Hopkins at 16th, but lacked comprehensive THE 2025 rankings.

Next, it is necessary to obtain the top five universities for each of the five subjects in QS by Subject 2025. Previous searches only identified the subject categories but not the specific rankings. Therefore, further searches for each subject's top five universities are required. For example, search queries like "QS World University Rankings by Subject 2025 Arts and Humanities top 5" for each subject separately.

Additionally, the overall QS and THE 2025 rankings for these universities are needed. Previous QS overall rankings results were too vague, so the complete QS 2025 overall ranking list must be retrieved. Similarly, THE 2025 rankings require more detailed data, such as the top 100 universities or a searchable database.

Furthermore, the user needs each university's official website homepage, application deadline, and application fee. These details must be obtained from each university's official website or reliable educational platforms. However, this information may vary by program and country, so it is necessary to confirm the general undergraduate regular decision deadlines and fees for each university.

Current missing information includes:
1. Top five universities for each of the five QS subjects.
2. QS 2025 overall rankings for these universities.
3. THE 2025 overall rankings for these universities.
4. Official website URLs, application deadlines, and fees for each university.

Next steps:
- For each of the five subjects, search for the top five universities in QS by Subject 2025. For example, use queries like "QS 2025 Arts and Humanities top 5 universities".
- Search for the complete QS 2025 overall rankings to find the rankings of these universities.
- Search for the complete THE 2025 overall rankings to find their respective positions.
- For each identified university, search for their official website, application deadlines, and fees. This may require visiting each university's admissions page or using educational portals like Common App for deadline information.

Potential challenges:
- QS and THE rankings might be published on official websites requiring access to detailed reports or tables.
- University-specific application deadlines and fees could vary by program, so the user may need the general undergraduate deadline, which may require checking each university's admissions page.

Given the complexity, the process should be broken down into steps:
1. Retrieve top five universities for each of the five QS subjects.
2. Collect QS and THE overall rankings for these universities.
3. Gather official websites, deadlines, and fees for each university.

Now, the first priority is to obtain the top five universities per subject. For example, searching for each subject individually:

Search queries could include:
- "QS World University Rankings by Subject 2025 Arts and Humanities top 5"
- "QS 2025 Engineering and Technology top universities"

- And similarly for the remaining three subjects.

Once these universities are identified, their overall rankings in QS and THE can be searched, followed by their specific application details.

Additionally, for QS and THE overall rankings, more precise searches are needed, such as "QS World University Rankings 2025 top 100" or accessing the official QS ranking table. Similarly for THE.

In summary, the next steps require multiple searches to collect detailed ranking information and university-specific application details. Given the volume of information, it may be necessary to search each subject's rankings separately and then verify each university's data individually.

**Tool Call**

```
{
  "tool_name": "search_global",
  "arguments": "{\"query\":\"QS 2025 Arts and Humanities top 5
      ↪ universities\",\"count\":5,\"summary_type\":\"short\"}",
  "tool_call_id": "call_hu7xl49h1kbwt7pby1xyxcnn"
}

{
  "tool_name": "search_global",
  "arguments": "{\"query\":\"QS 2025 Engineering and Technology top
      ↪ 5\",\"count\":5,\"summary_type\":\"short\"}",
  "tool_call_id": "call_4ykb7yycmve7n59kbg3626jo"
}

{
  "tool_name": "search_global",
  "arguments": "{\"query\":\"QS 2025 Life Sciences and Medicine top
      ↪ 5\",\"count\":5,\"summary_type\":\"short\"}",
  "tool_call_id": "call_kgb5q873xe96ixw1u7juoupf"
}

{
  "tool_name": "search_global",
  "arguments": "{\"query\":\"QS 2025 Natural Sciences top 5
      ↪ universities\",\"count\":5,\"summary_type\":\"short\"}",
  "tool_call_id": "call_a8omvnzy8sw7gntarrjdyvbz"
}

{
  "tool_name": "search_global",
  "arguments": "{\"query\":\"QS 2025 Social Sciences and Management
      ↪  top 5\",\"count\":5,\"summary_type\":\"short\"}",
  "tool_call_id": "call_tqu4khdcrnsf2oa2ff011w23"
}
```

**Tool Response**

```
[index] 0
[siteName] qs.com
...
(search content omitted due to length limit)
```

**Step 3**

**Reasoning Content**

Now I need to continue analyzing the user's question. The user wants the top five universities in each of the five major subjects from the QS World University Rankings by Subject 2025, along with their overall rankings in QS and THE 2025, official website homepage, regular application deadline, and application fee (without fee waiver). This information needs to be organized into a Markdown table.

First, previous searches have confirmed that the five major subjects in QS are: Arts and Humanities, Engineering and Technology, Life Sciences and Medicine, Natural Sciences, and Social Sciences and Management. The next step is to obtain the top five universities for each subject. Based on the results of the second round of searches (IDs 0-4), the information for each subject is as follows:

1. **Arts and Humanities**: From search results, ID 0 shows Harvard, Oxford, Cambridge, Stanford, but there is inconsistency with ID 1 (mentioning Stanford, Carnegie Mellon, etc.). The information is fragmented and lacks a complete top five list.

2. **Engineering and Technology**: Search result ID 1 mentions MIT, Oxford, Stanford, but the list is incomplete, with only partial rankings.

3. **Life Sciences and Medicine**: Search results are vague, mentioning Duke, University of Copenhagen, etc., but no clear top five ranking.

4. **Natural Sciences**: ID 3 mentions MIT is first in 11 subjects, including Natural Sciences, but the top five list is unclear. ID 0 shows Harvard, Oxford, Cambridge, Stanford, but it is unclear which subject this belongs to.

5. **Social Sciences and Management**: ID 4 mentions LSE ranked sixth globally, but lacks the top five list.

Clearly, the current subject rankings are incomplete or conflicting, requiring more precise searches. For example, searching for "QS 2025 Arts and Humanities top 5 universities" may yield more accurate results. Additionally, the overall QS and THE rankings, application deadlines, and fees for each university are still missing.

Next steps:
- For each subject, conduct a separate search to confirm the top five universities (e.g., "QS 2025 Arts and Humanities top 5 universities").
- Once the universities are confirmed, search for their QS and THE 2025 overall rankings.
- For each university, find official website URLs, regular application deadlines, and application fees.

Since the user requires detailed information for each university, a significant amount of search work is needed. For example, after confirming Harvard University is first in Arts and Humanities, further searches are required for its QS overall ranking, THE ranking, homepage URL, deadline, and fee. This process must be repeated for each university across all five subjects, totaling 25 universities. However, considering potential overlaps (e.g., Harvard may rank highly in multiple subjects), efficiency can be improved by consolidating searches for overlapping universities.

Current search results are insufficient, so continued searches are necessary. First, prioritize completing the top five universities for each subject, then gather their overall rankings and application information.

**Tool Call**

```
{
  "tool_name": "search_global",
  "arguments": " {\"query\": \"QS 2025 Arts and Humanities top 5
      ↪ universities official list\", \"count\": 3, \"summary_type
      ↪ \": \"short\"}",
  "tool_call_id": "call_j2tuhp5gh1tqckfperk9sxtv"
}

{
  "tool_name": "search_global",
  "arguments": " {\"query\": \"QS 2025 Engineering and Technology
      ↪ top 5 universities\", \"count\": 3, \"summary_type\": \"
      ↪ short\"}",
  "tool_call_id": "call_twfz734e6bw8llv551sdnkvp"
}

{
  "tool_name": "search_global",
  "arguments": " {\"query\": \"QS 2025 Life Sciences and Medicine
      ↪ top 5 universities\", \"count\": 3, \"summary_type\": \"
      ↪ short\"}",
```

```
  "tool_call_id": "call_xkxvd8heseo7dmlw94hcvd3q"
}

{
  "tool_name": "search_global",
  "arguments": " {\"query\": \"QS 2025 Natural Sciences top 5
      ↪ universities official\", \"count\": 3, \"summary_type\": \"
      ↪ short\"}",
  "tool_call_id": "call_wm9saujg9hzf49mivzpdegsx"
}

{
  "tool_name": "search_global",
  "arguments": " {\"query\": \"QS 2025 Social Sciences and
      ↪ Management top 5\", \"count\": 3, \"summary_type\": \"short
      ↪ \"}",
  "tool_call_id": "call_cbe9gtmyv592gs2uud4keazv"
}
```

**Tool Response**

```
[index] 0
[siteName] qs.com
...
(search content omitted due to length limit)
```

| Step 4 |
|---|

**Reasoning Content**

Now I need to continue analyzing the user's query. The user wants the top five universities in each of the five broad subjects from the QS World University Rankings by Subject 2025, along with their QS overall rankings, THE rankings, official website URLs, regular decision deadlines, and application fees. Previous search attempts have been made, but the results remain incomplete.

First, reviewing the earlier search results, the five broad subjects have been confirmed: Arts and Humanities, Engineering and Technology, Life Sciences and Medicine, Natural Sciences, and Social Sciences and Management. However, the top five universities for each subject are still unclear. For example, search results for Arts and Humanities mention Harvard, Oxford, Cambridge, and Stanford, but the fifth university is missing. Similarly, Engineering and Technology mentions MIT and Georgia Tech but lacks the full top five. Life Sciences and Medicine, Natural Sciences, and Social Sciences and Management also have incomplete or conflicting information.

Next, regarding the QS overall rankings and THE rankings, the previous search results for QS 2025 only provided basic information about the number of participating institutions and countries but did not list specific university rankings. THE rankings similarly lack detailed data, with only mentions of the University of Melbourne at 47th and Johns Hopkins at 16th, which is insufficient to cover all required universities.

Additionally, the user needs official website URLs, application deadlines, and fees. These details were not addressed in previous searches and will require separate research for each university.

Therefore, further searches are needed:
1. For each of the five subjects, confirm the exact top five universities and their QS subject rankings.
2. Find the QS 2025 overall rankings and THE 2025 rankings for these universities.
3. Collect each university's official website URL, regular decision deadline, and application fee.

Since the user requires detailed information for each university, a structured approach is necessary. First, clarify the top five universities per subject, then gather their overall rankings, and finally collect application-related details. However, given the time constraints, prioritizing the top five universities for each subject and their rankings may be necessary before proceeding to application details.

Current search results still lack clear top-five lists for each subject. For example, in the latest search for Arts and Humanities, topuniversities.com mentions Harvard, Stanford, Edinburgh, Princeton, Peking University, and Tokyo University, but it is unclear which are the top five. More precise official rankings are needed, possibly by accessing the official QS website or authoritative sources.

Similarly, Engineering and Technology search results mention MIT and Georgia Tech but lack the complete top five. Life Sciences and Medicine lists Harvard, Stanford, MIT, and Duke, but the fifth is missing. Natural Sciences mentions Tsinghua, Tokyo University, Toronto, Cornell, and Chicago, but their order is unclear. Social Sciences and Management includes Toronto, Melbourne, Princeton, and Northwestern, with LSE ranked sixth, but the top five remain uncertain.

Thus, it is necessary to conduct more targeted searches for each subject, using queries like "QS 2025 [Subject] top 5 universities official list" to obtain accurate rankings. Once the universities are confirmed, their QS and THE overall rankings can be searched individually. Finally, each university's official website, deadline, and fee information must be collected, which may require visiting each university's admissions page or reliable educational platforms.

In summary, current information is insufficient, and further searches are required for each subject's top five universities, followed by their overall rankings and application details. Given the complexity, it may be necessary to search each subject separately and then verify each university's information individually.

**Tool Call**

```
{
  "tool_name": "search_global",
  "arguments": " {\"query\": \"QS 2025 Arts and Humanities top 5
      ↪ universities official\", \"count\": 1, \"summary_type\": \"
      ↪ short\"}",
  "tool_call_id": "call_796e3kmb6m93z5kxgzkzpaxv"
}

{
  "tool_name": "search_global",
  "arguments": " {\"query\": \"QS 2025 Engineering and Technology
      ↪ top 5 official\", \"count\": 1, \"summary_type\": \"short
      ↪ \"}",
  "tool_call_id": "call_q2hqpgtcz1wwj3g65okfnr3f"
}

{
  "tool_name": "search_global",
  "arguments": " {\"query\": \"QS 2025 Life Sciences and Medicine
      ↪ top 5 list\", \"count\": 1, \"summary_type\": \"short\"}",
  "tool_call_id": "call_brxb0h0ob6inf2n1e2a2p2uo"
}

{
  "tool_name": "search_global",
  "arguments": " {\"query\": \"QS 2025 Natural Sciences top 5
      ↪ universities list\", \"count\": 1, \"summary_type\": \"
      ↪ short\"}",
  "tool_call_id": "call_q0ykprzmiecs8dogsv5wn06v"
}

{
  "tool_name": "search_global",
  "arguments": " {\"query\": \"QS 2025 Social Sciences and
      ↪ Management top 5 list\", \"count\": 1, \"summary_type\": \"
      ↪ short\"}",
  "tool_call_id": "call_hn90i1jx46pfy2qqxdomnka8"
}
```

**Tool Response**

```
[index] 0
[siteName] qs.com
...
(search content omitted due to length limit)
```

### Step 5

**Reasoning Content**

I now need to continue analyzing the user's query and the existing search results to determine the next steps. The user requires the top five universities in each of the five broad subject areas from the QS World University Rankings by Subject 2025, along with their overall QS and THE rankings, official website URLs, regular decision application deadlines, and application fees. The previous search attempts have yielded some information, but several critical points remain unclear.

First, the five broad subject areas identified from the initial search results are: Arts and Humanities, Engineering and Technology, Life Sciences and Medicine, Natural Sciences, and Social Sciences and Management. This information is clear and requires no further confirmation.

Next, for each subject area's top five universities, previous searches have provided partial results, but inconsistencies exist. For example, in Arts and Humanities, search result 0 from the first round mentions Harvard, Oxford, Cambridge, Stanford, and MIT, while another result lists Harvard, Stanford, Carnegie Mellon, etc. These discrepancies suggest unreliable data that may stem from incomplete or misaligned search results. Similarly, for Engineering and Technology, some results indicate MIT as first, while others mention Harvard, creating confusion. It appears the search results lack a definitive official list for each subject area's top five universities.

Furthermore, even if the top five universities per subject are confirmed, their overall QS and THE rankings for 2025 remain unknown. For instance, while the user needs the QS World University Rankings 2025 and THE rankings, existing search results for QS overall rankings mention data like the number of institutions but not specific university rankings. Similarly, THE rankings only reference specific universities (e.g., University of Melbourne at 47th) without a comprehensive list. Thus, retrieving the overall rankings for these universities is essential.

Additionally, the user requires official website URLs, application deadlines, and application fees. None of the current search results include this information, necessitating separate searches for each university's details.

Given these gaps, the next steps should include:

1. **Confirm the top five universities for each of the five subject areas**: Previous searches failed to yield clear, consistent official lists. More precise searches using queries like "QS 2025 [Subject] top 5 universities official list" may help, potentially requiring direct access to QS's official website or authoritative sources.

2. **Retrieve overall QS and THE 2025 rankings for these universities**: Once the universities are confirmed, search for their positions in the overall QS and THE rankings. For example, search "[University Name] QS World University Rankings 2025" and similarly for THE.

3. **Collect official website URLs, application deadlines, and fees**: For each confirmed university, search their official admissions pages for regular decision deadlines and application fees. This may require visiting each university's website individually or using reliable educational platforms.

However, considering the limitations of search tools—such as potential failure to retrieve detailed ranking lists or university-specific information—prioritizing confirmation of the top five universities per subject is critical. Without this foundation, subsequent steps cannot proceed. For example, if Arts and Humanities rankings remain unconfirmed, subsequent searches for overall rankings and application details will be meaningless.

Another challenge is that search results for QS subject rankings are fragmented. For instance, multiple attempts to search for Engineering and Technology rankings have returned inconsistent results, possibly due to ambiguous query wording or scattered official data. Us-

ing more precise keywords like "QS World University Rankings by Subject 2025 [Subject] top 5" or accessing the official QS ranking page directly (via text_browser_view) might resolve this.

For example, the first search result (name=0) includes a link to qs.com's article on subject rankings. Using text_browser_view to access this link could provide the complete top five list for each subject. Similarly, links from topuniversities.com (QS's official partner site) might contain detailed rankings.

Thus, the next logical step is to use the text_browser_view tool to open authoritative links from previous searches (e.g., the qs.com article link) and extract the official top five universities for each subject. This would ensure accuracy and resolve inconsistencies in prior search results.

Once the subject rankings are confirmed, searches for overall QS and THE rankings can proceed. For example, if Harvard University tops Arts and Humanities, search for "Harvard University QS World University Rankings 2025" and "Harvard University THE World University Rankings 2025" to obtain their overall positions.

Finally, for each university, search their official website for application deadlines and fees. For instance, querying "Harvard University undergraduate application deadline 2025" and "Harvard University application fee" would yield the required information.

In summary, the immediate priority is to confirm the top five universities per subject using text_browser_view on authoritative links. This will provide a reliable foundation for subsequent searches. Without this step, data inconsistencies will persist, hindering the completion of the user's requested table.

**Tool Call**

```
{
  "tool_name": "text_browser_view",
  "arguments": "{\"url\":\"https://www.qs.com/insights/articles/ qs
      ↪ -world-university-rankings-by-subject-2025-results-
      ↪ analysis/\", \"description\":\"Extract the top 5
      ↪ universities for each of the five broad subject areas in QS
      ↪  World University Rankings by Subject 2025\"}",
  "tool_call_id": "call_mrzqgwfaew3kf2nmeqwv05iz"
}
```

**Tool Response**

```
Page url:https://www.qs.com/insights/articles/ qs-world-university-
    ↪ rankings-by-subject-2025-results-analysis/
Page Summary:
...
(search content omitted due to length limit)
```

| Step 6 |
| --- |

**Reasoning Content**

Now I need to continue analyzing the user's query and determine the next steps. The user requires the top five universities in each of the five broad subject areas of the QS World University Rankings by Subject 2025, along with their QS overall ranking, THE ranking, official website homepage, regular decision application deadline, and application fee.

First, based on previous search results, I initially searched for the five broad subject categories of QS 2025 and confirmed them as Arts and Humanities, Engineering and Technology, Life Sciences and Medicine, Natural Sciences, and Social Sciences and Management. Then, I attempted to search for the top five universities in each subject area, but the initial results were incomplete or inconsistent. Therefore, I used the text_browser_view tool to access the official QS article link and successfully obtained the top five universities for each subject area:

- Arts and Humanities: Harvard, Oxford, Cambridge, Stanford, UC Berkeley
- Engineering and Technology: MIT, Oxford, Stanford, Cambridge, ETH Zurich
- Life Sciences and Medicine: Harvard, Oxford, Johns Hopkins, Stanford, MIT
- Natural Sciences: Harvard, MIT, Oxford, Cambridge, Stanford
- Social Sciences and Management: Harvard, Oxford, Stanford, MIT, Cambridge

Next, the remaining information to collect includes:
1. QS World University Rankings 2025 (overall ranking) for each university
2. Times Higher Education World University Rankings 2025 for each university
3. Official website homepage URL
4. Regular decision application deadline
5. Application fee (without fee waiver)
Current status:
- QS subject rankings are confirmed.
- QS overall rankings and THE rankings have been searched but not yet obtained. Previous search results for QS overall rankings (name=1) only mentioned the number of universities and countries but not specific rankings. THE rankings (name=2) also lack complete data, with only some universities mentioned (e.g., University of Melbourne at 47th, Johns Hopkins at 16th).
- Official website URLs, deadlines, and fees have not been searched yet.
Next steps:
1. **Obtain QS 2025 overall rankings for each university**: Need to search for the QS World University Rankings 2025 list, focusing on the top 50 universities to find the rankings of the 25 universities (5 subjects × 5 universities each, but there may be overlaps, such as Harvard appearing in multiple subjects).
2. **Obtain THE 2025 overall rankings**: Similarly, search for the THE World University Rankings 2025 list to find the rankings of these universities.
3. **Collect official website URLs**: For each university, search for their official homepage URL.
4. **Collect application deadlines and fees**: For each university, search for their undergraduate regular decision deadline and application fee (excluding fee waivers).
Since these universities are mostly top global institutions, their overall rankings and THE rankings may be available in official ranking releases. However, previous searches for QS and THE rankings did not yield detailed lists, so more precise searches are needed, such as "QS World University Rankings 2025 top 50" or searching for each university individually (e.g., "Harvard University QS ranking 2025").
Additionally, application deadlines and fees may vary by university and program, but the user specified "standard application deadline for regular decision" and "application fee without fee waiver," which should be obtainable from university admissions websites or official undergraduate admissions pages.
Given the large number of universities (approximately 25 unique institutions), it may be efficient to first compile a list of unique universities, then search for their QS overall ranking, THE ranking, website, deadline, and fee one by one. Alternatively, search for QS and THE ranking lists first to extract all required university rankings, then handle website, deadline, and fee information.
Possible steps:
- **Step 1**: Extract unique universities from the five subject rankings to avoid redundant searches. For example, Harvard, Oxford, Stanford, MIT, Cambridge, etc., appear in multiple subjects.
- **Step 2**: Search for QS World University Rankings 2025 top 100 to obtain overall rankings for these universities.
- **Step 3**: Search for THE World University Rankings 2025 top 100 to obtain their THE rankings.
- **Step 4**: For each unique university, search for their official website, regular decision deadline, and application fee.
However, considering time constraints and search efficiency, using batch searches may be better. For example:
- Search "QS World University Rankings 2025 top 50 list" to get a list of universities and their rankings.
- Similarly, search "THE World University Rankings 2025 top 50 list."
- For website URLs, deadlines, and fees, since each university's information is on their official site, it may be necessary to search individually for each university, such as "Harvard University undergraduate application deadline 2025" and "Harvard University application

fee 2025."

But given the number of universities (approximately 15 unique institutions), this could be time-consuming. Alternatively, use parallel searches where possible, but the tool may limit the number of concurrent searches.

Current missing information is critical, so further searches are required. Next, prioritize obtaining QS and THE overall rankings, followed by website URLs, deadlines, and fees.

First, search for QS 2025 overall rankings. Previous attempts (name=1) used the query "QS World University Rankings 2025," but results only mentioned the number of institutions without specific rankings. Need to adjust the query to "QS World University Rankings 2025 top 50" or "QS World University Rankings 2025 list" to get detailed rankings.

Similarly, for THE rankings, previous searches (name=2) lacked complete data. Need to search "Times Higher Education World University Rankings 2025 top 50" or similar to obtain the full list.

Once QS and THE rankings are obtained, proceed to collect website URLs, deadlines, and fees. For these, individual university searches may be necessary, such as "University of Oxford official website" or "MIT undergraduate application deadline 2025."

Therefore, the next steps should be:

1. Search for QS 2025 overall ranking list to extract rankings for the 25 universities.
2. Search for THE 2025 overall ranking list to extract their rankings.
3. For each university, search for official website, deadline, and fee.

Since the user requires organizing the data into a table, all information must be complete. Currently, QS and THE rankings are the most urgent missing pieces, followed by website URLs, deadlines, and fees.

Thus, the next search should focus on obtaining the QS 2025 and THE 2025 overall ranking lists.

**Tool Call**

```
{
  "tool_name": "search_global",
  "arguments": "{\"query\":\"QS World University Rankings 2025 top
      ↪ 50 list\",\"count\":5,\"summary_type\":\"short\"}",
  "tool_call_id": "call_35s50vzbgkie220sca9jbis5"
}

{
  "tool_name": "search_global",
  "arguments": "{\"query\":\"Times Higher Education World
      ↪ University Rankings 2025 top 50\",\"count\":5,\"
      ↪ summary_type\":\"short\"}",
  "tool_call_id": "call_k6qy347lzof4765gpf57dz0z"
}
```

**Tool Response**

```
[index] 0
[siteName] qs.com
...
(search content omitted due to length limit)
```

<table>
<tr><td align="center">Step 7</td></tr>
</table>

**Reasoning Content**

I now need to continue analyzing the user's question to determine the next steps. The user wants the top five universities in each of the five major subject areas from the QS World University Rankings by Subject 2025, along with their overall QS 2025 rankings, THE 2025 rankings, official website homepage, regular decision application deadline, and application fee. This information needs to be organized into a Markdown table.

First, based on previous searches, the five major subject areas have been confirmed: Arts and Humanities, Engineering and Technology, Life Sciences and Medicine, Natural Sciences, and Social Sciences and Management. Through the text_browser_view tool, the top five universities for each subject area have been obtained. For example, the top five in Arts

and Humanities are Harvard, Oxford, Cambridge, Stanford, and UC Berkeley.

Next, the QS overall rankings and THE rankings for these universities are required. From search results, the top five QS overall rankings are MIT, Imperial College London, Oxford, Harvard, and Cambridge. However, the user needs the rankings of all 25 universities (5 subjects × 5 universities), which may extend beyond the top five. For instance, Stanford might rank sixth in QS overall, requiring confirmation. Similarly, THE rankings are not fully provided in current search results; only partial information like Harvard and MIT's positions is available, while other universities' THE rankings remain unclear.

Additionally, the official website homepage, application deadline, and application fee for each university must be collected. These details are not covered in existing search results and require further searches for each university. For example, confirming Harvard's official website URL, regular decision deadline (typically January 1st), and application fee (around $85), but specific 2025 data may change and need verification.

Current missing information includes:

1. QS overall rankings for all 25 universities.
2. THE overall rankings for all 25 universities.
3. Official website URLs for each university.
4. Regular decision application deadlines for 2025.
5. Application fees (excluding fee waivers) for 2025.

Next steps:

- Search for the complete QS World University Rankings 2025 to obtain overall rankings for each university.
- Search for the complete THE World University Rankings 2025 for the same purpose.
- For each university, search its official website to confirm the homepage URL.
- Check each university's admissions page for 2025 regular decision deadlines and application fees.

Given the large volume of universities involved, parallel searches may be necessary. For example, first retrieve the full QS and THE rankings, then batch-search website URLs, deadlines, and fees. However, due to tool limitations, searches must be conducted step-by-step.

First, prioritize obtaining complete QS and THE rankings. Previous searches showed incomplete QS top 50 results, so further searches for the full QS 2025 rankings are needed. Similarly, THE rankings lack comprehensive data, requiring additional searches.

Once the overall rankings are confirmed, proceed to collect each university's official website, deadline, and fee. This may require searching each university individually, which is time-consuming but essential for accuracy.

In summary, the next steps involve:

1. Searching for the complete QS World University Rankings 2025 to get all 25 universities' overall rankings.
2. Searching for the complete THE World University Rankings 2025 for the same.
3. For each university, searching official website URLs, application deadlines, and fees.

Due to the complexity, multiple search steps will be required, potentially using parallel tool calls to improve efficiency. For example, searching QS and THE rankings simultaneously, then processing university-specific information in batches.

**Tool Call**

```
{
  "tool_name": "search_global",
  "arguments": " {\"query\": \"QS World University Rankings 2025
      ↪ full list\", \"count\": 10, \"summary_type\": \"short\"}",
  "tool_call_id": "call_v36pk5fyz3452obs81mlt492"
}

{
  "tool_name": "search_global",
  "arguments": " {\"query\": \"Times Higher Education World
      ↪ University Rankings 2025 full list\", \"count\": 10, \"
      ↪ summary_type\": \"short\"}",
  "tool_call_id": "call_3ae9mxm8edk2bloqggzyv485"
}
```

**Tool Response**

```
[index] 0
[siteName] qs.com
...
(search content omitted due to length limit)
```

**Step 8**

**Reasoning Content**

I now need to continue analyzing the user's query and existing search results to determine whether further searches are required. The user wants the top five universities in each of the five broad subjects from the QS World University Rankings by Subject 2025, along with their positions in the QS World University Rankings 2025 and Times Higher Education World University Rankings 2025, official website homepage, regular decision application deadline, and application fee (excluding fee waivers). The results should be organized into a Markdown table.

First, the previous search has successfully confirmed the five broad subjects: Arts and Humanities, Engineering and Technology, Life Sciences and Medicine, Natural Sciences, and Social Sciences and Management. Through text_browser_view, the top five universities for each subject were obtained:

- Arts and Humanities: Harvard, Oxford, Cambridge, Stanford, UC Berkeley
- Engineering and Technology: MIT, Oxford, Stanford, Cambridge, ETH Zurich
- Life Sciences and Medicine: Harvard, Oxford, Johns Hopkins, Stanford, MIT
- Natural Sciences: Harvard, MIT, Oxford, Cambridge, Stanford
- Social Sciences and Management: Harvard, Oxford, Stanford, MIT, Cambridge

Next, I need to find the QS World University Rankings 2025 and Times Higher Education (THE) 2025 rankings for these universities. The user's initial search for QS rankings found results indicating MIT is 1st, Imperial College London 2nd, Oxford 3rd, Harvard 4th, and Cambridge 5th. However, the list of top 50 QS rankings obtained later only partially mentions some universities, and full rankings are unclear. For example, Stanford's QS ranking is unspecified, as are UC Berkeley, ETH Zurich, Johns Hopkins, etc. Therefore, more detailed QS 2025 rankings are needed to confirm each university's position.

Regarding THE rankings, previous searches show fragmented information, such as Harvard being 1st, MIT and Oxford tied for 2nd, and Cambridge ranked 4th. However, there is insufficient data for universities like Stanford, UC Berkeley, ETH Zurich, and Johns Hopkins. The THE official website may require a more precise search for the complete 2025 rankings to obtain accurate positions for each university.

Additionally, the user needs official website URLs, regular decision deadlines, and application fees. This information is not covered in existing search results and must be obtained by searching each university's official admissions page. For example, Harvard's deadline is typically early January, with an application fee of $85, but confirmation is needed for 2025. Other universities like Oxford and MIT may have different deadlines and fees, requiring individual verification.

Current missing information includes:
1. QS 2025 overall rankings for each university (e.g., Stanford, UC Berkeley, ETH Zurich, Johns Hopkins)
2. THE 2025 overall rankings for each university
3. Official website URLs for each university
4. Regular decision application deadlines for 2025
5. Application fees (excluding fee waivers) for 2025

Next steps:
- Search for the complete QS World University Rankings 2025 to confirm the overall rankings of all 25 universities (5 subjects × 5 universities)
- Search for the complete THE World University Rankings 2025 to confirm their overall rankings
- For each university, search its official website to find the admissions page, record the homepage URL, regular decision deadline, and application fee

Since each university's deadline and fee may differ and require individual verification, this

could involve numerous searches. However, to ensure accuracy, it may be necessary to search each university separately. Alternatively, seek aggregated data, but given the specificity of the requirements, direct searches on official websites are more reliable.

Is further searching needed? Yes. Current information gaps include overall rankings for multiple universities, official website URLs, deadlines, and fees. These must be addressed through additional searches.

**Tool Call**

```
{
  "tool_name": "search_global",
  "arguments": " {\"query\": \"QS World University Rankings 2025
      ↪ full list top 100\", \"count\": 5, \"summary_type\": \"
      ↪ short\"}",
  "tool_call_id": "call_g27znvoi9uq9xhvqnee7dchb"
}

{
  "tool_name": "search_global",
  "arguments": " {\"query\": \"Times Higher Education World
      ↪ University Rankings 2025 top 100\", \"count\": 5, \"
      ↪ summary_type\": \"short\"}",
  "tool_call_id": "call_57j11arg3afrmbwmeb0wqu21"
}

{
  "tool_name": "search_global",
  "arguments": " {\"query\": \"Harvard University official website
      ↪ \", \"count\": 1, \"summary_type\": \"short\"}",
  "tool_call_id": "call_rypgud7jaj3ig6dbvledvlro"
}

{
  "tool_name": "search_global",
  "arguments": " {\"query\": \"University of Oxford official
      ↪ website\", \"count\": 1, \"summary_type\": \"short\"}",
  "tool_call_id": "call_z66541b2td4167ej3sibs6pi"
}

{
  "tool_name": "search_global",
  "arguments": " {\"query\": \"University of Cambridge official
      ↪ website\", \"count\": 1, \"summary_type\": \"short\"}",
  "tool_call_id": "call_79cb28tgy4vr9ku9jro1oiuj"
}
```

**Tool Response**

```
[index] 0
[siteName] topuniversities.com
...
(search content omitted due to length limit)
```

**Step 9**

**Reasoning Content**

Now I need to continue analyzing the user's question and existing search results to determine the next steps. The user requires the top five universities in each of the five broad subjects from the QS World University Rankings by Subject 2025, along with their overall rankings in QS and THE 2025, official website, application deadlines, and fees. These need to be organized into a Markdown table.

First, based on previous searches, the five broad subjects have been confirmed: Arts and Humanities, Engineering and Technology, Life Sciences and Medicine, Natural Sciences, and Social Sciences and Management. Through text_browser_view, the top five universities for

each subject have been obtained, such as Harvard University and MIT. This part is relatively clear, but there may be a need to verify consistency across sources.

Next, regarding the overall QS and THE rankings for these universities, existing search results show partial information. For example, QS's top five include MIT, Imperial College London, Oxford, Harvard, and Cambridge. However, the user needs the rankings of each university in both QS and THE. For instance, where does Stanford rank in QS overall? The current search results do not provide a complete top 100 list, making it impossible to accurately fill in the rankings for all 25 universities (5 subjects × 5 universities).

Additionally, official websites for some universities have been found, such as Oxford's ox.ac.uk and Cambridge's cam.ac.uk, but others like Stanford, MIT, and UC Berkeley still need verification. Application deadlines and fees are completely missing; these require accessing each university's official website or reliable admission pages to obtain, which will be time-consuming.

Current missing information includes:

1. QS and THE 2025 overall rankings for each of the 25 universities.

2. Official websites for all universities (some confirmed, others pending).

3. Regular decision application deadlines and fees for each university.

Next steps should prioritize obtaining the complete QS and THE 2025 rankings to fill in the overall ranking columns. If direct access to the full list is unavailable, searching for each university individually (e.g., "Stanford University QS World University Rankings 2025") might be necessary. However, this could be inefficient. Alternatively, searching for "QS World University Rankings 2025 top 100 list" or similar keywords might yield a complete ranking table.

For application deadlines and fees, since each university's official website must be consulted, this will require multiple searches or direct access via text_browser_view. For example, searching for "Harvard University undergraduate application deadline 2025" and "application fee" for each institution. However, given the large number of universities involved, this process will be time-intensive and may require parallel searches.

Additionally, verifying official websites is crucial to avoid incorrect links. For example, while Harvard's official site is likely harvard.edu, confirmation through authoritative sources is necessary.

Summary of next steps:

1. Continue searching for complete QS and THE 2025 top 100 rankings to fill in overall rankings for all 25 universities.

2. Collect official websites for all universities (some already found, others pending).

3. For each university, search for regular decision application deadlines and fees, preferably accessing official admissions pages directly.

Given the complexity, prioritizing the acquisition of overall rankings first is advisable, as this is critical for table completion. Application deadlines and fees may require subsequent searches or direct extraction from university websites via text_browser_view.

**Tool Call**

```
{
  "tool_name": "search_global",
  "arguments": " {\"query\": \"QS World University Rankings 2025
      ↪ top 100 list PDF\", \"count\": 3, \"summary_type\": \"short
      ↪ \"}",
  "tool_call_id": "call_o32n2jlg1vonl0rdtactogfm"
}

{
  "tool_name": "search_global",
  "arguments": " {\"query\": \"Times Higher Education World
      ↪ University Rankings 2025 top 100 list\", \"count\": 3, \"
      ↪ summary_type\": \"short\"}",
  "tool_call_id": "call_tkpqo3p0nq4utkv0koej8r6o"
}

{
```

```
    "tool_name": "search_global",
    "arguments": " {\"query\": \"Stanford University official website
        ↪ \", \"count\": 1, \"summary_type\": \"short\"}",
    "tool_call_id": "call_m7831weg4ied8s529o3wt3tj"
}

{
    "tool_name": "search_global",
    "arguments": " {\"query\": \"MIT official website\", \"count\":
        ↪ 1, \"summary_type\": \"short\"}",
    "tool_call_id": "call_w3299d1l03rweromhhhuo490"
}

{
    "tool_name": "search_global",
    "arguments": " {\"query\": \"UC Berkeley official website\", \"
        ↪ count\": 1, \"summary_type\": \"short\"}",
    "tool_call_id": "call_10v1kyhaqftgh95ve8f22ult"
}
```

**Tool Response**

```
[index] 0
[siteName] qs.com
...
(search content omitted due to length limit)
```

| **Step 10** |
| --- |

**Reasoning Content**

I now need to continue analyzing the user's query and existing search results to determine the next steps. The user requires the top five universities in each of the five broad subjects from the QS World University Rankings by Subject 2025, along with their rankings in the QS World University Rankings 2025 and Times Higher Education World University Rankings 2025, as well as official website links, regular decision application deadlines, and application fees. The results should be organized into a Markdown table.

First, reviewing previous search results, the five broad subjects have been confirmed: Arts and Humanities, Engineering and Technology, Life Sciences and Medicine, Natural Sciences, and Social Sciences and Management. Through the text_browser_view tool, the top five universities for each subject have been obtained. For example, Arts and Humanities includes Harvard University, University of Oxford, University of Cambridge, Stanford University, and UC Berkeley.

Next, the QS World University Rankings 2025 and Times Higher Education rankings for these universities need to be determined. Previous searches for QS rankings found the top five: MIT, Imperial College London, University of Oxford, Harvard University, and University of Cambridge. However, other universities like Stanford and UC Berkeley are not included in this top five, so their QS rankings must be confirmed. Similarly, the Times Higher Education rankings lack complete top 100 data, making it impossible to determine the rankings for all required universities.

Regarding official websites, some universities' official sites have been found, such as Harvard (harvard.edu), University of Oxford (ox.ac.uk), University of Cambridge (cam.ac.uk), Stanford (stanford.edu), and MIT (mit.edu). However, links for UC Berkeley and ETH Zurich may require verification.

Application deadlines and fees are entirely missing from current search results. The user specifically requires regular decision deadlines and fees without waivers, which must be searched for each university individually.

Next steps:

1. **Confirm QS 2025 rankings for all 25 universities (5 subjects × 5 universities)**. Existing results only cover top five QS rankings, so additional searches are needed for universities like Stanford, UC Berkeley, ETH Zurich, etc.

2. **Confirm Times Higher Education 2025 rankings for all 25 universities**. Current results lack sufficient data, requiring more precise searches, such as using each university's

name as a keyword.

3. **Verify official website links** for all universities to ensure accuracy, particularly for institutions like UC Berkeley and ETH Zurich.

4. **Search for regular decision application deadlines and fees** for each university, as this information is entirely missing and critical for the user.

Potential challenges:

- Incomplete QS and Times rankings may require multiple searches or accessing official ranking lists.

- Application deadlines and fees may vary by program and could be scattered across university admissions pages, necessitating precise searches for each institution.

For example, to find Stanford's QS 2025 ranking, search "Stanford University QS World University Rankings 2025." Similarly, search "Stanford University Times Higher Education World University Rankings 2025" for Times rankings. For deadlines and fees, use queries like "Stanford University undergraduate regular decision deadline 2025" and "Stanford University application fee undergraduate."

Since there are 25 universities, searching each individually would be time-consuming. However, given the current lack of data, this step is unavoidable. To improve efficiency, batch searches could be attempted, such as querying multiple universities' QS rankings simultaneously, but search engines may yield incomplete results. Individual searches per university remain more reliable.

Additionally, some universities' application deadlines and fees might not yet be updated for 2025, requiring reliance on 2024 data with notes indicating the year. However, the user specified 2025 rankings, so efforts should focus on finding the latest information.

Summary of next actions:

1. For each of the 25 universities, search their QS 2025 and Times Higher Education 2025 rankings.

2. Verify official website links for each university.

3. Search for regular decision application deadlines and fees for each university.

4. Organize all collected data into the required Markdown table format.

Given the complexity, it may be necessary to prioritize completing rankings first, followed by deadlines and fees, as rankings are foundational to the table structure. If some data cannot be found, note it as "Not Available" or similar, but the user likely expects comprehensive information, so maximum efforts should be made to find all required details.

**Tool Call**

```
{
  "tool_name": "search_global",
  "arguments": " {\"query\": \"Stanford University QS World
      ↪ University Rankings 2025\", \"count\": 1, \"summary_type\":
      ↪  \"short\"}",
  "tool_call_id": "call_uybs2c334p8m68f0qn1bz2g3"
}

{
  "tool_name": "search_global",
  "arguments": " {\"query\": \"UC Berkeley QS World University
      ↪ Rankings 2025\", \"count\": 1, \"summary_type\": \"short
      ↪ \"}",
  "tool_call_id": "call_ejauvh2s0s7utccme0ojnk9z"
}

{
  "tool_name": "search_global",
  "arguments": " {\"query\": \"ETH Zurich QS World University
      ↪ Rankings 2025\", \"count\": 1, \"summary_type\": \"short
      ↪ \"}",
  "tool_call_id": "call_hw4sfns9euxb8myvfy1pyb65"
}

{
```

```
  "tool_name": "search_global",
  "arguments": " {\"query\": \"Johns Hopkins University QS World
      ↪ University Rankings 2025\", \"count\": 1, \"summary_type\":
      ↪  \"short\"}",
  "tool_call_id": "call_ful9nsoaobi7icgy99spwv35"
}

{
  "tool_name": "search_global",
  "arguments": " {\"query\": \"Harvard University Times Higher
      ↪ Education World University Rankings 2025\", \"count\": 1,
      ↪ \"summary_type\": \"short\"}",
  "tool_call_id": "call_pnrv601zcc1zgtep50jhj9fn"
}
```

**Tool Response**

```
[index] 0
[siteName] qs.com
...
(search content omitted due to length limit)
```

| **Step 11** |
| --- |

**Reasoning Content**

I now need to continue analyzing the user's question to determine whether further searches are necessary. The user requires the top five universities in each of the five broad subject areas from the QS World University Rankings by Subject 2025, along with their QS World University Rankings 2025 and Times Higher Education World University Rankings 2025 positions, official website homepage, regular decision application deadline, and application fee (excluding fee waivers). These should be organized into a Markdown table.

First, based on previous searches, I have obtained the top five universities for each of the five broad subjects from the QS by Subject 2025 rankings. For example:
- Arts and Humanities: Harvard, Oxford, Cambridge, Stanford, UC Berkeley
- Engineering and Technology: MIT, Oxford, Stanford, Cambridge, ETH Zurich
- Life Sciences and Medicine: Harvard, Oxford, Johns Hopkins, Stanford, MIT
- Natural Sciences: Harvard, MIT, Oxford, Cambridge, Stanford
- Social Sciences and Management: Harvard, Oxford, Stanford, MIT, Cambridge

Next, I need to confirm the QS World University Rankings 2025 and Times Higher Education (THE) 2025 rankings for each of these universities. Previous searches indicate that the top five QS World University Rankings 2025 are MIT (1), Imperial College London (2), Oxford (3), Harvard (4), and Cambridge (5). However, other universities like Stanford, UC Berkeley, ETH Zurich, and Johns Hopkins are not in the top five, so their specific rankings need to be found. For example, in the latest search for Stanford's QS ranking, the results did not directly provide the 2025 data, requiring further confirmation.

Regarding THE rankings, previous search results were fragmented. For instance, Harvard is mentioned in THE's reputation rankings but not its overall ranking. Other universities' THE rankings remain unclear, necessitating targeted searches for each institution's THE 2025 ranking.

Additionally, the official websites of some universities have been confirmed:
- Harvard: harvard.edu (from search results)
- Oxford: ox.ac.uk
- Cambridge: cam.ac.uk
- Stanford: stanford.edu
- MIT: mit.edu
- UC Berkeley: berkeley.edu
- ETH Zurich: ethz.ch (to be confirmed)
- Johns Hopkins: jhu.edu (to be confirmed)

For application deadlines and fees, this information is not yet available and requires searches for each university's undergraduate regular decision deadline and application fee for 2025. For example, Harvard's deadline is typically January 1st, with a fee around $85, but 2025

specifics need verification.

Current missing information includes:

1. QS 2025 overall rankings for Stanford, UC Berkeley, ETH Zurich, Johns Hopkins.

2. THE 2025 overall rankings for all universities.

3. Official websites for ETH Zurich and Johns Hopkins.

4. Regular decision application deadlines and fees for each university.

Further searches are needed for each missing piece of information. For example:

- Search "[University Name] QS World University Rankings 2025" for each institution.

- Search "[University Name] Times Higher Education World University Rankings 2025".

- Confirm official websites for ETH Zurich and Johns Hopkins.

- Search "[University Name] undergraduate regular decision deadline 2025" and "[University Name] application fee 2025".

Given the extensive information required, parallel searches may be necessary to improve efficiency. For example, searching for QS and THE rankings for multiple universities simultaneously, followed by application deadlines and fees.

Potential challenges include inconsistent ranking data across sources or delayed publication of 2025 THE rankings. If official data cannot be found, it may be necessary to use the latest available information or note that data is pending release.

In summary, further searches are required for each university's QS and THE rankings, official websites (for remaining institutions), application deadlines, and fees. This step will involve multiple parallel searches to gather all necessary information for the table.

**Tool Call**

```
{
  "tool_name": "search_global",
  "arguments": " {\"query\": \"Stanford University QS 2025 ranking
      ↪ \", \"count\": 1, \"summary_type\": \"short\"}",
  "tool_call_id": "call_kp7wty6g8lluj0p3cqasipf9"
}

{
  "tool_name": "search_global",
  "arguments": " {\"query\": \"UC Berkeley QS 2025 ranking\", \"
      ↪ count\": 1, \"summary_type\": \"short\"}",
  "tool_call_id": "call_39wb2xvbg6xwdri4ymrw1azi"
}

{
  "tool_name": "search_global",
  "arguments": " {\"query\": \"ETH Zurich QS 2025 ranking\", \"
      ↪ count\": 1, \"summary_type\": \"short\"}",
  "tool_call_id": "call_2unbgy4tdac9tdue86rvqdk4"
}

{
  "tool_name": "search_global",
  "arguments": " {\"query\": \"Johns Hopkins University QS 2025
      ↪ ranking\", \"count\": 1, \"summary_type\": \"short\"}",
  "tool_call_id": "call_gbm9dntn5ozb6dka35i5m46a"
}

{
  "tool_name": "search_global",
  "arguments": " {\"query\": \"Harvard University THE 2025 ranking
      ↪ \", \"count\": 1, \"summary_type\": \"short\"}",
  "tool_call_id": "call_rvvp1fnq65wlw9a1rop3kt5k"
}
```

**Tool Response**

```
[index] 0
```

```
[siteName] qs.com
...
(search content omitted due to length limit)
```

**Step 12**

**Reasoning Content**

I now need to continue analyzing the user's question and the existing search results to determine the next steps. The user wants the top five universities in each of the five broad subjects from the QS World University Rankings by Subject 2025, along with their QS and THE rankings, official websites, application deadlines, and fees. The results should be organized into a Markdown table.

First, from previous searches, the five broad subjects have been confirmed: Arts and Humanities, Engineering and Technology, Life Sciences and Medicine, Natural Sciences, and Social Sciences and Management. The top five universities for each subject were extracted from the QS official website link using the text_browser_view tool, which is reliable. For example, Arts and Humanities includes Harvard, Oxford, Cambridge, Stanford, and UC Berkeley.

Next, the QS World University Rankings 2025 and THE World University Rankings 2025 for these universities are required. Previous searches found the top five QS rankings: MIT (1), Imperial College London (2), Oxford (3), Harvard (4), and Cambridge (5). However, the top five universities in each subject may not all be within the top five overall. For instance, Stanford and UC Berkeley are not listed here, requiring their specific QS rankings. Similarly, THE rankings are incomplete; only Harvard, MIT, and Oxford's reputational rankings were found, but not their overall THE rankings.

Additionally, official websites for some universities have been identified, such as Harvard (harvard.edu), Oxford (ox.ac.uk), Cambridge (cam.ac.uk), Stanford (stanford.edu), and UC Berkeley (berkeley.edu). However, websites for ETH Zurich, Johns Hopkins University, etc., still need confirmation.

Regarding application deadlines and fees, no information has been collected yet. These details are typically found on university admissions pages, requiring further searches for each university's regular decision deadline and application fee (excluding fee waivers).

Current missing information includes:
1. QS 2025 overall rankings for universities beyond the top five (e.g., Stanford, UC Berkeley, ETH Zurich, Johns Hopkins).
2. THE 2025 overall rankings for all listed universities.
3. Official websites for universities like ETH Zurich and Johns Hopkins.
4. Application deadlines and fees for each university.

Next steps:
- Search for QS 2025 rankings for Stanford, UC Berkeley, ETH Zurich, and Johns Hopkins.
- Search for THE 2025 overall rankings for each university.
- Confirm official websites for remaining universities.
- Collect application deadlines and fees for each university via their official sites or reliable sources.

Since some universities' QS rankings remain unconfirmed (e.g., Stanford), further targeted searches are needed. For THE rankings, as previous searches yielded limited results, more precise queries (e.g., "Harvard University THE World University Rankings 2025") may be required. Application deadlines and fees will likely necessitate visiting each university's admissions page, which could be time-consuming but critical for completing the table.

In summary, additional searches are required for QS and THE rankings, official websites, deadlines, and fees to ensure the table is comprehensive and accurate.

**Tool Call**

```
{
  "tool_name": "search_global",
  "arguments": "{\"query\":\"Stanford University QS 2025 overall
      ↪ ranking\",\"count\":1,\"summary_type\":\"short\"}",
  "tool_call_id": "call_m0wy33ya1qmtcnn0yy7easag"
}
```

```
{
  "tool_name": "search_global",
  "arguments": "{\"query\":\"UC Berkeley QS 2025 overall ranking
      ↪ \",\"count\":1,\"summary_type\":\"short\"}",
  "tool_call_id": "call_0h4oc2c1toyde96u4h9wwo8w"
}

{
  "tool_name": "search_global",
  "arguments": "{\"query\":\"ETH Zurich THE 2025 ranking\",\"count
      ↪ \":1,\"summary_type\":\"short\"}",
  "tool_call_id": "call_4tntudkekzis92mca01dqn9o"
}

{
  "tool_name": "search_global",
  "arguments": "{\"query\":\"Johns Hopkins University official
      ↪ website\",\"count\":1,\"summary_type\":\"short\"}",
  "tool_call_id": "call_esx3y7cxo36c0jfhkaiilkzx"
}

{
  "tool_name": "search_global",
  "arguments": "{\"query\":\"Harvard University 2025 regular
      ↪ decision deadline\",\"count\":1,\"summary_type\":\"short
      ↪ \"}",
  "tool_call_id": "call_k494jq5irmh3f325cmp2hw5y"
}
```

**Tool Response**

```
[index] 0
[siteName] qs.com
...
(search content omitted due to length limit)
```

**Step 13**

**Reasoning Content**

I now need to continue analyzing the user's question and determine the next steps. The user wants the top five universities in each of the five broad subjects from the QS World University Rankings by Subject 2025, along with their QS and THE overall rankings, official website, regular decision deadline, and application fee, organized into a table.

First, the previous searches have identified the five broad subjects: Arts and Humanities, Engineering and Technology, Life Sciences and Medicine, Natural Sciences, and Social Sciences and Management. The top five universities for each subject were obtained by accessing the QS official website link (via text_browser_view), which is reliable. For example:
- Arts and Humanities: Harvard, Oxford, Cambridge, Stanford, UC Berkeley
- Engineering and Technology: MIT, Oxford, Stanford, Cambridge, ETH Zurich
- Life Sciences and Medicine: Harvard, Oxford, Johns Hopkins, Stanford, MIT
- Natural Sciences: Harvard, MIT, Oxford, Cambridge, Stanford
- Social Sciences and Management: Harvard, Oxford, Stanford, MIT, Cambridge

Next, the QS and THE overall rankings for these universities are needed. Previous searches for the QS 2025 overall rankings only found the top five (MIT, Imperial College London, Oxford, Harvard, Cambridge), but most of the top five universities in the subject rankings (e.g., Stanford, UC Berkeley, ETH Zurich, Johns Hopkins) are missing their QS overall rankings. Similarly, THE rankings lack data for most universities.

For example, Stanford University is a top-five university in multiple subjects, but its QS 2025 overall ranking is unknown. Previous searches using "Stanford University QS 2025 ranking" yielded no results. The same applies to UC Berkeley and ETH Zurich. Johns Hopkins' QS ranking is also unclear. Additionally, THE rankings for Harvard, MIT, Oxford, etc., remain unspecified.

Regarding official websites, some universities have been confirmed:
- Harvard: harvard.edu
- Oxford: ox.ac.uk
- Cambridge: cam.ac.uk
- Stanford: stanford.edu
- MIT: mit.edu
- UC Berkeley: berkeley.edu
- Johns Hopkins: jhu.edu (from search results)
- ETH Zurich: ethz.ch (to be confirmed)
Application deadlines and fees are mostly missing. Only Harvard's regular decision deadline (January 1) was found via admissionsight.com, but this requires verification. Other universities' deadlines and fees are entirely unknown and need separate searches.
Current missing information includes:
1. QS 2025 overall rankings for Stanford, UC Berkeley, ETH Zurich, Johns Hopkins, etc.
2. THE 2025 overall rankings for all 25 universities (five subjects × five universities each).
3. Official websites for ETH Zurich (needs confirmation).
4. Regular decision deadlines and application fees for all 25 universities.
Next steps:
- Continue searching for QS 2025 overall rankings for each university (e.g., "Stanford University QS World University Rankings 2025").
- Search for THE 2025 overall rankings for each university (e.g., "Harvard University Times Higher Education World University Rankings 2025").
- Confirm ETH Zurich's official website.
- Search for each university's regular decision deadline and application fee (e.g., "Stanford University 2025 regular decision deadline" and "Stanford University application fee 2025").
Given the large number of universities, prioritizing key institutions or using batch searches may be necessary. However, due to tool limitations, individual searches are required. For example:
- Search for Stanford's QS and THE rankings, deadline, and fee.
- Repeat for UC Berkeley, ETH Zurich, Johns Hopkins, etc.
Additionally, some universities might have similar deadlines (e.g., many U.S. universities use January 1), but verification is still needed. Application fees vary by institution and may require official website confirmation.
In summary, further searches are required for rankings, deadlines, and fees for each university. This will involve multiple tool calls, potentially exceeding the current step limit, but given the user's request, it is necessary to proceed gradually.
**Tool Call**

```
{
  "tool_name": "search_global",
  "arguments": " {\"query\": \"Stanford University QS 2025 overall
      ↪ ranking\", \"count\": 1, \"summary_type\": \"short\"}",
  "tool_call_id": "call_ejdq3ihefg64a7atlycbh4rn"
}

{
  "tool_name": "search_global",
  "arguments": " {\"query\": \"UC Berkeley QS 2025 overall ranking
      ↪ \", \"count\": 1, \"summary_type\": \"short\"}",
  "tool_call_id": "call_z4u6fms4gph9r055ejqadlze"
}

{
  "tool_name": "search_global",
  "arguments": " {\"query\": \"ETH Zurich THE 2025 ranking\", \"
      ↪ count\": 1, \"summary_type\": \"short\"}",
  "tool_call_id": "call_xphp7bot7dzeivh7zcbzgaj6"
}
```

```
{
  "tool_name": "search_global",
  "arguments": " {\"query\": \"Johns Hopkins University THE 2025
      ↪ ranking\", \"count\": 1, \"summary_type\": \"short\"}",
  "tool_call_id": "call_1v1owztt49s08uv7cb0k6lmu"
}

{
  "tool_name": "search_global",
  "arguments": " {\"query\": \"Stanford University 2025 regular
      ↪ decision deadline\", \"count\": 1, \"summary_type\": \"
      ↪ short\"}",
  "tool_call_id": "call_olwmhxpmmze67gl38mohlnde"
}
```

**Tool Response**

```
[index] 0
[siteName] qs.com
...
(search content omitted due to length limit)
```

| Step 14 |
|---|

**Reasoning Content**

I now need to continue analyzing the user's query to determine if further searches are required to complete the response. The user needs the top five universities in each of the five major subject areas from the QS World University Rankings by Subject 2025, along with their QS overall ranking, THE ranking, official website, regular decision deadline, and application fee, all organized into a table.

First, reviewing previous search results: The five major subject areas have been confirmed as Arts and Humanities, Engineering and Technology, Life Sciences and Medicine, Natural Sciences, and Social Sciences and Management. Through multiple searches, the top five universities in each subject area were obtained from the official QS article, including Harvard, Oxford, Cambridge, Stanford, UC Berkeley, MIT, ETH Zurich, Johns Hopkins, etc. These names are clear.

Next, regarding the QS overall ranking and THE ranking: The QS 2025 overall ranking's top five include MIT, Imperial College London, Oxford, Harvard, and Cambridge. However, other universities like Stanford and UC Berkeley lack specific rankings. Previous attempts to search for their QS rankings yielded no direct results. Similarly, THE rankings for most universities remain unclear, with only Harvard mentioned as possibly No. 1, and Johns Hopkins ranked 16th. Other universities like Stanford, MIT, and Oxford lack THE ranking data.

Regarding official websites: Some universities' official sites have been found, such as Harvard (harvard.edu), Oxford (ox.ac.uk), Cambridge (cam.ac.uk), Stanford (stanford.edu), MIT (mit.edu), UC Berkeley (berkeley.edu), and Johns Hopkins (jhu.edu). However, ETH Zurich's official website still needs confirmation.

Application deadlines and fees: Only Harvard's regular decision deadline (January 1) and Stanford's deadline (around January 6) have been partially identified. Other universities' deadlines and fee information are completely missing, including Oxford, Cambridge, MIT, UC Berkeley, ETH Zurich, Johns Hopkins, etc.

Current missing information includes:

1. QS overall rankings for Stanford, UC Berkeley, ETH Zurich, and Johns Hopkins.

2. THE rankings for all universities except Harvard and Johns Hopkins.

3. Official website for ETH Zurich.

4. Regular decision deadlines and application fees for all universities.

Next steps:

- For QS rankings: Continue searching for Stanford, UC Berkeley, ETH Zurich, and Johns Hopkins' QS 2025 overall rankings. If no results, consider whether the rankings were not released or if search terms need adjustment.

- For THE rankings: Search individually for each university's THE 2025 ranking, e.g., "MIT THE 2025 ranking."

- ETH Zurich's official website: Directly search for "ETH Zurich official website."
- Deadlines and fees: For each university, search for "(University Name) 2025 regular decision deadline" and "(University Name) undergraduate application fee 2025."

Given the extensive missing information, particularly deadlines and fees requiring university-specific searches, multiple additional searches are needed. However, considering previous repeated searches for QS and THE rankings yielded limited results, it may be necessary to use more precise keywords or accept potential data unavailability. For deadlines and fees, since these are critical application details, further searches are essential even if time-consuming.

**Tool Call**

```
{
  "tool_name": "search_global",
  "arguments": " {\"query\": \"Stanford University QS 2025 overall
      ↪ ranking\", \"count\": 1, \"summary_type\": \"short\"}",
  "tool_call_id": "call_9r8byy6nyahid0l2gzyxmlmp"
}

{
  "tool_name": "search_global",
  "arguments": " {\"query\": \"UC Berkeley QS 2025 overall ranking
      ↪ \", \"count\": 1, \"summary_type\": \"short\"}",
  "tool_call_id": "call_b750zbm6c46mh7yfdno028aa"
}

{
  "tool_name": "search_global",
  "arguments": " {\"query\": \"ETH Zurich official website\", \"
      ↪ count\": 1, \"summary_type\": \"short\"}",
  "tool_call_id": "call_1fdd57iqkeq1e5kekp937ni2"
}

{
  "tool_name": "search_global",
  "arguments": " {\"query\": \"MIT THE World University Rankings
      ↪ 2025\", \"count\": 1, \"summary_type\": \"short\"}",
  "tool_call_id": "call_duakbexa02726fbquyh8twpy"
}

{
  "tool_name": "search_global",
  "arguments": " {\"query\": \"University of Oxford 2025 regular
      ↪ decision deadline\", \"count\": 1, \"summary_type\": \"
      ↪ short\"}",
  "tool_call_id": "call_fdz6k3ltspvyjeox1tipaju6"
}
```

**Tool Response**

```
[index] 0
[siteName] qs.com
...
(search content omitted due to length limit)
```

**Step 15**

**Reasoning Content**

I now need to continue analyzing the user's query and determine the next steps. The user wants the top five universities in each of the five broad subject areas from the QS World University Rankings by Subject 2025, as well as their rankings in the QS and THE World University Rankings 2025, official website, regular decision deadline, and application fee, organized into a table.

First, I have obtained the top five universities in each of the five broad subject areas from the

QS website, which is correct. Next, I need to confirm their overall rankings in QS and THE. However, previous searches for QS and THE rankings have yielded incomplete information. For example, the QS 2025 overall rankings only provided the top five, while THE rankings lack specific data for many universities.

Subsequent searches focused on individual universities' rankings, such as Stanford and UC Berkeley, but results remain unclear. For instance, Stanford's QS overall ranking is still unspecified, and THE rankings for ETH Zurich and MIT are missing. Additionally, information about application deadlines and fees for some universities, like UC Berkeley and ETH Zurich, has not been found.

Next, I need to:

1. Confirm the QS 2025 overall rankings for Stanford and UC Berkeley.
2. Find THE 2025 rankings for ETH Zurich, MIT, and UC Berkeley.
3. Obtain official website links for all universities (some are already confirmed, but ETH Zurich's needs verification).
4. Collect regular decision deadlines and application fees for each university, such as UC Berkeley, ETH Zurich, etc.

For example, the user has already confirmed Harvard's deadline as January 1, but Stanford's deadline is mentioned as resuming on January 6, 2025, which may require clarification on the specific application deadline date. Similarly, Oxford's deadline is October 15, but this might be for undergraduate applications and needs confirmation.

In summary, further searches are needed for:

- QS overall rankings for Stanford and UC Berkeley.
- THE rankings for ETH Zurich, MIT, UC Berkeley, etc.
- Application deadlines and fees for UC Berkeley, ETH Zurich, Johns Hopkins, etc.
- Official website links for ETH Zurich and Johns Hopkins (partially found but needing confirmation).

Thus, additional searches should target these missing pieces of information to ensure the table is complete and accurate.

**Tool Call**

```
{
  "tool_name": "search_global",
  "arguments": " {\"query\": \"Stanford University QS 2025 overall
      ↪ ranking\", \"count\": 1, \"summary_type\": \"short\"}",
  "tool_call_id": "call_6d1cfgv75mby78799l42yg29"
}

{
  "tool_name": "search_global",
  "arguments": " {\"query\": \"UC Berkeley QS 2025 overall ranking
      ↪ \", \"count\": 1, \"summary_type\": \"short\"}",
  "tool_call_id": "call_fk0ygfzx6o213maqtu3n06fw"
}

{
  "tool_name": "search_global",
  "arguments": " {\"query\": \"ETH Zurich THE 2025 ranking\", \"
      ↪ count\": 1, \"summary_type\": \"short\"}",
  "tool_call_id": "call_xcf142eolfb2fgwrs0z92kze"
}

{
  "tool_name": "search_global",
  "arguments": " {\"query\": \"UC Berkeley 2025 regular decision
      ↪ deadline\", \"count\": 1, \"summary_type\": \"short\"}",
  "tool_call_id": "call_b46om2dy07o6r8rpwycz9rv9"
}

{
  "tool_name": "search_global",
```

```
  "arguments": " {\"query\": \"MIT 2025 application fee\", \"count
      ↪ \": 1, \"summary_type\": \"short\"}",
  "tool_call_id": "call_2revjkshvu6vg7fec3hawqxf"
}
```

**Tool Response**

```
[index] 0
[siteName] qs.com
...
(search content omitted due to length limit)
```



**Step 16**



**Reasoning Content**

I now need to continue analyzing the user's query and determine the next steps. The user wants the top five universities in each of the five broad subjects from the QS World University Rankings by Subject 2025, along with their positions in the overall QS and THE rankings, official website, regular decision deadline, and application fee. The previous search results have covered some information, but several key points remain incomplete.

First, regarding the QS subject rankings, the top five universities in each subject were already obtained through the text_browser_view tool, which should be accurate. Next, the overall QS and THE rankings for these universities are still missing for some institutions. For example, Stanford University's QS overall ranking and UC Berkeley's QS overall ranking have not been confirmed despite multiple searches. Similarly, ETH Zurich's THE ranking and MIT's THE ranking are unclear.

Second, regarding the official websites, most have been found, such as Harvard, Oxford, Cambridge, Stanford, MIT, UC Berkeley, and Johns Hopkins, but ETH Zurich's official website was confirmed in the latest search result as www.ethz.ch.

Third, application deadlines and fees: Harvard's regular decision deadline is January 1, Stanford's deadline might be in early January (but needs confirmation), UC Berkeley's deadline is December 2, 2024 (for Fall 2025), Oxford's deadline is October 15, and MIT's application fee was mentioned in the search results as requiring further verification. Additionally, deadlines and fees for other universities like ETH Zurich and Johns Hopkins are still missing.

Current missing information includes:

1. QS overall rankings for Stanford University and UC Berkeley.

2. THE rankings for MIT, Stanford, UC Berkeley, ETH Zurich, and University of Cambridge.

3. Application deadlines for ETH Zurich, Johns Hopkins University, and University of Cambridge.

4. Application fees for all universities (only MIT's was partially mentioned but not clear).

Next steps should focus on filling these gaps. For example, search for Stanford's QS overall ranking, UC Berkeley's QS ranking, MIT's THE ranking, ETH Zurich's THE ranking, application deadlines for each university, and specific application fees. However, previous searches for Stanford and UC Berkeley's QS rankings have repeatedly yielded no results, possibly due to unavailable data or search limitations. In such cases, it might be necessary to use approximate values or indicate "Not Available," but the user requires accurate data, so further attempts with different keywords may be needed, such as "Stanford University QS World University Rankings 2025 position" or checking official QS reports.

For application fees, each university's official website should be checked, but since website access isn't possible, search queries like "Harvard University 2025 undergraduate application fee" would be needed. Similarly, deadlines for other universities need verification through searches.

Given the time-consuming nature and potential lack of results, prioritizing critical missing information is essential. For instance, confirm THE rankings for key universities, application deadlines, and fees. If searches continue to return no results, note the missing data and proceed with available information.

**Tool Call**

```
{
  "tool_name": "search_global",
```

```
  "arguments": " {\"query\": \"Stanford University QS 2025 overall
      ↪ ranking official\", \"count\": 1, \"summary_type\": \"short
      ↪ \"}",
  "tool_call_id": "call_zyddtf7vxlo03lxbjxn6uqzn"
}

{
  "tool_name": "search_global",
  "arguments": " {\"query\": \"UC Berkeley QS 2025 overall ranking
      ↪ official\", \"count\": 1, \"summary_type\": \"short\"}",
  "tool_call_id": "call_6rkemr2gk44i2brf3pz5nwq4"
}

{
  "tool_name": "search_global",
  "arguments": " {\"query\": \"MIT Times Higher Education 2025
      ↪ ranking\", \"count\": 1, \"summary_type\": \"short\"}",
  "tool_call_id": "call_gbvv78o7wn4oosb37co73hbz"
}

{
  "tool_name": "search_global",
  "arguments": " {\"query\": \"ETH Zurich 2025 undergraduate
      ↪ application deadline\", \"count\": 1, \"summary_type\": \"
      ↪ short\"}",
  "tool_call_id": "call_4wxkg8mb280cutabmmv7fhe3"
}

{
  "tool_name": "search_global",
  "arguments": " {\"query\": \"Harvard University 2025 application
      ↪ fee\", \"count\": 1, \"summary_type\": \"short\"}",
  "tool_call_id": "call_f3x86j4o03v6cz28fwgu74sk"
}
```

**Tool Response**

```
[index] 0
[siteName] qs.com
...
(search content omitted due to length limit)
```

---
**Step 17**
---

**Reasoning Content**

I now need to continue analyzing the user's question and determine if further searches are required to complete the response. The user needs the top five universities in each of the five broad subjects from the QS World University Rankings by Subject 2025, along with their positions in the overall QS and THE rankings, official website, regular decision deadline, and application fee (excluding fee waivers). The results should be organized into a Markdown table.

First, based on previous searches, I have successfully identified the top five universities in each of the five broad subjects (Arts and Humanities, Engineering and Technology, Life Sciences and Medicine, Natural Sciences, Social Sciences and Management). This information was obtained from the official QS website link, which is accurate. For example, Arts and Humanities are led by Harvard, Oxford, Cambridge, Stanford, and UC Berkeley; Engineering and Technology are led by MIT, Oxford, Stanford, Cambridge, and ETH Zurich, etc.

Next, regarding the overall QS and THE rankings for these universities, some information has been collected, but several gaps remain. For instance, the QS overall rankings for Stanford and UC Berkeley have not been confirmed, and THE rankings for ETH Zurich and UC Berkeley are missing. Previous searches attempted to find this information, but the results were inconsistent or incomplete. For example, Stanford's QS ranking searches yielded only

score information without specific rankings, while ETH Zurich's THE ranking results only mentioned ranges like 251-300, which are not precise enough.

Regarding official websites, most have been confirmed: Harvard (harvard.edu), Oxford (ox.ac.uk), Cambridge (cam.ac.uk), Stanford (stanford.edu), MIT (mit.edu), UC Berkeley (berkeley.edu), ETH Zurich (ethz.ch), Johns Hopkins (jhu.edu), etc. These should be accurate.

For application deadlines and fees, some information is available, such as Harvard's regular decision deadline of January 1 and application fee of $85, Stanford's deadline around January 6, UC Berkeley's deadline of December 2, etc. However, deadlines and fees for ETH Zurich, Johns Hopkins, and MIT are still unclear. For example, MIT's application fee was not found in search results, and ETH Zurich's undergraduate deadline is missing.

Current missing information includes:

1. QS overall rankings for Stanford and UC Berkeley.

2. THE rankings for Stanford, UC Berkeley, ETH Zurich, and MIT.

3. Application deadlines and fees for ETH Zurich, Johns Hopkins, MIT, and Oxford.

4. Confirmation of application fees for some universities (e.g., Stanford, UC Berkeley).

Previous attempts to search for QS and THE rankings yielded limited results, possibly due to the specificity of keywords. For example, searching for "Stanford University QS 2025 overall ranking" might need more precise terms or official sources. Similarly, THE rankings may require direct access to official THE reports or reliable education platforms.

Regarding application deadlines and fees, some universities might only list graduate information, requiring undergraduate-specific searches. For example, ETH Zurich's deadline might be found using keywords like "ETH Zurich undergraduate application deadline 2025."

Is further searching necessary? Yes. Specific missing points must be addressed:

- Confirm QS overall rankings for Stanford and UC Berkeley.

- Confirm THE rankings for key universities.

- Find undergraduate deadlines and fees for ETH Zurich, Johns Hopkins, MIT, and Oxford.

However, multiple previous searches for QS and THE rankings have failed to yield results, suggesting possible data unavailability or keyword issues. For example, THE's 2025 rankings might not be fully released, or official lists are not publicly accessible. In such cases, it may be necessary to indicate "Not Available" in the table, but the user requires accurate data, so further attempts are warranted.

Alternative search strategies could include using different keywords, such as "QS World University Rankings 2025 top 20" or checking university official websites for their own rankings. For example, MIT's official website might mention their THE ranking.

For application fees and deadlines, searching university admissions pages directly might help. For example, MIT's undergraduate admissions page should list the application fee and deadline.

In summary, further searches are needed for the missing rankings, deadlines, and fees. If these searches continue to yield no results, the final table may need to mark some fields as "Not Available," but every effort should be made to find accurate data first.

**Tool Call**

```
{
  "tool_name": "search_global",
  "arguments": " {\"query\": \"Stanford University QS 2025 ranking
      ↪ official\", \"count\": 1, \"summary_type\": \"short\"}",
  "tool_call_id": "call_4vg6cppw9u7xo1n88i54r8s7"
}

{
  "tool_name": "search_global",
  "arguments": " {\"query\": \"UC Berkeley THE 2025 ranking\", \"
      ↪ count\": 1, \"summary_type\": \"short\"}",
  "tool_call_id": "call_c0lqxg01n1evvcsbqop2ei2p"
}

{
  "tool_name": "search_global",
```

```
  "arguments": " {\"query\": \"ETH Zurich undergraduate application
      ↪  deadline 2025\", \"count\": 1, \"summary_type\": \"short
      ↪ \"}",
  "tool_call_id": "call_j5hc7amzhoo5zywr9513mjzi"
}

{
  "tool_name": "search_global",
  "arguments": " {\"query\": \"MIT undergraduate application fee
      ↪ 2025\", \"count\": 1, \"summary_type\": \"short\"}",
  "tool_call_id": "call_pk6esldo7dcwlr6kk9kir4zw"
}

{
  "tool_name": "search_global",
  "arguments": " {\"query\": \"University of Oxford undergraduate
      ↪ application fee 2025\", \"count\": 1, \"summary_type\": \"
      ↪ short\"}",
  "tool_call_id": "call_fjkq2wyfk8r130cfexq71l43"
}
```

**Tool Response**

```
[index] 0
[siteName] qs.com
...
(search content omitted due to length limit)
```

**Step 18**

**Reasoning Content**

I now need to continue analyzing the user's question and determine the next steps. The user requires the top five universities in each of the five broad subject areas from the QS World University Rankings by Subject 2025, along with their overall rankings in the QS and THE rankings, official website, regular decision deadline, and application fee. The previous search process has collected some information, but there are still missing details that need to be supplemented.

First, reviewing the search history, the five broad subject areas have been confirmed as Arts and Humanities, Engineering and Technology, Life Sciences and Medicine, Natural Sciences, and Social Sciences and Management. The top five universities for each subject have been obtained through the QS official website, such as Harvard, Oxford, and Cambridge for Arts and Humanities. However, some universities' overall QS and THE rankings remain unclear, such as Stanford and UC Berkeley's QS 2025 rankings, and ETH Zurich's THE ranking.

Next, regarding official websites, some universities' URLs have been confirmed, like Harvard's harvard.edu, Oxford's ox.ac.uk, etc., but Johns Hopkins University's official website might still need verification. Application deadlines and fees are missing for most universities, such as ETH Zurich's deadline and UC Berkeley's application fee.

Additionally, some universities' THE rankings are still unconfirmed, such as MIT and UC Berkeley. The regular decision deadlines and application fees for most universities are also incomplete, requiring further searches.

Given the repeated failure to find Stanford and UC Berkeley's QS rankings in previous searches, it may be necessary to adjust search keywords, such as using "Stanford University QS 2025 overall ranking" or checking the official QS rankings page directly. Similarly, for THE rankings, trying more precise keywords like "ETH Zurich Times Higher Education 2025 ranking" might yield better results.

For application deadlines and fees, direct searches for each university's undergraduate admissions page, such as "ETH Zurich undergraduate application deadline 2025" or "UC Berkeley application fee 2025," could provide accurate information. Some previously found deadlines, like Harvard's January 1st and Stanford's January 6th resumption date, need confirmation for correctness and completeness.

In summary, the next steps should include:
1. Continuing to search for QS overall rankings for Stanford, UC Berkeley, ETH Zurich, and Johns Hopkins.
2. Searching for THE rankings for MIT, UC Berkeley, ETH Zurich, etc.
3. Confirming official websites for Johns Hopkins and ETH Zurich.
4. Collecting application deadlines and fees for all 25 universities (5 subjects × 5 universities each).
Since the user requires organizing results into a table, each piece of information must be accurate. Given the current gaps, further searches are necessary to fill in missing data, particularly rankings, deadlines, and fees. If certain data remains unobtainable after multiple attempts, it may be noted as "Not Available," but efforts should be made to find official sources first.

**Tool Call**

```
{
  "tool_name": "search_global",
  "arguments": " {\"query\": \"QS World University Rankings 2025
      ↪ official full list\", \"count\": 1, \"summary_type\": \"
      ↪ short\"}",
  "tool_call_id": "call_f0o9pubreunrxpya0oq4b81q"
}

{
  "tool_name": "search_global",
  "arguments": " {\"query\": \"Times Higher Education World
      ↪ University Rankings 2025 official list\", \"count\": 1, \"
      ↪ summary_type\": \"short\"}",
  "tool_call_id": "call_vwu7jn7gb20f7nau6qgxbrzx"
}

{
  "tool_name": "search_global",
  "arguments": " {\"query\": \"Johns Hopkins University official
      ↪ website\", \"count\": 1, \"summary_type\": \"short\"}",
  "tool_call_id": "call_jzaegi9emrbylvoiyqh48ei5"
}

{
  "tool_name": "search_global",
  "arguments": " {\"query\": \"ETH Zurich undergraduate application
      ↪  fee 2025\", \"count\": 1, \"summary_type\": \"short\"}",
  "tool_call_id": "call_j61xc1l9w7zgg8zsmtt9lqez"
}

{
  "tool_name": "search_global",
  "arguments": " {\"query\": \"UC Berkeley undergraduate
      ↪ application fee 2025\", \"count\": 1, \"summary_type\": \"
      ↪ short\"}",
  "tool_call_id": "call_vj5x5vqpk1x44swehq5mkg34"
}
```

**Tool Response**

```
[index] 0
[siteName] qs.com
...
(search content omitted due to length limit)
```

**Step 19**

**Reasoning Content**

I now need to continue analyzing the user's query and the existing search results to determine whether further searches are required to complete the task. The user needs the top five universities in each of the five broad subjects from the QS World University Rankings by Subject 2025, along with their QS overall rankings, THE rankings, official websites, regular decision deadlines, and application fees. These details should be organized into a Markdown table.

First, reviewing previous search results:

1. **QS Subject Rankings**: Through multiple searches and using the text_browser_view tool, the top five universities for each of the five broad subjects have been confirmed:
- Arts and Humanities: Harvard, Oxford, Cambridge, Stanford, UC Berkeley
- Engineering and Technology: MIT, Oxford, Stanford, Cambridge, ETH Zurich
- Life Sciences and Medicine: Harvard, Oxford, Johns Hopkins, Stanford, MIT
- Natural Sciences: Harvard, MIT, Oxford, Cambridge, Stanford
- Social Sciences and Management: Harvard, Oxford, Stanford, MIT, Cambridge

2. **QS Overall Rankings**: The official QS website mentions MIT at 1st, Imperial College London at 2nd, Oxford at 3rd, Harvard at 4th, and Cambridge at 5th. However, rankings for other universities (e.g., Stanford, UC Berkeley, ETH Zurich, Johns Hopkins) remain unclear. Multiple searches for individual universities' rankings yielded no results, possibly due to incomplete or unpublicized full rankings.

3. **THE Rankings**: Results are even more limited. Only Harvard's THE ranking was mentioned but not confirmed, while others like MIT, Stanford, and Oxford lack clear data. The THE website's search results provided partial information but no complete top 100 list.

4. **Official Websites**: Most universities' official websites have been found, including Harvard, Oxford, Cambridge, Stanford, MIT, UC Berkeley, ETH Zurich, and Johns Hopkins (via graduate admissions links).

5. **Deadlines and Application Fees**:
- Deadlines: Harvard (January 1), Stanford (unspecified, but likely around January), UC Berkeley (December 2, 2024 for Fall 2025), Oxford (October 15), MIT (unspecified), ETH Zurich (varies by program, possibly January-April).
- Fees: Harvard ($85 for law school, undergraduate fee unclear), MIT ($75), UC Berkeley ($100 for bootcamp, possibly different for undergrad), ETH Zurich (CHF 150 for architecture program, general undergrad fee unclear).

**Missing Information**:
- QS overall rankings for Stanford, UC Berkeley, ETH Zurich, Johns Hopkins.
- THE rankings for nearly all universities.
- Undergraduate application deadlines and fees for some universities (e.g., ETH Zurich, Johns Hopkins, Cambridge).

**Challenges**:
- QS and THE official websites do not publicly disclose complete rankings, making it difficult to obtain accurate data for specific universities.
- Multiple searches for individual university rankings have failed, suggesting the information may not be publicly available or requires accessing paid reports.
- Application deadlines and fees vary by program and university, with some details only available on official admissions pages, which may require direct access to each university's website.

**Next Steps**:

1. **QS and THE Rankings**: Since repeated searches for full rankings or individual university rankings have failed, it may be necessary to abandon obtaining these data and note "Not Available" in the table.

2. **Deadlines and Fees**: For universities with unclear deadlines/fees (e.g., ETH Zurich, Johns Hopkins), attempt to search their official admissions pages directly or use more precise keywords (e.g., "Johns Hopkins University undergraduate application deadline 2025").

3. **Website Verification**: Confirm that all university websites are official and correct, particularly for Johns Hopkins.

**Conclusion**: Given the inability to obtain QS and THE rankings for some universities despite multiple searches, and the potential unavailability of public data, it is reasonable to conclude that further searches will not yield additional information. The remaining dead-

lines and fees can be supplemented through targeted searches, but if those also fail, they should be marked as "Not Available."

**Final Answer**

```markdown
| Subject                  | University
    ↪                                       | QS World University
    ↪ Rankings by Subject 2025 | QS World University Rankings 2025
    ↪ | Times Higher Education World University Rankings 2025 |
    ↪ Home Page                                 | Application
    ↪ Deadline      | Application Fee   |
|-------------------------|
    ↪ -------------------------------------------------------------------------
    ↪  ---------------------------------|
    ↪ ---------------------------------------------------|
    ↪ -------------------------------------------|
    ↪ -------------------------| -------------------|
| Arts and Humanities      | Harvard University
    ↪                          | 1
    ↪                                         | 4
    ↪                              | Not Available
    ↪                                      | https://www.harvard
    ↪ .edu                     | January 1               | $85
    ↪                  |
| Arts and Humanities      | University of Oxford
    ↪                          | 2
    ↪                                         | 3
    ↪                              | Not Available
    ↪                                      | https://www.ox.ac.
    ↪ uk                       | October 15              | Not
    ↪ Available     |
| Arts and Humanities      | University of Cambridge
    ↪                          | 3
    ↪                                         | 5
    ↪                              | Not Available
    ↪                                      | https://www.cam.ac.
    ↪ uk                       | October 15              | Not
    ↪ Available     |
| Arts and Humanities      | Stanford University
    ↪                          | 4
    ↪                                            | Not Available
    ↪                          | Not Available
    ↪                                         | https://www.
    ↪ stanford.edu                     | January 5
    ↪ | Not Available     |
| Arts and Humanities      | University of California, Berkeley (
    ↪ UCB)     | 5                                         | Not
    ↪ Available                 | Not Available
    ↪                                         | https://www.
    ↪ berkeley.edu                     | December 2, 2024
    ↪ | $100             |
| Engineering and Technology| Massachusetts Institute of Technology
    ↪  (MIT) | 1                                         | 1
    ↪                              | Not Available
    ↪                                      | https://web.mit.edu
    ↪                          | January 5               | $75
    ↪                  |
| Engineering and Technology| University of Oxford
    ↪                          | 2
    ↪                                         | 3
    ↪                              | Not Available
    ↪                                      | https://www.ox.ac.
```

```
    ↪ uk                    | October 15                  | Not
    ↪ Available     |
| Engineering and Technology| Stanford University
    ↪                        | 3
    ↪                                        | Not Available
    ↪                    | Not Available
    ↪                                        | https://www.
    ↪ stanford.edu            | January 5
    ↪ | Not Available     |
| Engineering and Technology| University of Cambridge
    ↪                        | 4
    ↪                                        | 5
    ↪                                | Not Available
    ↪                                        | https://www.cam.ac.
    ↪ uk                    | October 15                  | Not
    ↪ Available     |
| Life Sciences and Medicine| Harvard University
    ↪                        | 1
    ↪                                        | 4
    ↪                                | Not Available
    ↪                                        | https://www.harvard
    ↪ .edu                    | January 1                  | $85
    ↪                 |
| Life Sciences and Medicine| University of Oxford
    ↪                        | 2
    ↪                                        | 3
    ↪                                | Not Available
    ↪                                        | https://www.ox.ac.
    ↪ uk                    | October 15                  | Not
    ↪ Available     |
| Life Sciences and Medicine| Johns Hopkins University
    ↪                        | 3
    ↪                                        | Not Available
    ↪                        | 16
    ↪                                        | https://
    ↪ www.jhu.edu                        | January 3
    ↪                 | Not Available     |
| Life Sciences and Medicine| Stanford University
    ↪                        | 4
    ↪                                        | Not Available
    ↪                    | Not Available
    ↪                                        | https://www.
    ↪ stanford.edu            | January 5
    ↪ | Not Available     |
| Life Sciences and Medicine| Massachusetts Institute of Technology
    ↪   (MIT) | 5                                        | 1
    ↪                                | Not Available
    ↪                                        | https://web.mit.edu
    ↪                        | January 5                  | $75
    ↪                 |
| Natural Sciences        | Harvard University
    ↪                        | 1
    ↪                                        | 4
    ↪                                | Not Available
    ↪                                        | https://www.harvard
    ↪ .edu                    | January 1                  | $85
    ↪                 |
| Natural Sciences        | Massachusetts Institute of Technology
    ↪ (MIT) | 2                                        | 1
    ↪                                | Not Available
    ↪                                        | https://web.mit.edu
    ↪                        | January 5                  | $75
    ↪                 |
```

```
| Natural Sciences        | University of Oxford
    ↪                            | 3
    ↪                                            | 3
    ↪                                | Not Available
    ↪                                            | https://www.ox.ac.
    ↪ uk                     | October 15                | Not
    ↪ Available     |
| Natural Sciences        | University of Cambridge
    ↪                            | 4
    ↪                                            | 5
    ↪                                | Not Available
    ↪                                            | https://www.cam.ac.
    ↪ uk                     | October 15                | Not
    ↪ Available     |
| Natural Sciences        | Stanford University
    ↪                            | 5
    ↪                                            | Not Available
    ↪                        | Not Available
    ↪                                            | https://www.
    ↪ stanford.edu                 | January 5
    ↪ | Not Available     |
| Social Sciences and Management| Harvard University
    ↪                            | 1
    ↪                                            | 4
    ↪                                | Not Available
    ↪                                            | https://www.harvard
    ↪ .edu                   | January 1                 | $85
    ↪                |
| Social Sciences and Management| University of Oxford
    ↪                            | 2
    ↪                                            | 3
    ↪                                | Not Available
    ↪                                            | https://www.ox.ac.
    ↪ uk                     | October 15                | Not
    ↪ Available     |
| Social Sciences and Management| Stanford University
    ↪                            | 3
    ↪                                            | Not Available
    ↪                        | Not Available
    ↪                                            | https://www.
    ↪ stanford.edu                 | January 5
    ↪ | Not Available     |
| Social Sciences and Management| Massachusetts Institute of
    ↪ Technology (MIT) | 4
    ↪                                            | 1
    ↪                                | Not Available
    ↪                                            | https://web.mit.edu
    ↪                        | January 5                 | $75
    ↪                |
| Social Sciences and Management| University of Cambridge
    ↪                            | 5
    ↪                                            | 5
    ↪                                | Not Available
    ↪                                            | https://www.cam.ac.
    ↪ uk                     | October 15                | Not
    ↪ Available     |
```

**Notes**:
1. Rankings marked "Not Available" indicate insufficient public
    ↪ data from official QS/THE 2025 rankings.

```
2. Application deadlines and fees are primarily sourced from
   ↪ university admissions pages or verified third-party platforms
   ↪ ; some may vary by program.
3. For the most accurate and up-to-date information, refer to the
   ↪ official university admissions websites.
```

## Case Trajectory for Multi-Agent

**Question:** My son is about to start his university applications but he's still uncertain about both his major and which universities to apply to. Could you help me find the top five universities in each of the five broad subjects from the QS World University Rankings by Subject 2025, and also check their standings in the QS World University Rankings 2025 and the Times Higher Education World University Rankings 2025? And I need the home page of the university's official website, standard application deadline for regular decison as well as the application fee without the fee wavier.Please organize the results in one Markdown table with the following columns:
Subject, University, QS World University Rankings by Subject 2025, QS World University Rankings 2025, Times Higher Education World University Rankings 2025, Home Page, Application Deadline, Application Fee
Please use the universities' full official names in English.
Use only Arabic numerals in the ranking, for example: 1.
The output format is "'markdown
{data_content}
"'.

| **Step 1** |

**Reasoning Content**
Let me analyze this problem step by step. The user needs to help their child prepare for university applications by gathering information on the top five universities across five broad subjects from the QS World University Rankings by Subject 2025. Additionally, they require these universities' overall rankings in the QS World University Rankings 2025 and Times Higher Education (THE) World University Rankings 2025, along with their official website homepages, regular decision application deadlines, and application fees (excluding fee waivers). The results should be organized into a specific Markdown table.
First, I need to clarify the user's requirements:
1. **Five broad subjects**: The user mentions "five broad subjects" but does not specify which ones. Common broad disciplines include Engineering & Technology, Natural Sciences, Social Sciences & Management, Arts & Humanities, and Life Sciences & Medicine. However, I must confirm whether the QS Rankings by Subject 2025 uses these categories or others. This step requires research to identify the exact five broad subject areas covered in the 2025 rankings.
2. **Top five universities per subject**: For each of the five subjects, select the top five universities from the QS Subject Rankings 2025.
3. **Overall rankings**: For each selected university, obtain their positions in the QS World University Rankings 2025 and THE World University Rankings 2025.
4. **University details**: Official website homepage URL, regular decision application deadline, and application fee (without waivers).
Next, consider the challenges:
- **Timeliness**: The current time is April 2025, so the 2025 rankings should already be released. However, I need to verify whether the QS and THE 2025 rankings are publicly available. If not, the user may need the latest available data, but the question explicitly specifies 2025, so I must use that year's rankings.
- **Data accuracy**: University rankings, deadlines, and fees can change annually. For example, application deadlines may vary by program or country (e.g., US universities typically have deadlines in November-January, while UK universities may have different timelines). Fees also vary by country and program.
- **Standardization**: Ensure university names are in full official English, rankings use

Arabic numerals, and deadlines/fees are presented uniformly (e.g., deadline format: Month Day, Year; fees in USD or local currency? The user does not specify currency, so perhaps use the university's stated currency or convert to USD? Need to clarify, but the user may prefer the original currency as listed on the university's website).

**Step-by-Step Plan**:

1. **Identify the five broad subjects in QS Subject Rankings 2025**:
- Search for the official QS World University Rankings by Subject 2025 to confirm the classification of broad subject areas. For example, QS typically divides subjects into five broad categories: Arts & Humanities, Engineering & Technology, Life Sciences & Medicine, Natural Sciences, and Social Sciences & Management. However, this may change, so verification is essential.

2. **Retrieve top five universities for each subject**:
- For each confirmed broad subject, extract the top five universities from the QS Subject Rankings 2025. Note that some universities may appear in multiple subjects (e.g., MIT in Engineering and Natural Sciences), but they will be listed separately under each relevant subject.

3. **Collect overall rankings (QS and THE 2025)**:
- For each university, search both the QS World University Rankings 2025 and THE World University Rankings 2025 to obtain their overall positions. Note that some universities may not be ranked in one of the lists, but this is unlikely for top institutions.

4. **Gather university-specific details**:
- **Official homepage**: Directly from the university's website; ensure URLs are correct and start with http/https.
- **Regular decision deadline**: For undergraduate programs (assuming undergraduate applications, as the user mentions "university applications" generally). Deadlines vary by country and program; need to confirm the standard undergraduate regular decision deadline (e.g., January 1 for US universities, January 15 for others, etc.).
- **Application fee**: Excluding fee waivers; note the currency (e.g., USD, GBP) as listed by the university.

**Challenges and Solutions**:
- **Unspecified subjects**: Since the user did not specify the five subjects, I must first determine the exact broad subject categories from the QS 2025 rankings. This requires searching the QS official website or reliable sources for the 2025 subject classification.
- **Overlapping universities**: The same university may top multiple subjects (e.g., Harvard in Social Sciences). Each occurrence will need separate entries in the table under their respective subjects.
- **Data availability**: If the 2025 rankings are not yet fully published (e.g., some released, others pending), need to use the latest available data or clarify with the user. However, the user explicitly requests 2025, so assume all rankings are published.
- **Deadlines and fees**: These can vary by program (e.g., undergraduate vs. graduate). Since the user does not specify, focus on undergraduate regular decision deadlines and fees for the main undergraduate application portal (e.g., Common App, UCAS).

**Tool Utilization**:
- **Search tool**: Use 'search_global' to:
- Confirm QS 2025 broad subject categories.
- Retrieve top five universities per subject from QS Subject Rankings 2025.
- Obtain QS and THE 2025 overall rankings for each university.
- Find each university's application deadlines and fees.
- **Link reading tool**: After finding university websites via search, use 'text_browser_view' to extract deadlines and fees from official admissions pages (e.g., check "Undergraduate Admissions" or "How to Apply" sections).
- **Sub-agents**: Given the volume of tasks (five subjects × five universities = 25 universities, each requiring multiple data points), creating sub-agents to parallelize work could improve efficiency. For example:
- Sub-agent 1: Confirm QS 2025 broad subjects.
- Sub-agent 2: For each subject, retrieve top five universities and their QS Subject rankings.
- Sub-agent 3: Collect QS and THE overall rankings for each university.

- Sub-agent 4: Gather website URLs, deadlines, and fees for each university.
**Verification Steps**:
1. Ensure all rankings are for 2025.
2. Cross-check university names with official sources to avoid aliases (e.g., "MIT" vs. "Massachusetts Institute of Technology").
3. Confirm deadlines are for regular decision (not early action/decision) and undergraduate programs.
4. Verify application fees exclude waivers and note the correct currency.
**Potential Edge Cases**:
- A university may not have an undergraduate program (e.g., graduate-only institutions), but top-ranked universities typically offer undergrad programs.
- Deadlines may differ by country (e.g., UK universities via UCAS have a single deadline, while US universities vary).
- Application fees may be waived for certain regions, but the user explicitly requests fees without waivers.
**Execution Plan**:
1. **Step 1: Identify QS 2025 broad subject categories**
Use 'search_global' with query: "QS World University Rankings by Subject 2025 broad subject categories" to confirm the five categories. Assume results return Arts & Humanities, Engineering & Technology, Life Sciences & Medicine, Natural Sciences, Social Sciences & Management.
2. **Step 2: Retrieve top five universities per subject**
For each subject, search "QS Subject Rankings 2025 [Subject Name] top 5 universities". For example, "QS Subject Rankings 2025 Engineering & Technology top 5 universities". Record university names and their subject rankings.
3. **Step 3: Collect overall rankings**
For each university, search "QS World University Rankings 2025 [University Name]" and "THE World University Rankings 2025 [University Name]" to get their overall positions.
4. **Step 4: Gather university details**
For each university:
- Search official homepage URL (e.g., "Harvard University official website").
- Use 'text_browser_view' on the admissions page to find deadlines and fees. For example, visit "https://college.harvard.edu/admissions/apply/application-process/application-fees" to confirm the fee.
**Example Table Structure**:
— Subject — University — QS Subject 2025 — QS Overall 2025 — THE Overall 2025 — Home Page — Deadline — Application Fee —
________________________________________________________________________________
__________________________________________________________
— Engineering & Tech — Massachusetts Institute...— 1 — 1 — 2 — https://www.mit.edu — Jan 5, 2025— $75 —
**Now, determine the initial tool calls**:
- First, confirm the five broad subjects via 'search_global'.
- Then, for each subject, retrieve top five universities.
- For each university, collect overall rankings, website, deadline, fee.
Given the complexity, creating sub-agents to handle each subject's data collection in parallel would be efficient. For example:
- Sub-agent 0: Handle Arts & Humanities
- Sub-agent 1: Handle Engineering & Technology
- Sub-agent 2: Handle Life Sciences & Medicine
- Sub-agent 3: Handle Natural Sciences
- Sub-agent 4: Handle Social Sciences & Management
Each sub-agent would be tasked with:
1. Retrieving top five universities in their subject from QS 2025.
2. For each university, get QS overall and THE 2025 rankings.
3. Collect website, deadline, and fee.
Thus, the first step is to create these five sub-agents using 'create_sub_agents', each with a

prompt tailored to their subject. Then, each sub-agent can independently gather the required data, which can later be aggregated into the final table.
**Tool Call**:
Use 'create_sub_agents' to generate five sub-agents, each assigned to one of the five broad subjects (after confirming the subjects via initial search). However, since the subjects are not yet confirmed, the initial step must first search for the QS 2025 broad subject categories.
**Revised Plan**:
1. **Initial search**: Confirm the five broad subjects in QS Subject Rankings 2025.
- Tool: 'search_global' with query "QS World University Rankings by Subject 2025 broad subject categories".
- Purpose: Obtain the exact five subject areas to ensure accuracy.
2. **Create sub-agents per subject**: Once subjects are confirmed, create a sub-agent for each to collect data.
Thus, the first tool call is a 'search_global' to identify the five subjects. Once subjects are known, create sub-agents for each.
**Final Tool Call**:
First, execute a search to confirm the five broad subjects. Then proceed with sub-agents. Therefore, the initial action is:
<—FunctionCallBegin—> [{"name":"search_global","parameters":{"query":"QS World University Rankings by Subject 2025 broad subject categories","count":10,"summary_type":"short","use_english":false}}]
<—FunctionCallEnd—<
This search will return the necessary subject categories, enabling the creation of targeted sub-agents in subsequent steps.
**Tool Call**

```
{
  "tool_name": "search_global",
  "arguments": "{\"query\":\"QS World University Rankings by
      ↪ Subject 2025 broad subject categories\",\"count\":10,\"
      ↪ summary_type\":\"short\", \"use_english\":false}",
  "tool_call_id": "call_fq44yo2q4midba2zrovw1wop"
}
```

**Tool Response**

```
[index] 0
[siteName] qs.com
...
(search content omitted due to length limit)
```

**Step 2**

**Reasoning Content**
Now I need to continue analyzing the user's question. The user wants the top five universities in each of the five broad subjects from the QS World University Rankings by Subject 2025, along with their positions in the QS and THE rankings, official website homepage, application deadlines for regular decision, and application fees. The results must be organized into a Markdown table with specific columns.
First, the previous search results have confirmed the five broad subject categories in the QS 2025 Subject Rankings: Arts and Humanities, Engineering and Technology, Life Sciences and Medicine, Natural Sciences, and Social Sciences and Management. This addresses the subject classification issue.
Next, I need to obtain the top five universities for each subject. The current search results mention that the QS rankings cover these five broad subjects but do not provide specific university names or rankings. For example, the first result from qs.com states the five broad subjects but lacks detailed rankings. Other results either mention specific disciplines (e.g., Accounting & Finance, Pharmacy & Pharmacology) or discuss ranking expansions without providing the required top five list for each broad subject.
Therefore, the next step is to search for the top five universities in each of these five broad subjects. Since each subject requires separate searching, I should split this into five sub-

tasks. For instance, search for "QS World University Rankings by Subject 2025 Arts and Humanities top 5" and repeat similarly for the other four subjects. However, given the limitations of search tools, I might need to use multiple search queries or consider creating sub-agents to handle parallel searches.

Additionally, the user requires the universities' positions in the overall QS World University Rankings 2025 and THE World University Rankings 2025. The current search results do not include these overall rankings, so separate searches will be needed for both QS and THE's 2025 overall rankings. This adds two more search tasks.

Furthermore, for each university, the official website homepage, regular decision application deadline, and application fee (without fee waivers) must be collected. These details are specific to each university and may require visiting their official websites or trusted educational platforms. However, given the large number of universities (5 subjects × 5 universities = 25 total), this could be time-consuming. Using sub-agents to handle different subjects or universities in parallel might improve efficiency.

Now, considering the tool options: the create_sub_agents function allows creating sub-agents to perform specific tasks. Since the user needs information across multiple subjects and rankings, creating sub-agents for each broad subject to collect top five universities, overall rankings, and application details would be appropriate. Alternatively, split the tasks into:

1. Retrieve top five universities for each of the five QS broad subjects.
2. Retrieve QS 2025 overall rankings.
3. Retrieve THE 2025 overall rankings.
4. For each university, collect official website, deadline, and application fee.

But given the complexity, creating sub-agents for each broad subject might be more efficient. For example, create five sub-agents, each responsible for one subject, tasked with finding the top five universities, their QS subject ranking, overall QS and THE rankings, official website, deadline, and fee. However, sub-agents can use search and link-reading tools, so each sub-agent could independently complete its assigned subject's tasks.

Alternatively, first obtain the top five universities for each subject, then collect their overall rankings and application details. The first step is to confirm the top five universities per subject. Since the current search results lack this information, further searches are necessary.

Possible search strategies: For each subject, use precise keywords like "QS World University Rankings by Subject 2025 [Subject Name] top 5 universities". For example:
- QS World University Rankings by Subject 2025 Arts and Humanities top 5
- QS World University Rankings by Subject 2025 Engineering and Technology top 5
... and so on for the remaining three subjects.

Since each subject requires a separate search, using parallel tool calls to search for all five subjects simultaneously would save time. However, the current search_global tool allows only one query per call, so multiple search_global calls are needed. Alternatively, create sub-agents for each subject to perform searches in parallel.

Given that the user needs results for five subjects, creating five sub-agents (each assigned to one subject) with prompts instructing them to find the top five universities, their QS subject ranking, overall QS and THE rankings, official website, deadline, and fee would be efficient. Each sub-agent can independently search and compile data for its subject, then aggregate the results into a table.

However, the create_sub_agents tool requires specifying prompts and indexes for each sub-agent. For example:

Sub-agent 0: Prompt - "Find the top 5 universities in QS World University Rankings by Subject 2025 for Arts and Humanities, including their QS subject ranking, QS overall ranking 2025, THE overall ranking 2025, official website homepage, regular decision application deadline, and application fee without waiver."

Similarly for sub-agents 1 to 4, corresponding to the other four subjects.

But creating five sub-agents might be manageable. Each sub-agent would handle one subject, search for the required information, and return the results. Then, the main agent aggregates all data into the table.

However, there's a potential issue: Sub-agents might encounter difficulties in retrieving all required information (e.g., application deadlines and fees may not be easily available in search results and might require accessing university websites). Thus, sub-agents may need

to use the text_browser_view tool to extract details from official sites.

Another consideration: The QS and THE overall rankings for 2025 might not yet be publicly available. The current search results include a THE article from January 2025 and QS-related articles from March 2025, suggesting the rankings are released. However, specific university positions need verification.

Additionally, application deadlines and fees vary by program and country. The user specifies "standard application deadline for regular decision" and "application fee without fee waiver," which may require checking each university's undergraduate admissions page. This could be time-consuming, but sub-agents can handle this by searching for each university's admissions information.

In summary, the next steps should be:

1. Create five sub-agents, each responsible for one of the five broad subjects.

2. Each sub-agent searches for the top five universities in its assigned subject, retrieves their QS subject ranking, QS and THE overall rankings, official website, deadline, and fee.

3. Aggregate all sub-agent results into the required table.

Thus, the next tool call should use create_sub_agents to generate these five sub-agents. Each sub-agent's prompt must clearly specify the task: retrieve the top five universities for the subject, along with the required six data points (QS subject ranking, QS overall, THE overall, homepage, deadline, fee), ensuring the university's full official English name and numeric rankings.

Now, constructing the create_sub_agents parameters. The sub_agents array should contain five objects, each with prompt and index (0 to 4). The prompts must be detailed enough for sub-agents to understand the requirements.

For example, the prompt for index 0 (Arts and Humanities) could be: "Task: Retrieve the top 5 universities in the 'Arts and Humanities' category of the QS World University Rankings by Subject 2025. For each university, collect: 1. Full official English name; 2. QS World University Rankings by Subject 2025 rank (numeric); 3. QS World University Rankings 2025 overall rank (numeric); 4. Times Higher Education World University Rankings 2025 overall rank (numeric); 5. Official website homepage URL; 6. Standard regular decision application deadline (format: Month Day, Year, e.g., January 15, 2025); 7. Application fee (without fee waiver, specify currency, e.g., USD 80). Ensure all rankings are Arabic numerals. If any data is unavailable, mark as 'N/A'. Return results in a list of dictionaries with keys matching the required fields."

Similarly, adjust the subject name for the other four sub-agents (Engineering and Technology, Life Sciences and Medicine, Natural Sciences, Social Sciences and Management).

This approach ensures each sub-agent knows its task and data requirements. Using sub-agents allows parallel processing, improving efficiency.

Therefore, the next step is to call create_sub_agents with these five sub-agents.

**Tool Call**

```
{
  "tool_name": "create_sub_agents",
  "arguments": "{\"sub_agents\":[\n{\"prompt\":\"Retrieve the top 5
      ↪  universities in the 'Arts and Humanities' category of the
      ↪ QS World University Rankings by Subject 2025. For each
      ↪ university, collect: 1. Full official English name; 2. QS
      ↪ World University Rankings by Subject 2025 rank (numeric);
      ↪ 3. QS World University Rankings 2025 overall rank (numeric)
      ↪ ; 4. Times Higher Education World University Rankings 2025
      ↪ overall rank (numeric); 5. Official website homepage URL;
      ↪ 6. Standard regular decision application deadline (format:
      ↪ Month Day, Year, e.g., January 15, 2025); 7. Application
      ↪ fee (without fee waiver, specify currency, e.g., USD 80).
      ↪ Ensure all rankings are Arabic numerals. If any data is
      ↪ unavailable, mark as 'N/A'. Return results in a list of
      ↪ dictionaries with keys: university_name, qs_subject_rank,
      ↪ qs_overall_rank, the_overall_rank, homepage, deadline,
      ↪ application_fee.\",\"index\":0},\n{\"prompt\":\"Retrieve
      ↪ the top 5 universities in the 'Engineering and Technology'
```

```
        ↪ category of the QS World University Rankings by Subject
        ↪ 2025. For each university, collect: 1. Full official
        ↪ English name; 2. QS World University Rankings by Subject
        ↪ 2025 rank (numeric); 3. QS World University Rankings 2025
        ↪ overall rank (numeric); 4. Times Higher Education World
        ↪ University Rankings 2025 overall rank (numeric); 5.
        ↪ Official website homepage URL; 6. Standard regular decision
        ↪  application deadline (format: Month Day, Year, e.g.,
        ↪ January 15, 2025); 7. Application fee (without fee waiver,
        ↪ specify currency, e.g., USD 80). Ensure all rankings are
        ↪ Arabic numerals. If any data is unavailable, mark as 'N/A'.
        ↪  Return results in a list of dictionaries with keys:
        ↪ university_name, qs_subject_rank, qs_overall_rank,
        ↪ the_overall_rank, homepage, deadline, application_fee.\",\"
        ↪ index\":1},\n{\"prompt\":\"Retrieve the top 5 universities
        ↪ in the 'Life Sciences and Medicine' category of the QS
        ↪ World University Rankings by Subject 2025. For each
        ↪ university, collect: 1. Full official English name; 2. QS
        ↪ World University Rankings by Subject 2025 rank (numeric);
        ↪ 3. QS World University Rankings 2025 overall rank (numeric)
        ↪ ; 4. Times Higher Education World University Rankings 2025
        ↪ overall rank (numeric); 5. Official website homepage URL;
        ↪ 6. Standard regular decision application deadline (format:
        ↪ Month Day, Year, e.g., January 15, 2025); 7. Application
        ↪ fee (without fee waiver, specify currency, e.g., USD 80).
        ↪ Ensure all rankings are Arabic numerals. If any data is
        ↪ unavailable, mark as 'N/A'. Return results in a list of
        ↪ dictionaries with keys: university_name, qs_subject_rank,
        ↪ qs_overall_rank, the_overall_rank, homepage, deadline,
        ↪ application_fee.\",\"index\":2},\n{\"prompt\":\"Retrieve
        ↪ the top 5 universities in the 'Natural Sciences' category
        ↪ of the QS World University Rankings by Subject 2025. For
        ↪ each university, collect: 1. Full official English name; 2.
        ↪  QS World University Rankings by Subject 2025 rank (numeric
        ↪ ); 3. QS World University Rankings 2025 overall rank (
        ↪ numeric); 4. Times Higher Education World University
        ↪ Rankings 2025 overall rank (numeric); 5. Official website
        ↪ homepage URL; 6. Standard regular decision application
        ↪ deadline (format: Month Day, Year, e.g., January 15, 2025);
        ↪  7. Application fee (without fee waiver, specify currency,
        ↪ e.g., USD 80). Ensure all rankings are Arabic numerals. If
        ↪ any data is unavailable, mark as 'N/A'. Return results in a
        ↪  list of dictionaries with keys: university_name,
        ↪ qs_subject_rank, qs_overall_rank, the_overall_rank,
        ↪ homepage, deadline, application_fee.\",\"index\":3},\n{\"
        ↪ prompt\":\"Retrieve the top 5 universities in the 'Social
        ↪ Sciences and Management' category of the QS World
        ↪ University Rankings by Subject 2025. For each university,
        ↪ collect: 1. Full official English name; 2. QS World
        ↪ University Rankings by Subject 2025 rank (numeric); 3. QS
        ↪ World University Rankings 2025 overall rank (numeric); 4.
        ↪ Times Higher Education World University Rankings 2025
        ↪ overall rank (numeric); 5. Official website homepage URL;
        ↪ 6. Standard regular decision application deadline (format:
        ↪ Month Day, Year, e.g., January 15, 2025); 7. Application
        ↪ fee (without fee waiver, specify currency, e.g., USD 80).
        ↪ Ensure all rankings are Arabic numerals. If any data is
        ↪ unavailable, mark as 'N/A'. Return results in a list of
        ↪ dictionaries with keys: university_name, qs_subject_rank,
        ↪ qs_overall_rank, the_overall_rank, homepage, deadline,
        ↪ application_fee.\",\"index\":4}\n]}",
  "tool_call_id": "call_fgjik45vidu9usrr80ojuga7"
}
```

**Tool Response**

[{"index": 0, "prompt": "Retrieve the top 5 universities in the '
↪ Arts and Humanities' category of the QS World University
↪ Rankings by Subject 2025. For each university, collect: 1.
↪ Full official English name; 2. QS World University Rankings
↪ by Subject 2025 rank (numeric); 3. QS World University
↪ Rankings 2025 overall rank (numeric); 4. Times Higher
↪ Education World University Rankings 2025 overall rank (
↪ numeric); 5. Official website homepage URL; 6. Standard
↪ regular decision application deadline (format: Month Day,
↪ Year, e.g., January 15, 2025); 7. Application fee (without
↪ fee waiver, specify currency, e.g., USD 80). Ensure all
↪ rankings are Arabic numerals. If any data is unavailable,
↪ mark as 'N/A'. Return results in a list of dictionaries with
↪ keys: university_name, qs_subject_rank, qs_overall_rank,
↪ the_overall_rank, homepage, deadline, application_fee.", "
↪ response": "[\n    {\n        \"university_name\": \"Harvard
↪ University\",\n        \"qs_subject_rank\": 1,\n        \"
↪ qs_overall_rank\": 4,\n        \"the_overall_rank\": 3,\n
↪         \"homepage\": \"https://www.harvard.edu\",\n        \"
↪ deadline\": \"January 1, 2025\",\n        \"application_fee
↪ \": \"USD 85\"\n    },\n    {\n        \"university_name\":
↪ \"University of Oxford\",\n        \"qs_subject_rank\": 2,\n
↪         \"qs_overall_rank\": 3,\n        \"the_overall_rank\":
↪  1,\n        \"homepage\": \"https://www.ox.ac.uk\",\n
↪         \"deadline\": \"October 15, 2024\",\n        \"
↪ application_fee\": \"GBP 28.50\"\n    },\n    {\n        \"
↪ university_name\": \"University of Cambridge\",\n        \"
↪ qs_subject_rank\": 3,\n        \"qs_overall_rank\": 5,\n
↪         \"the_overall_rank\": 5,\n        \"homepage\": \"
↪ https://www.cam.ac.uk\",\n        \"deadline\": \"October 15,
↪  2024\",\n        \"application_fee\": \"GBP 60\"\n    },\n
↪     {\n        \"university_name\": \"Stanford University\",\n
↪         \"qs_subject_rank\": 4,\n        \"qs_overall_rank\":
↪  6,\n        \"the_overall_rank\": 6,\n        \"homepage\":
↪ \"https://www.stanford.edu\",\n        \"deadline\": \"
↪ January 5, 2025\",\n        \"application_fee\": \"USD 90\"\n
↪     },\n    {\n        \"university_name\": \"University of
↪ California, Berkeley (UCB)\",\n        \"qs_subject_rank\":
↪ 5,\n        \"qs_overall_rank\": 12,\n        \"
↪ the_overall_rank\": 8,\n        \"homepage\": \"https://www.
↪ berkeley.edu\",\n        \"deadline\": \"November 30,
↪ 2024\",\n        \"application_fee\": \"USD 80\"\n    }\n]"},
↪  {"index": 1, "prompt": "Retrieve the top 5 universities in
↪ the 'Engineering and Technology' category of the QS World
↪ University Rankings by Subject 2025. For each university,
↪ collect: 1. Full official English name; 2. QS World
↪ University Rankings by Subject 2025 rank (numeric); 3. QS
↪ World University Rankings 2025 overall rank (numeric); 4.
↪ Times Higher Education World University Rankings 2025 overall
↪  rank (numeric); 5. Official website homepage URL; 6.
↪ Standard regular decision application deadline (format: Month
↪  Day, Year, e.g., January 15, 2025); 7. Application fee (
↪ without fee waiver, specify currency, e.g., USD 80). Ensure
↪ all rankings are Arabic numerals. If any data is unavailable,
↪  mark as 'N/A'. Return results in a list of dictionaries with
↪  keys: university_name, qs_subject_rank, qs_overall_rank,
↪ the_overall_rank, homepage, deadline, application_fee.", "
↪ response": "[\n    {\n        \"university_name\": \"
↪ Massachusetts Institute of Technology (MIT)\",\n        \"
↪ qs_subject_rank\": 1,\n        \"qs_overall_rank\": 1,\n
↪         \"the_overall_rank\": 2,\n        \"homepage\": \"http

```
  ↪ ://web.mit.edu/\",\n        \"deadline\": \"January 6,
  ↪ 2025\",\n        \"application_fee\": \"USD 75\"\n    },\n
  ↪    {\n        \"university_name\": \"University of Oxford\",\
  ↪ n        \"qs_subject_rank\": 2,\n        \"qs_overall_rank
  ↪ \": 3,\n        \"the_overall_rank\": 1,\n        \"homepage
  ↪ \": \"https://www.ox.ac.uk/\",\n        \"deadline\": \"
  ↪ October 15, 2024\",\n        \"application_fee\": \"GBP
  ↪ 28.50\"\n    },\n    {\n        \"university_name\": \"
  ↪ Stanford University\",\n        \"qs_subject_rank\": 3,\n
  ↪       \"qs_overall_rank\": 6,\n        \"the_overall_rank\":
  ↪  6,\n        \"homepage\": \"https://www.stanford.edu/\",\n
  ↪        \"deadline\": \"January 5, 2025\",\n        \"
  ↪ application_fee\": \"USD 90\"\n    },\n    {\n        \"
  ↪ university_name\": \"University of Cambridge\",\n        \"
  ↪ qs_subject_rank\": 4,\n        \"qs_overall_rank\": 5,\n
  ↪       \"the_overall_rank\": 5,\n        \"homepage\": \"
  ↪ https://www.cam.ac.uk/\",\n        \"deadline\": \"October
  ↪ 15, 2024\",\n        \"application_fee\": \"GBP 75\"\n    },\
  ↪ n    {\n        \"university_name\": \"ETH Zurich\",\n
  ↪        \"qs_subject_rank\": 5,\n        \"qs_overall_rank\":
  ↪ 7,\n        \"the_overall_rank\": 11,\n        \"homepage\":
  ↪ \"https://www.ethz.ch/\",\n        \"deadline\": \"March 31,
  ↪ 2025\",\n        \"application_fee\": \"CHF 150\"\n    }\n
  ↪ ]"}, {"index": 2, "prompt": "Retrieve the top 5 universities
  ↪ in the 'Life Sciences and Medicine' category of the QS World
  ↪ University Rankings by Subject 2025. For each university,
  ↪ collect: 1. Full official English name; 2. QS World
  ↪ University Rankings by Subject 2025 rank (numeric); 3. QS
  ↪ World University Rankings 2025 overall rank (numeric); 4.
  ↪ Times Higher Education World University Rankings 2025 overall
  ↪  rank (numeric); 5. Official website homepage URL; 6.
  ↪ Standard regular decision application deadline (format: Month
  ↪  Day, Year, e.g., January 15, 2025); 7. Application fee (
  ↪ without fee waiver, specify currency, e.g., USD 80). Ensure
  ↪ all rankings are Arabic numerals. If any data is unavailable,
  ↪  mark as 'N/A'. Return results in a list of dictionaries with
  ↪  keys: university_name, qs_subject_rank, qs_overall_rank,
  ↪ the_overall_rank, homepage, deadline, application_fee.", "
  ↪ response": "[\n    {\n        \"university_name\": \"Harvard
  ↪ University\",\n        \"qs_subject_rank\": 1,\n        \"
  ↪ qs_overall_rank\": 4,\n        \"the_overall_rank\": 1,\n
  ↪        \"homepage\": \"https://www.harvard.edu\",\n        \"
  ↪ deadline\": \"January 1, 2025\",\n        \"application_fee
  ↪ \": \"USD 85\"\n    },\n    {\n        \"university_name\":
  ↪ \"University of Oxford\",\n        \"qs_subject_rank\": 2,\n
  ↪        \"qs_overall_rank\": 3,\n        \"the_overall_rank\":
  ↪  1,\n        \"homepage\": \"https://www.ox.ac.uk\",\n
  ↪        \"deadline\": \"N/A\",\n        \"application_fee\":
  ↪ \"GBP 28.50\"\n    },\n    {\n        \"university_name\": \"
  ↪ Johns Hopkins University\",\n        \"qs_subject_rank\": 3,\
  ↪ n        \"qs_overall_rank\": 32,\n        \"the_overall_rank
  ↪ \": 16,\n        \"homepage\": \"https://www.jhu.edu\",\n
  ↪        \"deadline\": \"January 2, 2025\",\n        \"
  ↪ application_fee\": \"USD 70\"\n    },\n    {\n        \"
  ↪ university_name\": \"Stanford University\",\n        \"
  ↪ qs_subject_rank\": 4,\n        \"qs_overall_rank\": 6,\n
  ↪        \"the_overall_rank\": 6,\n        \"homepage\": \"
  ↪ https://www.stanford.edu\",\n        \"deadline\": \"January
  ↪ 5, 2025\",\n        \"application_fee\": \"USD 90\"\n    },\n
  ↪     {\n        \"university_name\": \"Massachusetts Institute
  ↪  of Technology (MIT)\",\n        \"qs_subject_rank\": 5,\n
  ↪        \"qs_overall_rank\": 1,\n        \"the_overall_rank\":
  ↪  2,\n        \"homepage\": \"https://www.mit.edu\",\n
```

```
↪ \"deadline\": \"January 6, 2025\",\n        \"application_fee
↪ \": \"USD 75\"\n    }\n]"}, {"index": 3, "prompt": "Retrieve
↪ the top 5 universities in the 'Natural Sciences' category of
↪ the QS World University Rankings by Subject 2025. For each
↪ university, collect: 1. Full official English name; 2. QS
↪ World University Rankings by Subject 2025 rank (numeric); 3.
↪ QS World University Rankings 2025 overall rank (numeric); 4.
↪ Times Higher Education World University Rankings 2025 overall
↪  rank (numeric); 5. Official website homepage URL; 6.
↪ Standard regular decision application deadline (format: Month
↪  Day, Year, e.g., January 15, 2025); 7. Application fee (
↪ without fee waiver, specify currency, e.g., USD 80). Ensure
↪ all rankings are Arabic numerals. If any data is unavailable,
↪  mark as 'N/A'. Return results in a list of dictionaries with
↪  keys: university_name, qs_subject_rank, qs_overall_rank,
↪ the_overall_rank, homepage, deadline, application_fee.", "
↪ response": "[\n  {\n    \"university_name\": \"Harvard
↪ University\",\n    \"qs_subject_rank\": 1,\n    \"
↪ qs_overall_rank\": 4,\n    \"the_overall_rank\": 3,\n    \"
↪ homepage\": \"https://www.harvard.edu\",\n    \"deadline\":
↪ \"January 1, 2025\",\n    \"application_fee\": \"USD 85\"\n
↪ },\n  {\n    \"university_name\": \"Massachusetts Institute
↪ of Technology (MIT)\",\n    \"qs_subject_rank\": 2,\n    \"
↪ qs_overall_rank\": 1,\n    \"the_overall_rank\": 2,\n    \"
↪ homepage\": \"https://web.mit.edu\",\n    \"deadline\": \"
↪ January 6, 2025\",\n    \"application_fee\": \"USD 75\"\n
↪ },\n  {\n    \"university_name\": \"University of Oxford\",\n
↪     \"qs_subject_rank\": 3,\n    \"qs_overall_rank\": 3,\n
↪     \"the_overall_rank\": 1,\n    \"homepage\": \"https://www.
↪ ox.ac.uk\",\n    \"deadline\": \"October 15, 2024\",\n    \"
↪ application_fee\": \"GBP 28.50\"\n  },\n  {\n    \"
↪ university_name\": \"University of Cambridge\",\n    \"
↪ qs_subject_rank\": 4,\n    \"qs_overall_rank\": 5,\n    \"
↪ the_overall_rank\": 5,\n    \"homepage\": \"https://www.cam.
↪ ac.uk\",\n    \"deadline\": \"October 15, 2024\",\n    \"
↪ application_fee\": \"GBP 60\"\n  },\n  {\n    \"
↪ university_name\": \"Stanford University\",\n    \"
↪ qs_subject_rank\": 5,\n    \"qs_overall_rank\": 6,\n    \"
↪ the_overall_rank\": 6,\n    \"homepage\": \"https://www.
↪ stanford.edu\",\n    \"deadline\": \"January 5, 2025\",\n
↪ \"application_fee\": \"USD 90\"\n  }\n]"}, {"index": 4, "
↪ prompt": "Retrieve the top 5 universities in the 'Social
↪ Sciences and Management' category of the QS World University
↪ Rankings by Subject 2025. For each university, collect: 1.
↪ Full official English name; 2. QS World University Rankings
↪ by Subject 2025 rank (numeric); 3. QS World University
↪ Rankings 2025 overall rank (numeric); 4. Times Higher
↪ Education World University Rankings 2025 overall rank (
↪ numeric); 5. Official website homepage URL; 6. Standard
↪ regular decision application deadline (format: Month Day,
↪ Year, e.g., January 15, 2025); 7. Application fee (without
↪ fee waiver, specify currency, e.g., USD 80). Ensure all
↪ rankings are Arabic numerals. If any data is unavailable,
↪ mark as 'N/A'. Return results in a list of dictionaries with
↪ keys: university_name, qs_subject_rank, qs_overall_rank,
↪ the_overall_rank, homepage, deadline, application_fee.", "
↪ response": "[\n  {\n    \"university_name\": \"Harvard
↪ University\",\n    \"qs_subject_rank\": 1,\n    \"
↪ qs_overall_rank\": 4,\n    \"the_overall_rank\": 3,\n    \"
↪ homepage\": \"https://www.harvard.edu\",\n    \"deadline\":
↪ \"January 1, 2025\",\n    \"application_fee\": \"USD 85\"\n
↪ },\n  {\n    \"university_name\": \"University of Oxford\",\n
↪     \"qs_subject_rank\": 2,\n    \"qs_overall_rank\": 3,\n
```

```
↪    \"the_overall_rank\": 1,\n    \"homepage\": \"https://www.
↪ ox.ac.uk\",\n    \"deadline\": \"October 15, 2025\",\n    \"
↪ application_fee\": \"GBP 75\"\n  },\n  {\n    \"
↪ university_name\": \"Stanford University\",\n    \"
↪ qs_subject_rank\": 3,\n    \"qs_overall_rank\": 6,\n    \"
↪ the_overall_rank\": 6,\n    \"homepage\": \"https://www.
↪ stanford.edu\",\n    \"deadline\": \"January 5, 2025\",\n
↪ \"application_fee\": \"USD 90\"\n  },\n  {\n    \"
↪ university_name\": \"Massachusetts Institute of Technology (
↪ MIT)\",\n    \"qs_subject_rank\": 4,\n    \"qs_overall_rank
↪ \": 1,\n    \"the_overall_rank\": 2,\n    \"homepage\": \"
↪ http://web.mit.edu/\",\n    \"deadline\": \"January 6,
↪ 2025\",\n    \"application_fee\": \"USD 75\"\n  },\n  {\n
↪ \"university_name\": \"University of Cambridge\",\n    \"
↪ qs_subject_rank\": 5,\n    \"qs_overall_rank\": 5,\n    \"
↪ the_overall_rank\": 5,\n    \"homepage\": \"https://www.cam.
↪ ac.uk\",\n    \"deadline\": \"October 15, 2025\",\n    \"
↪ application_fee\": \"GBP 50\"\n  }\n]"}]
```

### Step 3

**Reasoning Content**

Got it, let's tackle this user's request. They need a Markdown table with the top five universities in each of five broad subjects from QS Subject Rankings 2025, along with their overall QS and THE rankings, homepage, application deadline, and fee.

First, I confirmed the five broad subjects from the initial search: Arts and Humanities, Engineering and Technology, Life Sciences and Medicine, Natural Sciences, Social Sciences and Management. Then I created sub-agents for each subject to gather the required data.

Now, the sub-agents have returned their responses. I need to check each one for completeness. Let's go through each subject's data:

1. **Arts and Humanities**: Top 5 are Harvard, Oxford, Cambridge, Stanford, UCB. All fields seem filled except maybe deadlines and fees. Harvard's deadline is Jan 1, 2025, fee USD 85. Oxford's deadline is Oct 15, 2024, fee GBP 28.50. Cambridge Oct 15, 2024, GBP 60. Stanford Jan 5, 2025, USD 90. UCB Nov 30, 2024, USD 80. Rankings are numeric. Looks good.

2. **Engineering and Technology**: MIT, Oxford, Stanford, Cambridge, ETH Zurich. MIT's deadline Jan 6, 2025, USD 75. Oxford same as before. Stanford same. Cambridge deadline Oct 15, 2024, fee GBP 75 (wait, earlier in Arts and Humanities, Cambridge's fee was GBP 60—need to check consistency. Maybe different programs? The user asked for standard regular decision, so perhaps it's correct if different subjects have different fees? Or maybe a mistake. But since the sub-agent provided it, I'll note it but proceed unless conflicting.) ETH Zurich deadline March 31, 2025, CHF 150. Rankings all numeric.

3. **Life Sciences and Medicine**: Harvard, Oxford, Johns Hopkins, Stanford, MIT. Harvard same. Oxford's deadline is N/A—sub-agent marked it as N/A, so we keep that. Johns Hopkins deadline Jan 2, 2025, USD 70. Stanford same. MIT same. Rankings okay.

4. **Natural Sciences**: Harvard, MIT, Oxford, Cambridge, Stanford. All data filled, deadlines and fees consistent with previous entries.

5. **Social Sciences and Management**: Harvard, Oxford, Stanford, MIT, Cambridge. Oxford's deadline here is Oct 15, 2025 (typo? Should be 2024?), Cambridge also Oct 15, 2025. That might be an error. The user needs accurate deadlines. Since the other subjects for Oxford and Cambridge have 2024, 2025 might be a mistake. But the sub-agent provided that, so maybe it's correct for this subject? Or perhaps a typo. Since I can't verify without further search, but the user wants the info as per the data, I'll include it but note that if there's a conflict, but since the sub-agent is supposed to retrieve accurate info, I'll proceed with what's given.

Now, compiling all into a table. The columns needed are: Subject, University, QS Subject Rank 2025, QS Overall 2025, THE Overall 2025, Home Page, Application Deadline, Application Fee.

Each subject will have 5 rows. Need to ensure university names are full official English

names. Check for duplicates in universities across subjects, but that's okay as they can top multiple subjects.

Check for any missing data marked as N/A (only Oxford in Life Sciences has N/A for deadline). All rankings are Arabic numerals. Currencies are specified (USD, GBP, CHF). URLs are complete.

Now, structure the Markdown table. Start with the header, then each row per university per subject. Make sure the Subject column repeats the broad subject for each of the 5 universities in that subject.

Let me draft the table:

— Subject — University — QS World University Rankings by Subject 2025 — QS World University Rankings 2025 — Times Higher Education World University Rankings 2025 — Home Page — Application Deadline — Application Fee —

__________________________________________________________________________
__________________________________________________________________________
__________

— Arts and Humanities — Harvard University — 1 — 4 — 3 — https://www.harvard.edu — January 1, 2025 — USD 85 —
— Arts and Humanities — University of Oxford — 2 — 3 — 1 — https://www.ox.ac.uk — October 15, 2024 — GBP 28.50 —
— ... and so on for each subject's 5 universities.

Need to fill in all 25 rows (5 subjects x 5 universities). Check each entry against the sub-agent responses to ensure accuracy.

For example, in Social Sciences and Management, Oxford's deadline is Oct 15, 2025, which differs from other subjects. Maybe it's correct for that program, so include as is. Cambridge's application fee here is GBP 50, whereas in Arts and Humanities it was GBP 60—again, possibly different programs, so acceptable.

Once all rows are filled, wrap in '''markdown and ensure the table is properly formatted with pipes and alignment. No markdown syntax errors.

I think that's all. The data from sub-agents seems complete, and the table can be constructed now.

**Final Answer**

```markdown
| Subject | University | QS World University Rankings by Subject
    ↪ 2025 | QS World University Rankings 2025 | Times Higher
    ↪ Education World University Rankings 2025 | Home Page |
    ↪ Application Deadline | Application Fee |
|---------|------------|---------------------------------------------- -------------------
    ↪
| Arts and Humanities | Harvard University | 1 | 4 | 3 | https://
    ↪ www.harvard.edu | January 1, 2025 | USD 85 |
| Arts and Humanities | University of Oxford | 2 | 3 | 1 | https://
    ↪ www.ox.ac.uk | October 15, 2024 | GBP 28.50 |
| Arts and Humanities | University of Cambridge | 3 | 5 | 5 | https
    ↪ ://www.cam.ac.uk | October 15, 2024 | GBP 60 |
| Arts and Humanities | Stanford University | 4 | 6 | 6 | https://
    ↪ www.stanford.edu | January 5, 2025 | USD 90 |
| Arts and Humanities | University of California, Berkeley (UCB) |
    ↪ 5 | 12 | 8 | https://www.berkeley.edu | November 30, 2024 |
    ↪ USD 80 |
| Engineering and Technology | Massachusetts Institute of
    ↪ Technology (MIT) | 1 | 1 | 2 | http://web.mit.edu/ | January
    ↪ 6, 2025 | USD 75 |
| Engineering and Technology | University of Oxford | 2 | 3 | 1 |
    ↪ https://www.ox.ac.uk/ | October 15, 2024 | GBP 28.50 |
| Engineering and Technology | Stanford University | 3 | 6 | 6 |
    ↪ https://www.stanford.edu/ | January 5, 2025 | USD 90 |
| Engineering and Technology | University of Cambridge | 4 | 5 | 5
    ↪ | https://www.cam.ac.uk/ | October 15, 2024 | GBP 75 |
```

```
| Engineering and Technology | ETH Zurich | 5 | 7 | 11 | https://
    ↪ www.ethz.ch/ | March 31, 2025 | CHF 150 |
| Life Sciences and Medicine | Harvard University | 1 | 4 | 1 |
    ↪ https://www.harvard.edu | January 1, 2025 | USD 85 |
| Life Sciences and Medicine | University of Oxford | 2 | 3 | 1 |
    ↪ https://www.ox.ac.uk | N/A | GBP 28.50 |
| Life Sciences and Medicine | Johns Hopkins University | 3 | 32 |
    ↪ 16 | https://www.jhu.edu | January 2, 2025 | USD 70 |
| Life Sciences and Medicine | Stanford University | 4 | 6 | 6 |
    ↪ https://www.stanford.edu | January 5, 2025 | USD 90 |
| Life Sciences and Medicine | Massachusetts Institute of
    ↪ Technology (MIT) | 5 | 1 | 2 | https://www.mit.edu | January
    ↪ 6, 2025 | USD 75 |
| Natural Sciences | Harvard University | 1 | 4 | 3 | https://www.
    ↪ harvard.edu | January 1, 2025 | USD 85 |
| Natural Sciences | Massachusetts Institute of Technology (MIT) |
    ↪ 2 | 1 | 2 | https://web.mit.edu | January 6, 2025 | USD 75 |
| Natural Sciences | University of Oxford | 3 | 3 | 1 | https://www
    ↪ .ox.ac.uk | October 15, 2024 | GBP 28.50 |
| Natural Sciences | University of Cambridge | 4 | 5 | 5 | https://
    ↪ www.cam.ac.uk | October 15, 2024 | GBP 60 |
| Natural Sciences | Stanford University | 5 | 6 | 6 | https://www.
    ↪ stanford.edu | January 5, 2025 | USD 90 |
| Social Sciences and Management | Harvard University | 1 | 4 | 3 |
    ↪  https://www.harvard.edu | January 1, 2025 | USD 85 |
| Social Sciences and Management | University of Oxford | 2 | 3 | 1
    ↪  | https://www.ox.ac.uk | October 15, 2025 | GBP 75 |
| Social Sciences and Management | Stanford University | 3 | 6 | 6
    ↪ | https://www.stanford.edu | January 5, 2025 | USD 90 |
| Social Sciences and Management | Massachusetts Institute of
    ↪ Technology (MIT) | 4 | 1 | 2 | http://web.mit.edu/ | January
    ↪ 6, 2025 | USD 75 |
| Social Sciences and Management | University of Cambridge | 5 | 5
    ↪ | 5 | https://www.cam.ac.uk | October 15, 2025 | GBP 50 |
```
```

