# OpenReview forum: "WideSearch: Benchmarking Agentic Broad Info-Seeking"
_ICLR.cc/2026/Conference — ICLR 2026 Poster_

### Official Review · Reviewer_tbLU · 2025-10-15

**Soundness:** 3
**Presentation:** 3
**Contribution:** 3
**Rating:** 6
**Confidence:** 4

**Summary:**

This paper introduces WideSearch, a benchmark for broad, multi-entity, verifiable information-seeking by LLM agents. Each task pairs a query with a predefined table schema.
The agent is required to 1) identify the complete entity set and 2) fill attributes for each entity using web tools.
A five-stage, human-in-the-loop curation pipeline ensures difficulty, verifiability, and temporal stability.
LLM-as-judge is used as an automated evaluator for semantically variable cells.
Experiments on a few models show very low table-level success, while item-level F1 can be much higher with test-time scaling.

**Strengths:**

1. Clearly scoped evaluation setting: The paper frames `wide information seeking` as distinct from general deep research style tasks, emphasizing breadth, completeness, and structured outputs rather than deep synthesis of a single topic.
2. Rigorous curation: The five-stage pipeline is thoughtful and practical for benchmark validity.
3. Hybrid automatic evaluation that mixes exact or approximate rules with LLM-as-judge, including column types and primary keys for deterministic alignment. This is a careful table-centric evaluation.

**Weaknesses:**

1. Although 18 topics are covered, the distribution still shows heavier representation in Business/Finance/Arts & Culture. A sensitivity analysis showing robustness of conclusions across domains/languages would be useful.
2. To the best of my knowledge, Manus is the only product claiming wide research. Is there any result from that system?

**Questions:**

1. Fonts are too small for Fig.1 and Fig. 2
2. In line 476, note that the left quote is wrong.
3. Too many numbers in Table 1. Could you bold important numbers?

---

> ### Author Response · Authors · 2025-11-16
>
> Thank you for taking the time to review our paper and providing valuable comments, which are of great significance for improving the quality of our paper!
>
> `Q1: Although 18 topics are covered, the distribution still shows heavier representation in Business/Finance/Arts & Culture. A sensitivity analysis showing robustness of conclusions across domains/languages would be useful.`
>
> The performance of different systems across various domains and languages is presented and analyzed in Appendix J and Figure J.1 of our paper. In general, Multi-agent consistently yields greater performance benefits compared to Single-agent, with essentially consistent conclusions across different domains and languages. This validates the consistency of other conclusions across diverse domains and languages: for instance, the WideSearch task involves a large number of parallel structures, making task decomposition and cross-validation extremely important for this task.
>
> Other conclusions: Overall, the performance in English is higher than that in Chinese. This may be attributed to the fact that English information is more mainstream on the Internet, allowing search systems to more easily access English information. Compared with existing benchmarks, the most fundamental characteristic of WideSearch lies in the simultaneous retrieval of a large amount of information. We believe this also indicates that WideSearch is the optimal benchmark for testing multi-agent systems.
>
> `Q2: To the best of my knowledge, Manus is the only product claiming wide research. Is there any result from that system?`
>
> Thank you for your suggestion. When we were building the benchmark, we had already paid attention to the Wide Research System proposed by Manus, and their focus is very similar to ours. The number of daily uses of a free Manus account is quite limited, and we have not been approved for funding to purchase a Manus membership account. Therefore, it is not very practical to manually test their system.
>
> `Q3: Font, punctuation errors, and issues with table display`
>
> Thank you for your careful observation, which has helped us enhance the clarity of our paper's presentation.
>
> We have adjusted the font as much as possible to ensure clearer display.
>
> We have revised the left quote.
>
> For Table 1, we have made modifications: the highest values are emphasized in bold, and the second-highest values are emphasized with underlines.
>
> We would like to thank you again for your valuable suggestions. If you have any further questions, please feel free to discuss them with us at any time.

---

> > ### Author Response · Authors · 2025-11-27
> >
> > Thank you again most sincerely for reviewing our manuscript. Please feel free to raise any further questions you may have and discuss them with us.

---

### Official Review · Reviewer_29pm · 2025-10-28

**Soundness:** 3
**Presentation:** 3
**Contribution:** 3
**Rating:** 6
**Confidence:** 4

**Summary:**

The paper introduces WideSearch, a benchmark targeting “agentic broad information-seeking” tasks where an LLM agent must find a complete set of entities and fill a structured table from the live web (e.g., “all DOJ attorney openings in a date window”). The dataset contains 200 manually curated questions (100 EN / 100 ZH) spanning 15 domains, built with a five-stage curation pipeline to ensure breadth, verifiability, and temporal stability, and scored with a hybrid, table-alignment evaluation (rule checks + LLM-as-judge). Benchmarking 10 systems shows very low table-level success (most ≈0%; best single/multi-agent ≈7%), while humans achieve ~20% in single-annotator mode; the gap stems from strict completeness requirements rather than difficulty of single facts (item-level F1 can be high with retries). Multi-agent frameworks improve F1 via divide-and-conquer but still rarely achieve perfect tables, highlighting deficiencies in planning, reflection, and evidence use.

**Strengths:**

Overall, it's a very good work that extends the frontiers of evaluating newly emerged deep research systems and tools.

1. This paper proposes a novel dataset aimed at breadth and completeness across many entities. Rigorous five-stage human-centered curation ensures realistic and verifiable tasks.

2. Results expose a fundamental limitation of current agents—completeness at scale—and show multi-agent setups help but don’t solve it, offering a concrete target for future research.

**Weaknesses:**

1. The evaluation protocol might not be robust enough. This paper uses markdown format as a protocol, applies several fuzzy or exact matches to number/date/urls, and finally applies LLM-as-a-judge to evaluate complex answers. This paradigm replies on pre-crafted ground truths and would be sensitive to those queries whose answer might change with time (for example, the top-selling electrical toothbrush in the past week).

2. The current success criterion may be too brittle, which requires perfect table equality; small, arguably negligible deviations lead to failure, which can conflate usability with strict exactness.

3. Tasks are claimed “temporally invariant”, yet the benchmark relies on the live web; concrete procedures for refreshing ground truths and handling site changes are under-specified.

**Questions:**

1. What exactly do the source queries come from? Are they purely curated by human annotators or modified from existing data sources (like NaturalQuestions)? Are there any time-sensitive queries (for example, the top-selling electrical toothbrush in the past week)?

---

> ### Author Response · Authors · 2025-11-16
>
> Thank you for taking the time to review our paper and providing valuable comments, which are of great significance for improving the quality of our paper!
>
> `Q1: The evaluation protocol might not be robust enough. This paper uses markdown format as a protocol, applies several fuzzy or exact matches to number/date/urls, and finally applies LLM-as-a-judge to evaluate complex answers. This paradigm replies on pre-crafted ground truths and would be sensitive to those queries whose answer might change with time (for example, the top-selling electrical toothbrush in the past week).`
>
> Our evaluation system is a hybrid one, where the rule-based method and the LLM-as-judge approach operate independently. For certain columns, the rule-based method (such as exact match) is adopted, while for other columns, the LLM-as-judge approach is used.
>
> We ensure that the questions we design do not change over time, which is mentioned in Line 154. The issue you raised and the examples you provided will not occur. For some tasks, we guarantee the certainty and stability of results by clearly defining their time scopes. Examples like the one you mentioned—"the top-selling electrical toothbrush in the past week"—will not be included in our benchmark. In accordance with our principles, we might rephrase the query you mentioned into something like "a set of information (xxx) about the top 10 best-selling electric toothbrushes in terms of total sales across all Walmart Supermarkets in New York City during July 2018".
>
> `Q2: The current success criterion may be too brittle, which requires perfect table equality; small, arguably negligible deviations lead to failure, which can conflate usability with strict exactness.`
>
> We have incorporated compatibility considerations into many of our evaluations. For instance, numerical values are allowed to have a certain degree of error (e.g., 5%); proper nouns such as personal names and place names are mapped using LLMs (for example, "Beijing Daxing International Airport" and "Daxing Airport" are considered consistent). For cells containing multiple pieces of atomic information, we inform the LLM-judge by means of annotating criteria that an answer will be deemed correct if it captures part of the ground truth. For more complex scenarios, we have annotated the criteria for the LLM-as-judge. Examples of these criteria can be found in the cases provided in Appendix C. Our evaluation method is compatible with minor discrepancies.
>
> At the same time, we aim to emphasize that completeness and reliability are actually crucial characteristics of LLM-based search systems, and these attributes are also of great importance for certain specific data analysis tasks. On the other hand, among our metrics, only the success rate requires absolute accuracy. The row-level and item-level F1 scores can serve as fine-grained performance indicators to reflect the degree of task completion, and their values are typically highly continuous.

---

> > ### Author Response · Authors · 2025-11-16
> >
> > `Q3: Tasks are claimed “temporally invariant”, yet the benchmark relies on the live web; concrete procedures for refreshing ground truths and handling site changes are under-specified.`
> >
> > Our answers have undergone repeated verification by multiple people through multiple rounds, and the corresponding information points in the answers barely change over time. For instance, the task examples provided in Appendix C—such as the locations of Taylor Swift's concerts from May 2010 to May 2025—are established facts that will not alter. Facts of this kind typically remain unchanged regardless of changes to the web or individual sites. Therefore, for our task, the ground truth table does not require refreshing.
> >
> > There was a two-month interval between the initial version and the final confirmed version of this benchmark. During these two months, numerous revisions were made, most of which were prompted by unclear descriptions in the task queries. By continuously refining and specifying the restrictive information in the queries, we ensured that all information in the answers is objective facts that do not change with time.
> >
> > `Q4: What exactly do the source queries come from? Are they purely curated by human annotators or modified from existing data sources (like NaturalQuestions)? Are there any time-sensitive queries (for example, the top-selling electrical toothbrush in the past week)?`
> >
> > We conducted user research on different AI products, analyzed search-related queries, and drew insights from the real search queries made by users in these actual AI systems. Subsequently, we asked annotators to design questions based on this foundation. Notably, WideSearch's data does not utilize any existing datasets, and all its queries are time-invariant.
> >
> >
> > We would like to thank you again for your valuable suggestions. If you have any further questions, please feel free to discuss them with us at any time.

---

> ### Comment · Reviewer_29pm · 2025-11-19
> **Response to Authors' Rebuttal**
>
> Thanks for the detailed response and clarification. Overall, it's a good and timely work for benchmarking agentic search and deep research systems. I'll stand for my score.

---

> > ### Author Response · Authors · 2025-11-27
> >
> > Thank you for your response. Your valuable comments have significantly improved the quality of our paper. We welcome any further questions you may have and encourage you to discuss them with us at any time.

---

### Official Review · Reviewer_dPH5 · 2025-11-01

**Soundness:** 2
**Presentation:** 2
**Contribution:** 2
**Rating:** 4
**Confidence:** 5

**Summary:**

The paper introduces WideSearch, a benchmark designed to evaluate search agents on broad information-seeking tasks, focusing on collecting atomic factual data across many entities that are objectively verifiable. The benchmark comprises 200 manually curated tasks (100 English, 100 Chinese). It also evalutes a broad set of LLMs and the LMs under agent frameworks.

**Strengths:**

The benchmark is unique from previous ones, which extend the commonly used browsecomp-style simple-answer questions to broad information gathering.

**Weaknesses:**

- Most importantly, even though the paper's motivation is very clear and reasonable, it remains a straightforward extension to the existing browsecomp-style benchmarks (from simple fact to a list of fact). Despite its potential usefulness (given the success of browsecomp), it's hard to admire from a research perspective. Also, it's still constrained to this very specific type of tasks, for the convenience of evaluation.

- Regarding the experiments:
  - 1) the tasks appear to be too challenging for simple LLMs (with searches) because they usually only execute very limited search steps. This makes related analysis trivial to know. And it's also not necessary to call them as "single agent"
    2) The tasks appear to be more suitable for Deep Research agents such as Deep Research from OpenAI/Gemini/Grok/Qwen/Kimi/.... But according to the content in the appendix, the evaluted systems are more or less from self-implemented agent scaffoldings, while leaving those important frontier deep research systems omitted? The paper could be much more benefited by including experiments on more frontier agents and their analysis.

- Writing and experiments: might as well include more details in the main content.

**Questions:**

- It's suggested to include a table of comparison to the recent deep research benchmarks on key dimensions.
- The paper would be benefited by including more evaluated agents, and at least OpenAI Deep Research, which is usually the leading agent across recent benchmarks.

---

> ### Author Response · Authors · 2025-11-16
>
> Thank you for taking the time to review our paper and providing valuable comments, which are of great significance for improving the quality of our paper!
>
> `Q1: it remains a straightforward extension to the existing browsecomp-style benchmarks (from simple fact to a list of fact). Despite its potential usefulness (given the success of browsecomp), it's hard to admire from a research perspective. Also, it's still constrained to this very specific type of tasks, for the convenience of evaluation.`
>
> What we want to emphasize is that the motivation behind building our benchmark stems from the genuine search intentions reflected in the queries of real users when using AI applications. The "A list of fact" you mentioned is merely a superficial feature. What WideSearch aims to address is the problem of "I could do it, but the sheer volume is overwhelming". We will demonstrate the value of WideSearch through comparisons as follows:
>
> BrowseComp is solely designed to test the capabilities of Large Language Models (LLMs) in terms of deep reasoning and multi-turn tool calling for long-chain tasks. The distribution of task queries in BrowseComp is completely different from that of real users' queries. Moreover, the difficulty level of these tasks is so high that real users of AI applications rarely have such information-seeking needs. For instance, here is an example from BrowseComp:
>
> "Give me the title of the scientific paper published in the EMNLP conference between 2018-2023 where the first author did their undergrad at Dartmouth College and the fourth author did their undergrad at University of Pennsylvania. (Answer: Frequency Effects on Syntactic Rule Learning in Transformers, EMNLP 2021)"
>
> In contrast, WideSearch is intended to reflect real users' authentic demands for searching large volumes of information. Based on our online research, more often than not, users do not aim to find an extremely hard-to-locate piece of information (like the example in BrowseComp). Instead, they want to use AI tools to reduce a large amount of manual and repetitive work.
>
> For example, when a parent is choosing a school for their child, they may need to refer to the admission requirements of target schools over the years. While individual pieces of information are not difficult to find, collecting and summarizing admission information from previous years requires browsing various official websites multiple times and takes a significant amount of time. For a data analyst, there may be occasions where they need to analyze data from various competing companies; in such cases, they must browse official websites and financial report data repeatedly. These are all highly realistic user needs.
>
> Regarding your comment that "it's still constrained to this very specific type of tasks", we have always believed that searching is one of the most fundamental needs of humans when using the Internet, and research on it has persisted for decades without fading. What our WideSearch can achieve is redirecting the academic community's attention—currently focused on synthetic deep search tasks similar to BrowseComp—toward the real search needs of users. Deep search tasks like BrowseComp are only used to test the capabilities of models and deviate far from the distribution of real queries. In contrast, WideSearch focuses on how to transform LLMs into genuine productivity tools for searching, which is towards real-world scenario.

---

> > ### Author Response · Authors · 2025-11-16
> >
> > `Q2: the tasks appear to be too challenging for simple LLMs (with searches) because they usually only execute very limited search steps. This makes related analysis trivial to know. And it's also not necessary to call them as "single agent"`
> >
> > We believe that the proposal of WideSearch is precisely intended to test the capability when a large number of relevant search behaviors are involved, as well as to challenge the ability of Large Language Models (LLMs) in scenarios requiring extensive information retrieval. Currently, the overall success rate appears to be very low, which precisely indicates the lack of this crucial capability—relevant research is therefore needed to enhance the WideSearch capability of LLMs. Meanwhile, based on more fine-grained performance analysis, we can observe that the item-level F1 score is not particularly low; in the context of test-time scaling, its max@4 score even approaches 80. This implies that most atomic information points have been correctly identified. It further suggests that the success rate places greater emphasis on the completeness and reliability of information retrieval, which is exactly what current systems lack.
> >
> > Both our single-agent and multi-agent frameworks are constructed manually. The multi-agent mode enables explicit task decomposition, autonomously creates sub-agents, distributes subtasks to these sub-agents, and allows the sub-agents to return relevant information to the main agent after completing their tasks. Notably, each sub-agent has an independent context window, which is isolated from the main agent. We consider this design necessary for large-scale information retrieval: this mode can reduce the information load of each agent, lower the difficulty for each agent to complete its tasks, decrease the possibility of exceeding the context length limit of LLMs, and increase the success probability of identifying each atomic information point. Additionally, due to multi-task parallelism, it also improves the efficiency of completing the overall task. The single-agent mode here corresponds to our multi-agent mode—it only has one set of LLM context windows. As more information is processed within a single context window, there is a higher likelihood of exceeding the length limit and a greater tendency for errors to occur.
> >
> > `Q3: The tasks appear to be more suitable for Deep Research agents such as Deep Research from OpenAI/Gemini/Grok/Qwen/Kimi/.... But according to the content in the appendix, the evaluted systems are more or less from self-implemented agent scaffoldings, while leaving those important frontier deep research systems omitted? The paper could be much more benefited by including experiments on more frontier agents and their analysis.`
> >
> > In the DeepResearch agents mentioned by you, such as those from OpenAI and Gemini, we initially conducted experiments on these DeepResearch systems. Unfortunately, the current DeepResearch systems seem to have been specifically optimized for Report Generation. They automatically search for a large amount of information, but when outputting results, they generate a complete analysis report—breaking down our task into different subtasks and scattering relevant information throughout the entire report. No matter how we modify the instructions, these systems cannot output a complete Markdown table in its entirety, which is essential for our automated evaluation.
> >
> > We have attempted to manually consolidate the information into a single comprehensive table, but this workload is extremely heavy and nearly impossible to complete. Furthermore, due to the inconsistent formats of the information from different systems, it is impossible to fairly reflect the capabilities of various DeepResearch systems. We attribute this issue to the over-optimization of DeepResearch systems, which prevents them from adhering to basic instruction-following requirements regarding format (e.g., outputting only relevant data in Markdown format).
> >
> > Our solution is as follows: On the official web interfaces of companies including OpenAI, Gemini, and Claude, we use the web search function (not the DeepResearch system). Annotators manually input task queries and extract relevant responses. To ensure a fair test of these web-based systems' capabilities, we require annotators to use multi-turn prompting or re-enter the task when the system refuses to answer or outputs content in an incorrect format. This process continues until the system produces a response in the correct format, which is then stored. The relevant data is presented in Table 1, and the corresponding analysis is provided in Section 3.2.

---

> > > ### Author Response · Authors · 2025-11-16
> > >
> > > `Q4: It's suggested to include a table of comparison to the recent deep research benchmarks on key dimensions.`
> > >
> > > Thank you for your comments, which will help enhance the completeness of our paper.
> > >
> > > In the revised version of the paper, we have added a more detailed comparison table, which is presented in Table 5 of Appendix E.
> > >
> > > `Q5: The paper would be benefited by including more evaluated agents, and at least OpenAI Deep Research, which is usually the leading agent across recent benchmarks.`
> > >
> > > As mentioned in Q3, we have already conducted tests. Typically, they cannot generate a complete Markdown table containing all data properly. If we directly evaluate their generation results, the performance will be extremely low, so it is not meaningful for analysis. We tested their official LLM (with web browse) system, and it remains a high-performance end-to-end deep research system.
> > >
> > > We would like to thank you again for your valuable suggestions. If you have any further questions, please feel free to discuss them with us at any time.

---

> ### Comment · Reviewer_dPH5 · 2025-11-17
>
> Thank you for the authors’ detailed responses. After carefully reading all replies, I must note that most of my original weaknesses and concerns remain largely unaddressed, and they are very fundamental issues of the work.
>
> ## Regarding Q1
>
> 1. **Novelty concerns remain:** While I fully understand the distinction the authors draw between WideSearch and BrowseComp, I still do not find the benchmark to introduce substantial novelty from a research perspective. Although the authors claim that “wide search” represents a more realistic and common need, **the evaluation and benchmark creation themselves offer limited innovation** among **the recent surge of search benchmarks**. In fact, the tasks appear comparatively easier to construct, and the benchmark benefits from pre-specified answer structures that make evaluation convenient (which may not be a good thing, and will be discussed in the response related to Q3).
>
> 2. **Scope and task nature is overly narrow and repetitive:** The authors’ response also raises an additional concern: **the inherent restriction in the scope and nature of the tasks**. It is true that WideSearch focuses on wide rather than deep search, which is a reasonable design choice and represents some  realistic needs. However, this design simultaneously narrows the diversity of task types and the capabilities being tested. If the benchmark mostly consists of large-scale but structurally similar lookup tasks, it becomes easy that **a model trained specifically on this pattern could saturate the benchmark quickly**, limiting its long-term research value (which is actually a common issue of some benchmarks).
>
> ## Regarding Q2
>
> I did not have major concerns with this point originally. My point was simply that **it is expected for basic LLMs (most of which are primarily chat models) augmented with search tools to underperform on tasks requiring large-scale information aggregation**. Evaluating them on WideSearch is therefore unsurprising and arguably of limited value. (But models that explicitly optimized for tool use or search could be a good fit, such as Kimi k2, claude, glm or tongyi's search models)
>
> Additionally, I must note that prior to reading the appendix, I did not clearly understand how the “single agent” was implemented. **The main text should provide at least a minimal description of what tools the single agent can use and what its agent loop looks like, rather than merely listing the models.**
>
> ## Regarding Q3
>
> **The inability to evaluate frontier search agents such as OpenAI Deep Research and Gemini remains, in my view, a major flaw, and it reflects insufficient consideration of evaluation and exp design.**
>
> 1.	**A two-stage evaluation seems both feasible and natural?**
> Even if these systems output long-form analyses or fixed-format reports (e.g., Gemini’s multi-section documents), why not adopt a two-stage approach?
> 	•	First, allow the system to output in its preferred format.
> 	•	Second, use a frontier LLM or itself to summarize or convert that output into the required Markdown table.
> This process mirrors realistic usage patterns: users of ChatGPT probably won't bother to issue an additional instruction such as “Please summarize the above into a Markdown table.” The authors’ justification that manual consolidation is too costly does not fully address why an automated LLM-based consolidation was not considered. If the concern will be about the evaluating of “instruction-following,” the authors can simply report both forms of evaluation (raw output and two-stage processing), or leave some notes there, as several recent benchmarks do.
>
> 2.	**Overly strict formatting requirements reduce applicability and exclude real systems.**
> The benchmark’s  structured Markdown design, despite being convenient for the evaluation , directly prevents testing the most widely used  search agents. This is a serious limitation for a benchmark. By comparison, recent benchmarks such as Mind2Web 2, LiveResearchBench (I remember there are a few others) rely on LLM-based evaluators to assess "wide" or multi-faceted answers and do not impose rigid Markdown entity tables. This design choice allows them to evaluate real-world search systems without forcing rigid output constraints.

---

> ### Author Response · Authors · 2025-11-18
>
> We express great thanks to you for responding us. It helps a lot for our paper!
>
> `Q1:`
>
> **On the innovation of the benchmark construction method** :
>
> Our benchmark is entirely human-annotated. As shown in Figure 3 of our paper, we have established a strict five-stage construction and validation process. Our data sources are diverse, including 18 different real-world domains from both Chinese and English. **It is particularly noteworthy that existing research on LLM-as-judge shows that there is often inconsistency between human and LLM evaluations. Therefore, our annotation pipeline is specifically optimized for this issue**. In Stage 5, based on the current version of the task, we use the agent framework to generate an inference-predicted answer table. This table is then evaluated both by our automatic evaluation system and by human annotators. Based on these two evaluation results, we assess the consistency between the automatic evaluation system and human evaluations (you can think of it as a form of iterative meta-evaluation, i.e., evaluating the “evaluation system”). If the consistency between the two evaluation systems is not sufficiently high, we modify and re-annotate the task until the consistency between the automatic evaluation system and human evaluations reaches a satisfactory level. Because of this, we annotated the evaluation methods for different columns, which ensures that our evaluation system is both objective and fair.
>
> In addition, I’m sorry, I don’t fully understand the point you raised. If you’re referring to the dataset construction method, based on my understanding, there are generally four main methods for dataset construction: 1. Specific modification of existing data sources; 2. Fully human-annotated; 3. Fully LLM-generated according to instructions; 4. Mixed human and LLM annotation. These methods have been extensively studied in the academic community, and both Browsecomp and DeepResearch Bench are human-annotated. Generally, high-quality, widely tested benchmarks are manually annotated in detail by humans. If you are referring to the details of the annotation pipeline itself, I believe that the optimizations we made regarding the consistency between the "automatic evaluation system and human evaluation" are unique and innovative.
> On the innovation of the benchmark evaluation method:
>
> Current methods are typically either entirely rule-based (Exact Match) or based on LLM-as-judge. Our evaluation system is a hybrid system. We have annotated all columns, some evaluated based on rules, and others based on LLM-as-judge. We did this to ensure the reliability and robustness of the evaluation system as much as possible. For comparison, Browsecomp is entirely rule-based, and DeepResearch Bench is fully LLM-based. The evaluation in DeepResearch Bench requires dynamically generated evaluation criteria and corresponding weights, which we believe makes the evaluation unstable and inconsistent. As mentioned in Section 4.3.2 of their paper, the Pairwise Agreement Rate metric reflects this, where their evaluation framework only has 71.33% consistency with human evaluation, even after optimizing the prompts, with a vanilla prompt consistency of only 58.89%. The Mindweb system you mentioned below is fully LLM-based, with its evaluation framework consisting of an LLM-based extractor and verifier, requiring LLMs to extract information and conduct relevant evaluations. As for the LiveResearchBench you mentioned, we reviewed the paper, and its evaluation system is also LLM-as-judge. In Section 4, it’s mentioned that different LLMs show varying levels of consistency with human evaluations. For example, for their "Presentation & Organization" task, GPT-5’s consistency with human evaluation is only 85.0%. I don’t think that’s a strong number. Furthermore, I don’t think we should discuss the LiveResearchBench paper as it was published on arXiv in October 2025, which is after the ICLR 2026 paper submission deadline. Our automatic evaluation system has a high consistency of 97% with human evaluations. Additionally, our view is that rules are generally more reliable than LLM-as-judge, like the method used in Browsecomp, and rule-based evaluation is more efficient as it doesn’t require waiting for LLM outputs. However, some outputs cannot be covered by rules, so a certain amount of LLM-as-judge is necessary. Balancing these two aspects is an important issue. Based on this, we have used a repeatedly iterated hybrid evaluation system (rule + LLM-as-judge verifier), and we’ve annotated appropriate evaluation methods for each column. I believe this innovation is significant and non-trivial.

---

> > ### Author Response · Authors · 2025-11-18
> >
> > **On the point of "Scope and task nature is overly narrow and repetitive"**:
> >
> >  We don’t agree with this statement. Our dataset includes both Chinese and English, covers 18 domains, and comes from real-world scenarios. I believe this sufficiently demonstrates the broad scope of tasks in our benchmark.
> >
> > **On the statement "It is true that WideSearch focuses on wide rather than deep search, which is a reasonable design choice and represents some realistic needs. However, this design simultaneously narrows the diversity of task types and the capabilities being tested"**:
> >
> >  We don’t quite agree with this claim. Following this logic, the Browsecomp and other deep search test sets should also have narrow scopes because they only test deep search, but that’s not the case.
> >
> > **On the statement "If the benchmark mostly consists of large-scale but structurally similar lookup tasks, it becomes easy that a model trained specifically on this pattern could saturate the benchmark quickly, limiting its long-term research value"**:
> >
> >  We are unsure about the evidence supporting this conclusion. If you have any, please feel free to share it with us. We don’t agree with this claim. The current test results (including test-time scaling results) show that current model performance still has a lot of room for improvement, with success rates still far from optimal. WideSearch emphasizes reliability and completeness, and there’s still significant room for improvement. Moreover, in our ongoing work, we’ve found that improving performance on this training set is challenging and requires considerable effort.
> >
> > We believe that the WideSearch benchmark is extremely important for the following cutting-edge research topics and is a natural testing benchmark:
> >
> > 1. Context memory management: Recent research has seen an explosion in context memory management studies, but these have mostly been tested in deep search frameworks. They are not yet mature, but context management is naturally suited for testing on WideSearch. This is because WideSearch requires gathering large amounts of relevant information and eliminating redundant information from web pages. Tasks with more than 5000 information points and tasks with more than 100 web pages are involved. An agent may need to search far more pages than this.
> >
> > 2. Multi-agent scaling: Deep search tasks like Browsecomp often require information with dependencies, so the benefits of scaling agents are primarily seen in exploring different paths and making multiple attempts at a task. In contrast, WideSearch includes more parallel substructures. Studying multi-agent scaling and related RL algorithms on this benchmark will highlight more benefits of multi-agent systems, especially in terms of efficiency.
> >
> > We believe that the main innovation of the benchmark lies in exploring the boundaries of model capabilities, focusing on abilities that previous benchmarks have not tested, and providing insights for future model development. For example:
> >
> > - Browsecomp tests the ability of long-chain deep search, where the answer is a difficult-to-find entity.
> >
> > - DeepResearch Bench tests the ability of a model to generate research reports, where the answer is a subjective report that’s difficult to evaluate, and the evaluation framework is unstable.
> >
> > - WideSearch tests the ability of models to perform breadth-first searches, where the answer is a set of relevant atomic information. This is completely different from deep search. It focuses on completeness and reliability in search. We believe this ability has been widely overlooked by previous systems, and it is worth emphasizing. It is also one of our future development goals.

---

> > > ### Author Response · Authors · 2025-11-18
> > >
> > > `Q2:`
> > >
> > > **On your statement "it is expected for basic LLMs (most of which are primarily chat models) augmented with search tools to underperform on tasks requiring large-scale information aggregation"**:
> > >
> > >  We’ve tested several closed-source state-of-the-art reasoning models like GPT-5, O3, and Gemini. They have strong tool-calling capabilities, and they are not just simple chat models. You can review their official benchmark results. Our results also include models like Kimi K2 and Claude, which you mentioned. Furthermore, this baseline focuses more on the fundamental capabilities of the model. We don’t want the complex agent scaffold to introduce too many human priors.
> > >
> > > Our single-agent system is essentially a ReAct system. It can call two tools: search and web browsing. It’s a loop of tool-calling: thinking, calling a tool, getting feedback, and continuing until no tool is needed, at which point the final answer is output.
> > >
> > >
> > > `Q3:`
> > > **Regarding your suggestion of "A two-stage evaluation seems both feasible and natural"**:
> > >
> > >  Thank you for suggesting this method. The two-stage approach you propose is indeed feasible, and we may try this method in future versions of our paper. However, unfortunately, at this stage, we don’t have the manpower or bandwidth to collect responses from the various Deep Research systems. This task is very time-consuming and requires multiple pro accounts and annotators. In our initial experiments, we asked annotators to download reports from web pages. For instance, with Gemini, generating a report takes a very long time, and there are often errors where the system doesn’t return anything. We hope you can understand our difficulties. Additionally, the annotators we employed were temporary, and they are no longer working with us.
> > >
> > > **On your comment "Overly strict formatting requirements reduce applicability and exclude real systems"**:
> > >
> > >  We don’t consider this a problem. Consistency and stability in evaluation are very important. Our formatting requirements are reasonable and help our evaluation. Moreover, we disagree with your statement that "directly prevents testing the most widely used search agents." Our benchmark can test deep research systems, but without human intervention, their performance is too poor to be meaningful in an experiment. However, excessive human intervention (including the two-stage method you mentioned) or failure to generate results according to specified instructions indicates that the system is not sufficiently intelligent. Our research is focused on building general artificial intelligence, and currently, these agents cannot even follow simple instructions like "output in markdown format." This represents a step back in intelligence, contrary to the goal of researching intelligence. Even small 7B models can follow such instructions, but these deep research systems cannot. This is a typical case of over-optimized systems, where they likely contain a highly fine-tuned agent scaffold and detailed system prompts. We believe our benchmark reveals this phenomenon, which is also a contribution of WideSearch.
> > >
> > > Additionally, we tested end-to-end systems such as OpenAI/Gemini/Claude’s web versions, which are also real systems, just not overly optimized deep research systems.
> > >
> > > Lastly, the two benchmarks you mentioned, Mind2Web 2 and LiveResearchBench, we have already discussed above. Their evaluations are not stable or consistent.

---

> ### Comment · Reviewer_dPH5 · 2025-11-18
>
> Thank you for the follow-up responses. I appreciate the efforts the authors have put into both the work and the rebuttals. That said, it seems we ultimately differ on several key points, and I will therefore keep my negative score.
>
> One minor clarification: my mention of Mind2Web 2 and LiveResearchBench was intended to provide additional context on concurrent developments in this area and on the points discussed there (and espeicially given the surge of works in this field), not to imply that the authors must include them in the manuscript (though discussing recent or concurrent efforts is still encouraged).

---

> > ### Author Response · Authors · 2025-11-27
> >
> > Thank you for your response. Your valuable comments have significantly improved the quality of our paper. We welcome any further questions you may have and encourage you to discuss them with us at any time.

---

### Official Review · Reviewer_kg32 · 2025-11-04

**Soundness:** 4
**Presentation:** 3
**Contribution:** 3
**Rating:** 8
**Confidence:** 3

**Summary:**

This paper introduces WideSearch, a benchmark designed to evaluate the reliability and comprehensiveness of LLM-based search agents in large-scale information-gathering tasks. WideSearch assesses agents on tasks requiring exhaustive and structured collection of numerous atomic facts across multiple entities. The benchmark consists of 200 manually curated queries (100 English, 100 Chinese) spanning 18 diverse domains, each paired with a well-defined schema and gold-standard table.

A five-stage human-in-the-loop curation pipeline ensures that all tasks are complex, objectively verifiable, and reliant on real web tool usage. The evaluation framework integrates rule-based scoring with LLM-as-a-judge scoring, achieving over 97.8% agreement with human judgment.

The authors further perform an extensive benchmarking study across state-of-the-art agentic systems, revealing severe limitations in current models—particularly in maintaining completeness and fidelity at scale—highlighting the urgent need for more advanced planning, reflection, and evidence-grounding capabilities in next-generation agents.

**Strengths:**

(1) The paper tackles an underexplored yet practically crucial dimension of agent evaluation—broad, high-fidelity information gathering—which complements existing reasoning- and synthesis-oriented benchmarks. The conceptual framing of WideSearch as the “breadth” counterpart to DeepSearch and DeepResearch is clear, coherent, and well-motivated.

(2) The five-stage human-in-the-loop curation pipeline is rigorous and systematic, ensuring that all tasks are complex, verifiable, and genuinely dependent on real web search. The inclusion of both English and Chinese queries further enhances the benchmark’s generality and cross-lingual coverage.

(3) The evaluation and analysis are comprehensive and insightful. The experiments benchmark over ten single-agent, multi-agent, and end-to-end commercial systems using multiple aggregation metrics (Avg@N, Pass@N, Max@N), providing a well-rounded assessment of agent reliability at scale. The error taxonomy—separating advanced agentic failures (planning, reflection, evidence use) from basic operational errors—is particularly insightful. The test-time scaling analysis further isolates completeness, rather than retrieval accuracy, as the key bottleneck, offering a concrete and data-driven direction for future work.

**Weaknesses:**

The benchmark construction and evaluation are solid in general. It would be great if the authors could further conduct quantitative analysis regarding the major challenges and failure patterns mentioned in sections 4.1 and 4.2.

**Questions:**

(1) The paper reports 97.8% agreement between LLM-as-judge and human evaluation, which is strong. Could you provide more details on the specific disagreement cases? It would be great if the authors could provide more insights into this.

(2) The benchmark relies on live web data. How do you plan to ensure its stability over time? Is there a plan for versioning or periodic re-validation as the web content evolves?

---

> ### Author Response · Authors · 2025-11-16
>
> Thank you for taking the time to review our paper and providing valuable comments, which are of great significance for improving the quality of our paper!
>
> `Q1: It would be great if the authors could further conduct quantitative analysis regarding the major challenges and failure patterns mentioned in sections 4.1 and 4.2.`
>
> R1: Your suggestions are very insightful. When we completed the initial experiments and the first draft of the paper, we organized five annotators to manually check the reasoning trajectories of different models, and summarized 4 failure modes of advanced agentic capabilities and 4 basic failure modes.
>
> Among them, regarding the "CHALLENGES IN ADVANCED AGENTIC CAPABILITIES" mentioned in Section 4.1, almost all failure cases exhibited these four failure behaviors. Therefore, we did not add an explicit quantitative analysis in the first draft.
> For the "BASIC FAILURE MODES" mentioned in Section 4.2, we re-statistically analyzed the distribution of failure modes of different models under different agent modes, as shown below:
>
>
> Single-Agent Mode Error Distribution
>
> | Model | Tool Call Error | Context Length Exceedance | Output Format Error | Response Refusal |
> | --- | --- | --- | --- | --- |
> | deepseek-r1 | 4.12% | 0.12% | 0.50% | 3.00% |
> | doubao-1.6 | 3.62% | 12.62% | 0.25% | 1.62% |
> | doubao-1.6-non-thinking | 4.00% | 3.88% | 0.25% | 1.12% |
> | gcp-claude4-sonnet-thinking | 0.62% | 1.25% | 0.00% | 1.50% |
> | gemini-2.5-pro | 3.00% | 0.00% | 0.38% | 7.25% |
> | gpt-5-medium | 13.12% | 6.75% | 0.00% | 10.75% |
> | k2 | 0.50% | 0.50% | 0.38% | 2.38% |
> | o3-high_medium | 1.88% | 0.38% | 0.00% | 16.00% |
> | qwen3-235b-a22b-thinking-2507 | 1.50% | 0.50% | 2.00% | 0.25% |
>
> Multi-Agent Mode Error Distribution
>
> | Model | Tool Call Error | Context Length Exceedance | Output Format Error | Response Refusal |
> | --- | --- | --- | --- | --- |
> | deepseek-r1 | 22.75% | 0.00% | 0.00% | 4.00% |
> | doubao-1.6 | 29.88% | 4.25% | 0.12% | 1.38% |
> | doubao-1.6-non-thinking | 9.88% | 0.75% | 0.25% | 1.12% |
> | gcp-claude4-sonnet-thinking | 9.12% | 0.25% | 0.00% | 0.88% |
> | gemini-2.5-pro | 6.75% | 0.00% | 0.00% | 3.38% |
> | gpt-5-medium | 23.75% | 1.38% | 0.00% | 10.62% |
> | k2 | 1.62% | 0.00% | 0.38% | 0.38% |
> | o3-high_medium | 4.38% | 0.62% | 0.00% | 10.12% |
> | qwen3-235b-a22b-thinking-2507 | 6.00% | 0.50% | 0.75% | 0.12% |
>
> As can be seen from the data, the proportion of tool call errors is relatively high. After analyzing the cases, we found that most of the failures are caused by either the failure to generate parameters correctly or the generation of incorrect tool call formats.
> In the single-agent mode, some thinking models have a relatively high proportion of Context Length Exceedance due to overthinking.
>
> The proportion of response refusal is relatively high in OpenAI's models and Gemini's models. When these models consider the current task to be relatively difficult, they tend to give up directly rather than providing a partially correct answer.

---

> ### Author Response · Authors · 2025-11-16
>
> `Q2: Could you provide more details on the specific disagreement cases? It would be great if the authors could provide more insights into this.`
>
> We have provided an example of LLM-Judge in Appendix G to help you understand the entire process more easily.
>
> The point you raised regarding misalignment is highly insightful. This is an unavoidable pain point in the current field of LLM-as-Judge. When we initially completed our evaluation framework, we identified cases where the framework’s results were inconsistent with human evaluations. Typically, such inconsistencies arise from discrepancies between the criteria used by humans and those adopted by the model. To address this, we opted for manual rubric annotation to align the model’s judgments with human judgments. Through multiple rounds of annotation, we have continuously improved the consistency between model evaluations and human evaluations. However, there are still some unresolved corner cases, as mentioned in the revised version of our paper:
>
> For example, regarding ”carlosslimhelu family” and ”carlosslimhel´u”, human annotators consider them consistent and believe the score should be 1. However, when the LLM conducted the evaluation, it output: “Standard answer: ‘carlosslimhelufamily’; Answer provided: ‘carlosslimhel´u’. The word ‘family’ is missing, making it insufficient to determine factors such as children. Score: 0”. In the current practice of LLM-Judge, such cases are usually unavoidable.
> If we aim to further enhance consistency, the only approach is to annotate more fine-grained rubrics tailored to the specific characteristics of each category.
>
> `Q3: The benchmark relies on live web data. How do you plan to ensure its stability over time? Is there a plan for versioning or periodic re-validation as the web content evolves?`
>
> As stated in the task construction principles mentioned in our paper, when formulating questions, we consider knowledge that remains unchanged over time. For instance, take the example provided in Appendix C of the paper: "Could you list every single concert on Taylor Swift’s tour from January 1, 2010, to May 1, 2025, including the specific date, the concert’s English name, the country, the city, and the venue?"
>
> The answer to this question is based on established facts that have already occurred and will not change with the passage of time.
>
> Yes, we plan to regularly maintain and verify the accuracy of the entire dataset.
>
>
> We would like to thank you again for your valuable suggestions. If you have any further questions, please feel free to discuss them with us at any time.

---

> > ### Author Response · Authors · 2025-11-27
> >
> > Thank you again most sincerely for reviewing our manuscript. Please feel free to raise any further questions you may have and discuss them with us.

---

### Author Response · Authors · 2025-11-16
**General response to reviewers, ACs, SACs, and PCs**

In response to the specific comments from the reviewers, we have provided corresponding answers and made certain revisions to the paper. The revised parts are highlighted in blue, as detailed below:

1.We have enlarged Figure 1 and Figure 2 to present the content more clearly.

2.We have emphasized the key numbers in Table 1: the highest and the second-highest values are displayed in bold and underlined.

3.We have added a discussion in Section 3.3 regarding the inconsistencies between LLM-judge evaluations and human evaluations.

4.We have added a discussion in Section 4.1 about the failure modes of advanced agent capabilities.

5.We have added a quantitative analysis of basic failure modes in Section 4.2 and Table 7.

6.We have added a detailed comparison of different benchmarks in Section 6.1 and Table 5.

---

### Author Response · Authors · 2025-11-30
**Summary of Rebuttal & Key Contributions**

We thank the reviewers for their constructive feedback. Three reviewers (kg32, 29pm, tbLU) responded positively, recognizing WideSearch as a rigorous and well-motivated benchmark that addresses the underexplored dimension of "breadth" in information seeking. We have successfully responded to their specific inquiries regarding evaluation stability, temporal invariance, and domain consistency in our rebuttal.
Below, we respectfully clarify the divergence in perspective with Reviewer dPH5.
1. Distinct Research Perspective

Reviewer dPH5 viewed WideSearch as an extension of existing "Deep Search" benchmarks (e.g., BrowseComp). We wish to highlight the fundamental difference in research goals:

- **Depth vs. Breadth**: Existing benchmarks often focus on Deep Reasoning (finding a "hard-to-find" information point via complex reasoning paths). WideSearch focuses on Broad Information Gathering (collecting a comprehensive, reliable set of atomic information points).
- **Productivity Focus**: WideSearch simulates realistic "productivity" scenarios (e.g., market research, data aggregation) where completeness and reliability at scale are paramount. This complements, rather than duplicates, the focus of existing reasoning-heavy benchmarks. According to our user survey, the scenarios of WideSearch are closer to real user scenarios. In contrast, the Browsecomp-type benchmarks (as emphasized in the original paper) are artificially constructed solely for testing model capabilities, and there is a significant gap between them and real-world scenarios.
- **Evaluation Stability & Objectivity**: Benchmarks focused on "Deep Research" reports typically rely entirely on LLM-as-a-judge to evaluate subjective, long-form narratives. As observed in recent studies (and our own analysis), this approach often suffers from instability and lower alignment with human judgment (might below 85%). In contrast, WideSearch leverages structured outputs to implement a hybrid evaluation protocol (combining deterministic rules with a calibrated LLM verifier). This design achieves over 97.8% agreement with human annotators, providing a significantly more reliable and objective standard.
2. On the Evaluation of "Deep Research" Systems

Reviewer dPH5 suggested that the benchmark should accommodate the unstructured reports typical of commercial Deep Research systems (e.g., OpenAI/Gemini Deep Research) rather than enforcing structured Markdown tables.

- **Current Overoptimized Deep Research Systems**: Our preliminary experiments revealed that while commercial "Deep Research" systems excel at writing verbose reports, they struggle to strictly follow simple formatting instructions (output a overall data table), which is even very easy for a small model (such as a 7b llm). We believe this may be due to the presence of an over-optimized agent scaffold and system prompt behind current systems. We have found that the existing DeepResearch system often generates a complex report but fails to output a complete table of data, making it impossible to evaluate its performance on WideSearch. We have attempted to evaluate it, and its performance is so low that it holds no experimental significance.


- **Evalution on End2end Systems in Real World Scenario**: To test the capabilities of real-world commercial systems, we conducted tests on the official websites of OpenAI, Gemini, and Claude (with the web search function activated). This test reflects the performance of current cutting-edge AI systems in WideSearch.

---

> ### Author Response · Authors · 2025-11-30
>
> The key contributions of WideSearch are as follows:
>
> 1. **Pioneering a Novel Paradigm for Broad Information Seeking**: Distinct from existing Deep Search benchmarks that focus on deep reasoning for isolated facts (i.e., "needle-in-a-haystack" problems), WideSearch is the first to systematically define and evaluate agent performance on large-scale, cross-entity information aggregation. This fills a critical gap in evaluating "Completeness" and "Breadth," reflecting real-world productivity scenarios (e.g., comparative data analysis) and offering a vital complement to existing reasoning-heavy benchmarks.
> 2. **Establishing a Robust and Objective Evaluation Framework**: To address the instability and subjectivity often associated with "Deep Research" evaluations—which typically rely entirely on LLM-as-a-judge for scoring long-form reports (often yielding <85% human agreement)—we propose a Hybrid Evaluation Framework. By combining deterministic rules with a calibrated LLM verifier, we achieve 97.8% agreement with human annotators. This approach moves beyond the ambiguity of unstructured report scoring, providing a scientifically rigorous standard for measuring performance on complex, long-context tasks.
> 3. **Identifying "Completeness" as the Critical Bottleneck for Scaling**: Through fine-grained error attribution and test-time scaling analysis, we reveal that the primary limitation of SOTA models is information completeness (especially for long-tail data), rather than individual retrieval accuracy. WideSearch demonstrates that current black-box optimizations or simple tool-use paradigms are insufficient for large-scale aggregation, thereby pinpointing clear directions for future research in long-horizon planning, multi-agent collaboration, and complex context management.

---

### Meta-Review · Area_Chair_hiZ5 · 2026-01-07

**Summary:**

The reviewers generally agree that the paper introduces a well-motivated and carefully constructed benchmark for evaluating agentic broad information-seeking **[kg32, 29pm, tbLU]**.

Reviewers highlight the rigor of the five-stage human-in-the-loop curation pipeline, the bilingual and multi-domain coverage, and the hybrid evaluation framework that achieves high agreement with human judgment **[kg32, 29pm, tbLU]**.

The benchmarking results reveal an important limitation of current agentic systems: they struggle to achieve completeness and reliability at scale, even when many individual facts can be correctly retrieved **[kg32, 29pm, tblU]**.

Reviewer **dPH5** raises concerns about the novelty and scope of the benchmark as a straightforward extension of existing benchmarks and questions the exclusion of advanced deep-research systems. The rebuttal clarifies the distinction between broad information gathering and deep search benchmarks like Browsecomp and explains the evaluation challenges of current deep-research systems, though the concern remains.

Overall, reviewers see the work as useful and timely, with concerns centered on scope, evaluation design choices, and long-term research value rather than correctness or execution quality.

**Reviewer Concerns:**

**1. Novelty and research contribution of the benchmark**
Reviewer dPH5 questions whether the benchmark goes beyond a straightforward extension of existing BrowseComp-style benchmarks and whether it constitutes a substantial research contribution.
Other reviewers take a different view and emphasize that the focus on breadth, completeness, and reliability reflects a qualitatively different evaluation axis that is more closely aligned with real-world information-seeking needs **[kg32, 29pm, tbLU]**.

**2. Evaluation protocol and formatting constraints**
Reviewer dPH5 raises concerns that the structured table output requirement may limit applicability to certain frontier deep-research systems. Reviewer 29pm also notes potential brittleness arising from reliance on pre-specified ground truths. In the rebuttal, the authors clarify the motivation for the hybrid rule-based and LLM-judge evaluation framework and discuss the trade-off between evaluation stability and inclusiveness, although some concerns remain **[29pm, dPH5]**.

**3. Coverage of evaluated systems**
Reviewer dPH5 notes the lack of direct evaluation of frontier deep-research agents and suggests alternative evaluation approaches. Other reviewers consider the range of evaluated single-agent, multi-agent, and end-to-end systems sufficient to support the paper’s conclusions, given the benchmark’s design goals *[kg32, 29pm, tbLU]*.

**Reviewer Scores:**

Reviewer kg32: Unchanged. The reviewer was already positive, and the rebuttal clarified details without affecting the overall assessment.

Reviewer 29pm: Unchanged. The rebuttal addressed evaluation-related questions, but the reviewer’s overall mildly positive stance would likely remain the same.

Reviewer tbLU: Unchanged. The reviewer was positive and aligned with the paper’s goals, and the discussion does not suggest a score change.

Reviewer dPH5: Unchanged. The reviewer explicitly states that their concerns remain and that they maintain their negative score.

---

### Decision · Program_Chairs · 2026-01-26

Accept (Poster)